# Consistency of Physics-Informed Neural Networks for Second-Order Elliptic Equations

**Yuqian Cheng**
Department of
Mathematical Science,
Tsinghua University
cyq21@mails.tsinghua
.edu.cn

**Zhuo Chen**
Department of
Mathematical Science,
Tsinghua University
chenzhuo@tsinghua.edu.cn

**Qian Lin**
Department of Statistics
and Data Science,
Tsinghua University
qianlin@tsinghua.edu.cn

## Abstract

The physics-informed neural networks (PINNs) are widely applied in solving differential equations. However, few studies have discussed their consistency. In this paper, we consider the consistency of PINNs when applied to second-order elliptic equations with Dirichlet boundary conditions. We first provide the necessary and sufficient condition for the consistency of the physics-informed kernel gradient flow algorithm. And then, as a direct corollary, when the neural network is sufficiently wide, we derive a necessary and sufficient condition for the consistency of PINNs based on the neural tangent kernel theory. Additionally, we provide non-asymptotic loss bounds for physics-informed kernel gradient flow and PINN under suitable stronger assumptions. Finally, these results inspire us to construct a notable pathological example in which the PINN method is inconsistent.

## 1 Introduction

The basic settings of data-driven scientific computing problem usually start with a PDE problem in the following form:

$$\mathcal{T}u = f \text{ in } \Omega, \quad \mathcal{B}u = g \text{ on } \partial\Omega, \tag{1}$$

where $\Omega \subset \mathbb{R}^d$ is the domain of the problem, $\mathcal{T}$ is a differential operator, $\mathcal{B}$ summarizes the boundary conditions, and the functions $f$ and $g$ are the non-homogeneous terms of the problem. Assume that (1) has a unique solution $u^*$.

Physics-informed machine learning ([33]) searches for numerical estimators of the solution to (1) by training a network $\hat{u}_\theta$, called a physics-informed neural network (PINN). Ideally, the network $\hat{u}_\theta$ is trained to minimize the *population PINN loss function*, which is defined as the mean square residual of (1):

$$L(\hat{u}_\theta) = \int_{\partial\Omega} |\mathcal{B}\hat{u}_\theta(x) - g(x)|^2 dx + \int_{\Omega} |\mathcal{T}\hat{u}_\theta(y) - f(y)|^2 dy \tag{2}$$

However, this loss function (2) is numerically intractable, as all we can utilize are only finitely many samples $\{(X_i, g(X_i) + \eta_i) : X_i \in \partial\Omega, i = 1, \ldots, N_u\}$ and $\{(Y_j, f(Y_j) + \varepsilon_j) : Y_j \in \Omega, j = 1, \ldots, N_f\}$, where $\eta_i$ and $\varepsilon_j$ are independent noises. Thus, in practice, we train the network to minimize the following empirical version of PINN loss function:

$$\hat{L}(\hat{u}_\theta) = \sum_{i=1}^{N_u} |\mathcal{B}\hat{u}_\theta(X_i) - g(X_i) - \eta_i|^2 + \sum_{j=1}^{N_f} |\mathcal{T}\hat{u}_\theta(Y_j) - f(Y_j) - \varepsilon_j|^2. \tag{3}$$

39th Conference on Neural Information Processing Systems (NeurIPS 2025).

Compared to traditional deep learning methods, physics-informed neural networks (PINNs) feature a unique training process that is guided by physical laws described by partial differential equations. This distinctive approach is expected to combine the impressive approximation power and flexibility of neural networks with the robustness and intepretability of physics-informed modeling. Consequently, a thorough theoretical analysis of PINNs is crucial to provide mathematical guarantees for the efficiency of the corresponding algorithms and to offer guidance for model improvement.

In this paper, we discuss the consistency of PINNs, that is, the convergence of the population PINN loss to zero under appropriate conditions. We focus primarily on the convergence of PINNs for an important PDE problem: the second-order elliptic equation with homogeneous Dirichlet boundary condition. This is a special form of (1) where we take $\mathcal{T}$ to be a second-order elliptic operator $\mathcal{L}$, $\mathcal{B}u = u$ and $g = 0$. To be more specific, we are mainly interested in the following question:

*Question: For the second-order elliptic equation with Dirichlet boundary condition, under what condition does the PINN $\hat{u}_\theta$ converge to the PDE solution $u^*$?*

In the study of the convergence of PINNs, challenges arise when we analyze the training dynamics of PINNs, as the optimization of network training is typically a highly non-convex problem. A recent line of works such as [11, 22, 30, 34] focused on the generalization ability of some sort of empirical minimizer of the (regularized or non-regularized) PINN loss function. Compared with these prior works, in order to put the training dynamics of PINNs into consideration, in this paper we introduce the framework of neural tangent kernel (NTK) ([21]) which connects the network training with kernel-based algorithm [4, 9]. In this framework, a network trained by gradient flow converges to a kernel gradient flow estimator [44, 28, 32]) as the width of network goes to infinity. The convexity of the kernel gradient flow model makes the neural tangent kernel a powerful tool for analyzing training dynamics of wide networks. For example, [25, 27] proved the minimax optimality of wide multi-layer ReLU networks on bounded subdomains of Euclidean spaces.

There exist some previous works that applied NTK theory to analyze PINNs. For example, [40, 17] compute the training gradient flow of infinitely wide PINNs in the form of neural tangent kernel in order to analyze the spectral bias of PINNs. [5] applied the NTK theory and proved a convergence result for the empirical loss of wide two-layer PINNs along the training gradient flow. All of these prior works shed us some light on understanding the dynamics of the training process and analyzing the convergence of PINNs with great network width.

## 1.1 Our contributions

Our main theorem (**Theorem 3.7**) states *the necessary and sufficient condition for the consistency of physics-informed kernel gradient flow*. In this paper, the term "consistency" refers to the convergence of the population PINN loss $L$ to zero, and a formal definition of consistency is given in Definition 3.1. We will show that the physics-informed kernel gradient flow is consistent to the problem (4) if and only if the solution $u^*$ lies in an abstract space $\bar{\mathcal{H}}$ (defined in Definition 3.6). Then, using the technique of NTK (Theorem 5.4), we promote the main theorem to networks, and prove the sufficient and necessary conditions for the consistency of PINN (**Theorem 5.8**).

We list our technical contributions as follows, which provide essential details and key components of our main results:

(1) Although the main theorem is concerned with the convergence of PINN loss (2), we are still interested in the performance of PINNs in terms of mean square loss. As will be shown in **Lemma 3.9**, we prove that the PINN loss can control the mean square loss.

(2) We present novel mathematical techniques to solve the ill-posed convex optimization problems, which is the key to the estimation for the NTK gradient flow in our main results. To the best of our knowledge, this is the first convergence result for ill-posed convex optimization on Hilbert space. See **Lemma 4.2** and its remark for details.

(3) As an application, we prove that, in the case of Poisson equations on sphere and torus, $\bar{\mathcal{H}}$ contains the Sobolev space $H_0^1(\Omega) \cap H^2(\Omega)$, which provides a convenient criterion for verifying whether the PINN method is consistent in this case. We note that this convergence result does not require the source-capacity assumption framework which is commonly adopted in prior works.

(4) As an important corollary of the main theorem, in Section 5.2 we construct a pathological example in which the PINN method is inconsistent.

| | involving analysis of network training dynamics | involving NTK | containing convergence result of the empirical loss $\hat{L}$ | containing convergence result of the population loss $L$ (consistency) |
|---|---|---|---|---|
| This paper | Yes | Yes | Yes (Lemma 4.1 and 4.2) | Yes (Theorem 3.7 and 5.8) |
| [11, 22, 30, 34] | No | No | No | Yes |
| [17, 40] | Yes | Yes | No | No |
| [5, 18, 26, 42] | Yes | Yes | Yes | No |

Table 1: A table comparison between this work and related prior works mentioned above or below.

*Remark* 1.1. (**About the terminology "consistency"**) In this paper, we choose to adopt this term from the learning theory instead of directly using the more common term "convergence" mainly due to the following reasons: 1. The term "consistency" emphasizes more on the convergence of the population loss $L$, and we hope to distinguish the main result (Theorem 3.7) in this work from the convergence results of the empirical loss $\hat{L}$ [5, 18, 26, 42]; 2. We hope to distinguish this work from the existing and forthcoming results that focus on the convergence rate, because the term "consistency", in general, emphasizes more on general results about convergence under weak conditions, while the convergence rates are stronger results under stronger conditions.

## 2 Background

### 2.1 Elliptic equations of second order

Our work focuses on the performance of PINNs on the problem of second-order elliptic equation with homogeneous Dirichlet boundary condition:

$$\mathcal{L}u(y) = f(y), \quad y \in \Omega; \quad u(x) = 0, \quad x \in \partial\Omega, \tag{4}$$

where $\Omega$ is a bounded subdomain of $\mathbb{R}^d$, $f \in L^2(\Omega)$ is the nonhomogeneous term, and $\mathcal{L}$ is a second-order elliptic operator defined in the following divergence form: $\mathcal{L}u(y) = \sum_{i,j=1}^d \partial_i (a_{ij}(y)\partial_j u(y))$, $a_{ij}(y) = a_{ji}(y)$. We assume the strong elliptic condition on $\mathcal{L}$. In other words, there is a positive constant $\lambda > 0$ such that $\sum_{i,j} a_{ij}(y)\xi_i\xi_j \geq \lambda > 0$ for any $\xi \in \mathbb{R}^d$, $|\xi| = 1$ and $y \in \Omega$.

Let $H_0^1(\Omega)$ be the closure of $C_c^\infty(\Omega)$ in the Sobolev space $H^1(\Omega)$. For any $f \in L^2(\Omega)$, the problem (4) has a unique weak solution $u^* \in H_0^1(\Omega)$ (see Section 8.2 of [19]).

Denote as $\partial\Omega$ the boundary of $\Omega$, and as $\bar{\Omega}$ the closure of $\Omega$. Assume that $\partial\Omega$ is smooth, and $a_{ij} \in C^\infty(\bar{\Omega})$. Then by the regularity theorem (Section 6.3 of [14]), for any $f \in L^2(\Omega)$, the weak solution $u^*$ is in the Sobolev space $H^2(\Omega)$. Moreover, the solution $u^*$ is also a strong solution (Chapter 9 of [19]). In other words, $u^* \in C(\bar{\Omega})$, $u^* = 0$ on $\partial\Omega$ and $\mathcal{L}u^* = f$ a.e. in $\Omega$.

### 2.2 Physics-Informed Neural Networks

We train a neural network $\hat{u}_\theta$ to estimate the ground-truth solution $u^*$. The network $\hat{u}_\theta$ is set in the following structure:

$$z_p^0(x) = W_p^0 x + b_p^0, \quad W_p^0 \in \mathbb{R}^{m_1 \times d}, \quad b_p^0 \in \mathbb{R}^{m_1}, \quad p = 1, 2;$$

$$z_p^{l+1}(x) = \frac{1}{\sqrt{m_l}} W_p^l \sigma(z^{(l,p)}(x)) + b_p^l, \quad W_p^l \in \mathbb{R}^{m_l \times m_{l+1}}, b_p^l \in \mathbb{R}^{m_{l+1}}, \quad l = 1, \ldots, L, p = 1, 2;$$

$$\hat{u}_\theta(x) = \frac{1}{\sqrt{2}}(z_1^{L+1}(x) - z_2^{L+1}(x)), \quad m_{L+1} = 1.$$

$$\tag{5}$$

When $L = 1$, $\hat{u}_\theta$ is actually a fully-connected network with one hidden layer with a special initialization [25]. We also note that the NNK and NTK of $\hat{u}_\theta$ coincide with those of a fully-connected network (see Appendix F.3 for details). We initialize the parameters $\theta = (W_p^l, b_p^l)$ of $\hat{u}_\theta$ in the following form

of *mirror initialization*, which is suggested in [8, 25, 27]:

$$(W_1^l)_{i,j} = (W_2^l)_{i,j} \sim N(0,1), \quad (b_1^l)_i = (b_2^l)_i \sim N(0,1), \quad l = 0,\ldots,L. \tag{6}$$

This special initialization is adopted in this paper for only one purpose: to ensure that $\hat{u}_\theta = 0$ at the beginning of training. In fact, the choice of initialization is a delicate matter that needs careful attention in NTK theory, because it is proved in [8] that an infinitely wide fully-connected network will not lie in the RKHS of its NTK at initial time, if all parameters are initialized as i.i.d. standard Gaussian random variables, in which case the NTK kernel gradient flow will no longer be mathematically well-defined. Thus, we adopted the special initialization (6) to avoid unnecessary disgression.

We also assume that the activation function $\sigma$ is sufficiently smooth such that $\mathcal{L}\hat{u}_\theta$ is well-defined.

Under the setting of (4), the PINN loss function $L$ defined in (2) is now written in the following form:

$$L(u) = \int_{\partial\Omega} |u(x)|^2 dx + \int_\Omega |\mathcal{L}u(y) - f(y)|^2 dy. \tag{7}$$

(Here we set both the total measures of $\partial\Omega$ and $\Omega$ to be 1 after necessary rescaling.) The corresponding empirical loss function is

$$\hat{L}(u) = \frac{1}{N_u} \sum_{i=1}^{N_u} |u(X_i)|^2 + \frac{1}{N_f} \sum_{j=1}^{N_f} |\mathcal{L}u(Y_j) - f(Y_j) - \varepsilon_j|^2, \tag{8}$$

where $X_i$ and $Y_j$ are i.i.d. sample points drawn from the uniform distributions of $\partial\Omega$ and $\Omega$, respectively, and $\varepsilon_j$ are independent noises. (Here we set the noise $\eta_i$ on the boundary in (3) to be zero.)

Since the dimension of $\partial\Omega$ is smaller than the dimension of $\Omega$, we further assume that $N_u \leq N_f$.

## 2.3 Gradient Flow of Network Training and Neural Tangent Kernel

During training, the network parameter $\theta$ evolves along the following gradient flow:

$$\frac{d}{dt}\theta(t) = -\nabla_\theta \hat{L}(\hat{u}_{\theta(t)}). \tag{9}$$

Equivalently, the gradient flow of the function $\hat{u}_\theta$ is given by

$$\begin{aligned}
\frac{d}{dt}\hat{u}_{\theta(t)}(x) &= (\nabla_\theta \hat{u}_{\theta(t)}(x))^T \cdot \frac{d}{dt}\theta(t) \\
&= -\frac{2}{N_u} \sum_{i=1}^{N_u} K_{\theta(t)}(x, X_i) \cdot \hat{u}_{\theta(t)}(X_i) - \frac{2}{N_f} \sum_{j=1}^{N_f} \mathcal{L}_y K_{\theta(t)}(x, Y_j) \cdot (\mathcal{L}\hat{u}_{\theta(t)}(Y_j) - f(Y_j) - \varepsilon_j),
\end{aligned} \tag{10}$$

where $K_\theta(x,y) = (\nabla_\theta \hat{u}_\theta(x))^T \cdot \nabla_\theta \hat{u}_\theta(y)$ is called the neural network kernel (NNK) function.

Here, the notation $\mathcal{L}_y K_{\theta(t)}(x,y)$ that means $\mathcal{L}$ is operated only on the variable $y$. The notation $\mathcal{L}_x$ is defined in the similar way, and clearly $\mathcal{L}_x \mathcal{L}_y K_{\theta(t)}(x,y) = \mathcal{L}_y \mathcal{L}_x K_{\theta(t)}(x,y)$.

We will show in Section 5 that the NNK function $K_{\theta(t)}(x,y)$ converges to the NTK function $K_{NT}(x,y)$ along the PINN gradient flow (10). Moreover, the derivatives of $K_{\theta(t)}$ also converges to the derivatives of $K_{NT}$ respectively. Thus, it is reasonable to consider the following limit gradient flow:

$$\frac{d}{dt}\hat{u}_t(x) = -\frac{2}{N_u} \sum_{i=1}^{N_u} K_{NT}(x, X_i) \cdot \hat{u}_t(X_i) - \frac{2}{N_f} \sum_{j=1}^{N_f} \mathcal{L}_y K_{NT}(x, Y_j) \cdot (\mathcal{L}\hat{u}_t(Y_j) - f(Y_j) - \varepsilon_j). \tag{11}$$

## 2.4 Reproducing Kernel Hilbert Space

Suppose that we have a positive-definite kernel function $K(x,y)$ In other words, the integration operator $T : L^2(\Omega) \to L^2(\Omega)$, $Tf(x) = \int_\Omega K(x,y)f(y)dy$ has eigenvalues $\lambda_0 \geq \lambda_1 \geq \lambda_2 \geq$

$\cdots > 0$. Let $\phi_i$ be the corresponding eigenfunction of the eigenvalue $\lambda_i$. By Mercer decomposition ([38]), $K(x,y) = \sum_{i=0}^{\infty} \lambda_i \phi_i(x) \phi_i(y)$ converges absolutely and uniformly.

Denote $K_y(x) = K(x,y)$, and define an inner product $\langle \cdot, \cdot \rangle_{\mathcal{H}}$ on the space $\{K_y : y \in \bar{\Omega}\}$ by $\langle K_{y_1}, K_{y_2} \rangle_{\mathcal{H}} = K(y_1, y_2)$. The closure of $\{K_y\}$ under this inner product is denoted as $\mathcal{H}$, called the reproducing kernel Hilbert space (RKHS) of $K$. This space is characterized as

$$\mathcal{H} = \left\{ f(x) = \sum_{i=0}^{\infty} f_i \sqrt{\lambda_i} \phi_i(x) : f_i \in l^2 \right\}, \tag{12}$$

and the inner product can be characterized by $\langle f, g \rangle_{\mathcal{H}} = \sum_{i=0}^{\infty} f_i g_i$ for $f = \sum_{i=1}^{\infty} f_i \sqrt{\lambda_i} \phi_i$ and $g = \sum_{i=1}^{\infty} g_i \sqrt{\lambda_i} \phi_i$ ([9]). Thus, the Hilbert space $\mathcal{H}$ has a natural isomorphism to $l^2$.

### 2.5 Physics-Informed Kernel Gradient Flow

With the language of RKHS, the limit gradient flow (11) can be interpreted in the following sense: Suppose that we aim to search for an estimator of the ground-truth solution $u^*$ in the RKHS of a kernel $K(x,y)$. Give $u(x) = \sum_{i=0}^{\infty} u_i \sqrt{\lambda_i} \phi_i(x) \in \mathcal{H}$, we will prove later that this expansion can be differentiated term by term: $\mathcal{L}u(x) = \sum_{i=0}^{\infty} u_i \sqrt{\lambda_i} \mathcal{L}\phi_i(x)$. Consider $\mathcal{H}$ as a parametric model space with parameters $\{u_i\}_{i=0}^{\infty}$. In other words, we parametrize an estimator $\hat{u}_t \in \mathcal{H}$ by $\hat{u}_t(x) = \sum_{i=0}^{\infty} \hat{u}_i(t) \sqrt{\lambda_i} \phi_i(x)$. Then the gradient flow of the parameters along the time $t \in [0,\infty)$ is $\frac{d}{dt}\hat{u}_i(t) = \frac{\partial \hat{L}}{\partial u_i}(\hat{u}_t)$, and equivalently, the gradient flow of the function $\hat{u}_t$

$$\begin{aligned}
\frac{d}{dt}\hat{u}_t(x) &= \sum_{i=0}^{\infty} \frac{\partial \hat{u}_t}{\partial u_i} \cdot \frac{d}{dt}\hat{u}_i(t) = -\sum_{i=0}^{\infty} \frac{\partial \hat{u}_t}{\partial u_i} \cdot \frac{\partial \hat{L}}{\partial \hat{u}_i}(\hat{u}_t) \\
&= -\frac{2}{N_u} \sum_{i=1}^{N_u} K(x, X_i) \cdot \hat{u}_t(X_i) - \frac{2}{N_f} \sum_{j=1}^{N_f} \mathcal{L}_y K(x, Y_j) \cdot (\mathcal{L}\hat{u}_t(Y_j) - f(Y_j) - \varepsilon_j).
\end{aligned} \tag{13}$$

We call this method the *physics-informed kernel gradient flow*. If we set the kernel function $K(x,y)$ to be the NTK function $K_{NT}(x,y)$, then (13) is precisely the gradient flow (11). Thus, the training of an infinitely wide PINN can be interpreted as the physics-informed kernel gradient flow of NTK.

## 3 Main Results

Before we state our main results, we need to define the consistency formally:

**Definition 3.1.** (1) We say that the physics-informed kernel gradient flow (13) is consistent to the problem (4), if $\lim_{N_u, N_f \to \infty} L(\hat{u}_T) = 0$ for a suitable training stopping time $T = T(N_u, N_f)$ depending on $N_u$ and $N_f$.

(2) We say that the PINN method (described in Subsection 2.2) is consistent to the problem (4), if $\lim_{N_u, N_f \to \infty} L(\hat{u}_{\theta(T)}) = 0$ for a suitable training stopping time $T = T(N_u, N_f)$ and suitable network width $m_i = m_i(N_u, N_f)$, $i = 1, \ldots, L$ depending on $N_u$ and $N_f$.

In other words, consistency means that we can select appropriate hyperparameters (network widths and stopping time) such that the PINN loss can be controlled to be arbitrarily small, provided that the sample size is sufficiently great.

*Remark* 3.2. The PINN loss function $L$ defines a natural metric

$$d_{pinn}(u_1, u_2)^2 = \int_{\partial \Omega} |u_1 - u_2|^2 + \int_{\Omega} |\mathcal{L}u_1 - \mathcal{L}u_2|^2, \tag{14}$$

called the PINN distance, and $L(u) = d_{pinn}(u, u^*)^2$, hence it is reasonable that we use the PINN loss $L(\hat{u}_{\theta(t)})$ in Definition 3.1 to measure the distance between the PDE solution and the solution estimator. As will be shown in Lemma 3.9, the PINN distance can control the $L^2$ distance.

We list the assumptions required as follows:

**Assumption 3.3.** The kernel function $K(x,y)$ is positive-definite.

**Assumption 3.4.** $K(x,y)$ is smooth enough so that $\partial_x^\alpha \partial_y^\beta K(x,y)$ is well-defined for any multi-indices $\alpha, \beta$ with $|\alpha|, |\beta| \leq 2$.

**Assumption 3.5.** Assume that the term $f$ in (4) is measurable and bounded, hence is also $L^2$.

**Definition 3.6.** Let $\mathcal{H}$ be the RKHS of $K(x,y)$, and let $\bar{\mathcal{H}}$ be the closure of $\mathcal{H}$ in the Sobolev space $H^2(\Omega)$ under the metric defined by (14).

We will show in Theorem A.1 that under Assumption 3.3 and Assumption 5.2, we have $\mathcal{H} \subset C^2(\bar{\Omega})$, hence $\bar{\mathcal{H}}$ is well-defined.

### 3.1 Sufficient and necessary conditions of consistency

Now we state our main theorem:

**Theorem 3.7. (Consistency of physics-informed kernel gradient flow)** *Let $u^*$ be the unique solution to (4), and let $\hat{u}_t$ be the kernel gradient flow estimator defined in (13. Assume that the samples $\{X_1, \ldots, X_{N_u}\}$ and $\{Y_1, \ldots Y_{N_f}\}$ are drawn from the uniform distributions of $\partial\Omega$ and $\Omega$, respectively, and $N_u \leq N_f$. We further assume that $\varepsilon_j$ are independent sub-Gaussian noises.*

*Suppose that Assumption 3.3, 3.4 and 3.5 are all satisfied. Let $\bar{\mathcal{H}}$ be the space defined in Definition 3.6. Then we have:*

*(1) If $u^* \in \bar{\mathcal{H}}$, then by setting the training stopping time to be $T = \Theta(N_u^\alpha)$ for $\alpha \in (0, \frac{1}{3})$, we have*

$$\lim_{N_u, N_f \to \infty} L(\hat{u}_T) \to 0 \tag{15}$$

*in probability, hence by Definition 5.2, the physics-informed kernel gradient flow is consistent in this case;*

*(2) If $u^* \notin \bar{\mathcal{H}}$, then*

$$\inf_{t, X_i, Y_j} L(\hat{u}_t) \geq C > 0 \tag{16}$$

*for some positive constant $C$ depending on $\Omega$, $\mathcal{L}$, $f$ and $K$. Here, the infimum is taken over all $t \in [0, \infty)$ and all possible choices of samples $X_i, Y_j$.*

*Remark* 3.8. A trivial consequence of (16) is that, if $u^* \notin \bar{\mathcal{H}}$, then for any possible stopping time $T = T(N_u, N_f)$, we have

$$\liminf_{N_u, N_f \to \infty} L(\hat{u}_T) \geq C > 0; \tag{17}$$

In other words, the physics-informed kernel gradient flow is always inconsistent in this case. Thus, Theorem 3.7 indicates that $u^* \in \bar{\mathcal{H}}$ is the necessary and sufficient condition for the consistency of physics-informed kernel gradient flow.

### 3.2 The Convergence of Mean Square Loss

It is natural to ask how close $\hat{u}_{\theta(t)}$ and the solution $u^*$ are in terms of mean square loss. Fortunately we have the following $L^2$ norm estimation:

**Lemma 3.9. (PINN distance controls $L^2$ distance)** *For any $u \in C^2(\bar{\Omega})$, we have*

$$L(u) \geq C\|u - u^*\|_{L^2}^2, \tag{18}$$

*where the constant $C > 0$ depends only on $\mathcal{L}$ and $\Omega$.*

*Proof.* The proof is based on the technique of Green's function. See Appendix D for details. $\square$

*Remark* 3.10. This estimation for $L^2$ loss is nontrivial, and the standard elliptic estimation for weak solutions is far from sufficient to obtain Lemma 3.9, because the kernel gradient flow estimator $\hat{u}_t$ or the PINN estimator $\hat{u}_{\theta(t)}$ does not satisfy the Dirichlet boundary condition generally.

Since we will shown in Theorem A.1 that $\mathcal{H} \subset C^2(\bar{\Omega})$, then we immediately obtain

**Corollary 3.11. ($L^2$ convergence)** *Under the same assumptions of* Theorem 3.7, *if PINN is consistent to (4), i.e. $u^* \in \bar{\mathcal{H}}$, then we also have*

$$\lim_{N_u, N_f \to \infty} \lim_{m \to \infty} \|\hat{u}_T - u^*\|_{L^2(\Omega)} \to 0 \quad \text{in probability.} \tag{19}$$

### 3.3 Non-Asymptotic Bound

Under the settings of our Theorem 3.7, the condition for consistency we provide is necessary and sufficient, which means that we characterize *all* of the cases where $L(\hat{u}_t)$ converges to 0, including those very bad cases. In other words, the convergence speed can be arbitrarily low, hence it is impossible to obtain a non-asymptotic bound under the same settings of our main theorem.

However, if suitable stronger assumptions hold, then a non-asymptotic bound is accessible, as is shown in the following theorem:

**Theorem 3.12.** *Under the same settings of Theorem 3.7, if we further assume that $u^* \in \mathcal{H}$, then with probability at least $1 - \mathcal{O}(N_u^{-1})$, we have*

$$L(\hat{u}_T) = \mathcal{O}\left(N_u^{-\frac{1-3\alpha}{2}}\sqrt{\log N_u} + N_u^{-\alpha}\right), \quad \|\hat{u}_T - u^*\|_{L^2(\Omega)} = \mathcal{O}\left(N_u^{-\frac{1-3\alpha}{2}}\sqrt{\log N_u} + N_u^{-\alpha}\right) \tag{20}$$

*at stopping time $T = \Theta(N_u^\alpha)$, $\alpha \in (0, \frac{1}{3})$. Here, the convention $\mathcal{O}$ hides all the terms involving $K$, $\Omega$, $\mathcal{L}$ and $f$.*

*Proof.* See Appendix E $\qquad\square$

*Example* 3.13. A typical example of RKHS is the Sobolev space $H^r(\bar{\Omega})$ for $r > \frac{d}{2}$ [13]. Thus, a Sobolev kernel of $H^r(\bar{\Omega})$ with $r$ will satisfy the condition $u^* \in \mathcal{H}$.

## 4 Proof of the Main Theorem

In contrast with traditional methods such as kernel ridge regression [9] and kernel gradient flow [44] where estimators can be explicitly computed and directly analyzed, the peculiar form of the PINN loss function complicates the analysis of the gradient flow of network training. Many previous works on kernel-based physics-informed modeling such as [24, 10, 41, 31] ignored the boundary term for convenience, while our work put the boundary condition into consideration. We estimate the gradient flow of PINN training in an indirect taste by studying the evolution of PINN loss function and approximating its derivative with respect to training time.

Our proof of Theorem (3.7) is based on the decomposition $L(\hat{u}_t) = (L(\hat{u}_t) - L(v_t)) + L(v_t)$, where $v_t$ is defined as the kernel gradient flow of the population loss (7):

$$\frac{d}{dt}v_t(x) = -2\int_{\partial\Omega} K(x,\xi) \cdot v_t(\xi)d\xi - 2\int_\Omega \mathcal{L}_y K(x,\eta) \cdot (\mathcal{L}v_t(\eta) - f(\eta))d\eta. \tag{21}$$

The following two lemmata provide estimations for $L(\hat{u}_t) - L(v_t)$ and $L(v_t)$, respectively:

**Lemma 4.1.** *With probability $1 - \mathcal{O}(N_u^{-1} + N_f^{-1})$, we have*

$$|L(\hat{u}_t) - L(v_t)| = \mathcal{O}\left(T^{\frac{3}{2}}(\sqrt{\frac{\log N_f}{N_f}} + \sqrt{\frac{\log N_u}{N_u}})\right) \tag{22}$$

*for all $t \in [0, T]$. Here, the randomness comes from the selection of samples.*

**Lemma 4.2.** *Along the gradient flow (21), we have $L(v_t) \to \inf_{v \in \mathcal{H}} L(v)$ as $t \to \infty$. Thus, $\lim_{t\to\infty} L(v_t) = 0$ if and only if $u^* \in \bar{\mathcal{H}}$.*

The proofs of the two lemmata are delayed to Appendix B and C, respectively.

*Remark* 4.3. Although Lemma 4.2 looks quite natural at first glance, the convergence of $L(v_t)$ is in fact highly nontrivial. This is because we do not assume that the minimizer $u^*$ lies in the model space $\mathcal{H}$, but rather in a larger space $\bar{\mathcal{H}}$, which makes the optimization problem ill-posed. Traditional results of convex optimization ([6] for example) are usually based on the well-posedness; In other words, they assume that there exists a minimizer of the target convex function in the interior of the model domain. Lemma 4.2 shows that the PINN loss still converges to its infimum, even though there is no minimizer in the RKHS $\mathcal{H}$ and the RKHS norm of $v_t$ diverges to infinity along the gradient flow (21). To the best of our knowledge, this is the first convergence result for a specific ill-posed convex optimization problem on infinite-dimensional Hilbert space. In fact, the convergence of general ill-posed convex optimization on Hilbert space remains an open problem.

Now we are ready to prove the main theorem (Theorem 3.7):

*Proof.* **(of the main theorem)** If $u^* \in \bar{\mathcal{H}}$, we take the stopping time to be $T = \Theta(N_u^\alpha)$, $\alpha \in (0, \frac{1}{3})$. By Lemma 4.1, $\lim_{N_u, N_f \to \infty} (L(\hat{u}_T) - L(v_T))) = 0$ in probability; And by Lemma 4.2, $\lim_{N_u, N_f \to \infty} L(v_T) = \lim_{t \to \infty} L(v_t) = 0$ if and only if $u^* \in \bar{\mathcal{H}}$. Combining the three results above, we obtain (15).

If $u^* \notin \bar{\mathcal{H}}$, since $\hat{u}_t \in \mathcal{H}$ for any $t \geq 0$ and any possible choices of samples $X_i, Y_j$, then $L(\hat{u}_t) \geq \inf_{v \in \mathcal{H}} L(v) > 0$. Combining this estimation with Lemma 5.6, we obtain (16). $\square$

## 5 Results on PINN

As was discussed in Section 2, the technique of NTK would help us promote results about kernel methods to wide neural networks. In this section, we first prove the convergence of NTK (**Theorem 5.4**), based on which we extend our main theorem (Theorem 3.7) to PINNs (see **Theorem 5.8**).

The following assumptions are the analogues of Assumption 3.3 and 3.4 with respect to NTK:

**Assumption 5.1.** Assume that the NTK function $K_{NT}$ (described in 2.3) is positive-definite.

**Assumption 5.2.** Assume that $\sigma \in C^3(\mathbb{R})$. Moreover, the derivatives of $\sigma$ are polynomially bounded: $|\sigma^{(i)}(r)| \leq C(1 + |r|^p)$, $i = 0, 1, 2, 3$ for some university constants $C > 0$ and $p \geq 1$.

*Remark* 5.3. Many of the previous works have proposed criteria to verify the positiveness of neural tangent kernel $K_{NT}(x, y)$, such as [7]. The smoothness of $\sigma$ is a fundamental assumption to ensure that the derivatives of the network $\hat{u}_{\theta(t)}$ are well-defined. The polynomial bound condition is easily satisfied by various types of activation functions, such as tanh and ReLU$^k$ for $k > 3$.

**Theorem 5.4. (Uniform convergence of NTK)** *Assume that* Assumption 5.1 *and* 5.2 *are satisfied.*

*(I) For a shallow network with only one hidden layer ($L = 1$ in (145)), as the network width $m_1 \to \infty$, the NNK function $K_{\theta(t)}(x, y)$ converges in probability to the NTK function $K(x, y)$ uniformly with respect to $x, y \in \bar{\Omega}$ and $t \in [0, T]$. In other words,*

$$\sup_{t \in [0,T], \, x,y \in \bar{\Omega}} |K_{\theta(t)}(x, y) - K_{NT}(x, y)| \to 0 \quad \text{in probability} \tag{23}$$

*as $m \to \infty$. Here, the randomness arises from the initialization of network.*

*Moreover, $\mathcal{L}_x K_{\theta(t)}(x, y)$, $\mathcal{L}_y K_{\theta(t)}(x, y)$, $\mathcal{L}_x \mathcal{L}_y K_{\theta(t)}(x, y)$ also converge in probability to $\mathcal{L}_x K_{NT}(x, y)$, $\mathcal{L}_y K_{NT}(x, y)$, $\mathcal{L}_x \mathcal{L}_y K_{NT}(x, y)$ respectively, and the convergence is uniform with respect to $t \in [0, T]$, $x, y \in \bar{\Omega}$.*

*(II) Furthermore, if $\Omega$ is convex, then the above results also holds for deep neural networks with depth $L \geq 9$ and activation function $\sigma = \tanh$, as the network widths satisfies $cm \leq m_1, \ldots, m_L \leq Cm$ for some $0 < c < C$ and $m \to \infty$.*

*Proof.* The proof is delayed to Appendix F. $\square$

An experimental illustration for Theorem 5.4 (I) is provided in Figure 1.

*Remark* 5.5. Theorem 5.4 (I) is the uniform version of Theorem 4.4 of [40]. We expect that Theorem 5.4 (II) remains valid for $L = 2, 3, \ldots, 8$, while we defer the detailed proof to future work due to its technical complexity in computations. We offer numerical results that roughly verify Theorem 5.4 (II) for $L = 4$ in Appendix I.

Note that both (10) and (11) are finite-order linear ODE systems. With the help of Theorem 5.4 and ODE theory, we show that the distance between the network $\hat{u}_{\theta(t)}$ and its corresponding physics-informed kernel gradient flow $\hat{u}_t$ converges in probability to 0:

**Lemma 5.6.** *Under the same assumptions of Theorem 5.4, given $N_u$, $N_f$ and $T > 0$, as $m \to \infty$, we have*

$$\sup_{x \in \bar{\Omega}, \, t \in [0,T]} \sup_{X_i, Y_j} |\hat{u}_{\theta(t)}(x) - \hat{u}_t(x)| \to 0, \qquad \sup_{x \in \bar{\Omega}, \, t \in [0,T]} \sup_{X_i, Y_j} |\mathcal{L}\hat{u}_{\theta(t)}(x) - \mathcal{L}\hat{u}_t(x)| \to 0 \tag{24}$$

*in probability. Here, the randomness arises from the initialization of network.*

*Proof.* The proof is delayed to Appendix G. □

*Remark* 5.7. Both $\hat{u}_{\theta(t)}(x)$ and $\hat{u}_t(x)$ depend on the samples $X_i$, $Y_j$, and the notation $\sup_{X_i,Y_j}$ in Lemma 5.6 emphasizes the fact that the convergence is uniform with respect to all possible choices of samples $X_i$, $Y_j$ with sample sizes $N_u$, $N_f$ fixed.

The experimental illustrations of this lemma can be found in Figure 2 and 3. A quantitative version of this lemma for shallow networks ($L = 1$) can be found in (330) in the appendix.

As a direct corollary of Lemma 5.6 and the main theorem (Theorem 3.7), we have the following theorem, which states the necessary and sufficient condition for the consistency of PINN:

**Theorem 5.8. (Consistency of PINN)** *Let $u^*$ be the unique solution to (4), and let $\hat{u}_{\theta(t)}$ be the neural network in the form (145). Assume that the samples $\{X_1, \ldots, X_{N_u}\}$ and $\{Y_1, \ldots Y_{N_f}\}$ are drawn from the uniform distributions of $\partial\Omega$ and $\Omega$, respectively, and $N_u \leq N_f$.*

*Let $\mathcal{H}_{NTK}$ be the RKHS of $K_{NT}$, and let $\bar{\mathcal{H}}_{NTK}$ be the closure of $\mathcal{H}_{NTK}$ under the metric $d_{pinn}$ (14). Then PINN is consistent to the problem (4) if and only if $u^* \in \bar{\mathcal{H}}_{NTK}$.*

## 5.1 A Positive Example

Unfortunately, the computation of eigenfunctions of general NTK is still an open problem, hence Theorem 5.8 is not applicable to general network setting at present. However, we note that in many previous works on physics-informed modeling such as [24, 10, 41], a co-diagonalization condition, which states that the kernel function $K$ and the reverse of elliptic operator share the same eigenspaces, was assumed to simplify the computation. Typical examples of kernel functions satisfying the co-diagonalization assumption are the inner-product kernels on two-point homogeneous spaces [3], including spheres, real and complex projective spaces, etc. These previous works inspire us to construct the following example.

Consider the Poisson equation problem:

$$\Delta u = f \quad \text{on } M, \quad u(p) = u_p, \tag{25}$$

where the manifold $M$ is set to be $S^d \subset \mathbb{R}^{d+1}$, the $d$-dimensional sphere, or $M = \mathbb{T}^d = \mathbb{S}^1 \times \cdots \times \mathbb{S}^1 \subset \mathbb{R}^{2d}$, the $d$-dimensional torus; $p$ is a fixed point on $M$; $u_p \in \mathbb{R}$ is a fixed value; $f$ is an $L^2$ function on $M$ such that $\int_M f = 0$. $\Delta$ is the Laplace-Beltrami operator on $M$, and it is a classical result that the problem (25) has a unique solution $u^* \in H^2(M)$ (see [2] for example).

In this case, the PINN loss function of (25) and the corresponding empirical loss are in the form

$$L_M(u) = |u(p)|^2 + \int_M |\Delta u - f|^2, \quad \hat{L}_M(u) = |u(p)|^2 + \frac{1}{N_f}\sum_{j=1}^{N_f} |\Delta u(Y_j) - f(Y_j) - \varepsilon_j|^2, \tag{26}$$

where $Y_i$ are i.i.d. samples on $M$. Define a metric on $C^2(M)$ by setting

$$d_{pinn}^M(u_1, u_2)^2 = |u_1(p) - u_2(p)|^2 + \int_M |\Delta u_1 - \Delta u_2|^2, \tag{27}$$

then $L(u) = d_{pinn}^M(u, u^*)^2$. We still denote the RKHS of NTK function $K$ as $\mathcal{H}$, and the closure of $\mathcal{H}$ in $H^2(M)$ under the metric $d_{pinn}^M$ as $\bar{\mathcal{H}}$. Since $\mathbb{S}^d$ is a bounded smooth subspace of $\mathbb{R}^{d+1}$ and $\mathbb{T}^d$ is a bounded smooth subspace of $\mathbb{R}^{2d}$, all of the results in Section 3 still hold true for this specific problem (25). Specifically, $\mathcal{H}$ is contained in $C^2(M)$, hence $\bar{\mathcal{H}}$, the closure of $\mathcal{H}$ in $H^2(M)$, is well-defined. Moreover, we can show that for any bounded measurable $f$, the solution $u^*$ is in $\mathcal{H}$:

**Theorem 5.9.** *Suppose that* Assumption 3.3, Assumption 5.2 *and* Assumption 3.5 *are all satisfied. Then $u^* \in \bar{\mathcal{H}}_{NTK}$, hence PINN is always consistent to the problem (25). In other words, for $T = \Theta(N_f^\alpha)$, $\alpha \in (0, \frac{1}{3})$, we have*

$$\lim_{N_f\to\infty}\lim_{m\to\infty} L_M(\hat{u}_{\theta(t)}) = 0, \quad \lim_{N_f\to\infty}\lim_{m\to\infty} \|\hat{u}_{\theta(t)} - u^*\|_{L^2(M)} = 0 \tag{28}$$

*in probability, as $m$, $N_f$, $T \to \infty$.*

*Proof.* See Appendix H. An experimental illustration of this theorem can be found in Figure 5. □

*Remark* 5.10. The strength of this theorem lies in its ability to prove consistency even under very weak assumptions. Specifically, we only require $f$ to be bounded and measurable and do not assume the typical capacity-source conditions which are adopted in many important prior works such as [46, 45, 15, 35, 37]. Furthermore, the boundedness assumption of $f$ can be relaxed in some special cases, as will be shown in an example in H.2. It is worth noting that in this example, the kernel gradient flow method is consistent while the convergence speed of PINN loss can be arbitrarily low.

*Remark* 5.11. Although the results in **Section 3** requires $N_u \to \infty$, taking only one boundary sample $p$ does not make any trouble here, because taking $N_u = 1$, $X_1 = p$ is equivalent with taking $N_u = N_f$ and $X_1 = \cdots = X_{N_f} = p$: both can reach the same form of the empirical loss (26).

### 5.2 A Negative Example

Consider the following Poisson equation problem on the 1-dimensional ring:

$$\Delta u = 2 \quad \text{on } \mathbb{S}^1 - \{p\}; \quad u(p) = 0, \tag{29}$$

where $p = e^{-i} = (-1, 0)$ is a fixed point on $\mathbb{S}^1$. This problem has a unique solution $u^*(e^{i\pi r}) = \pi^2(r^2 - 1)$, $r \in (-1, 1)$. Note that $u^*$ is in $H^1(\mathbb{S}^1)$ but not in $H^2(\mathbb{S}^1)$, because its first derivative has a discontinuous point at $p$. In this case, we can show that the solution $u^*$ is not in $\bar{\mathcal{H}}$:

**Proposition 5.12.** *The solution $u^*$ to* (29) *has a positive distance away from the RKHS $\mathcal{H}$ of NTK function $K$, hence does not lie in $\bar{\mathcal{H}}$.*

*Moreover, the infimum of the PINN loss is $\inf_{u \in \mathcal{H}} L(u) = L(0) = 4 > 0$, and the kernel gradient flow of the PINN loss* (21) *is solved as $v_t(x) = 0$ for all $x \in \mathbb{S}^1$ and $t \in [0, \infty)$.*

The proof can be found in Appendix H.1. An experimental illustration of this result is provided in Figure 6 in the appendix. Thus, by Theorem 3.7, the PINN method is inconsistent to the problem (29). To be more explicit, as an immediate corollary, for any fixed time $t < \infty$, the PINN estimator $\hat{u}_{\theta(t)}$ converges to 0 as $m, N_f \to \infty$, hence stays away from the PDE solution $u^*$.

*Remark* 5.13. The existence of the negative example (29) highlights the risks of directly applying PINNs to specific problems without theoretical assessment of reliability. We notice that [29] also provided a negative example of the inconsistency of PINNs. Their example involves an elliptic equation with discontinuous coefficients: in contrast, the equation in our example has continuous coefficients, while it is solved on an unusual space ($\mathbb{S}^1$ with a point removed).

## 6 Discussions and Conclusion

In this paper, we establish the sufficient and necessary conditions for the consistency of physics-informed kernel gradient flow and sufficiently wide PINN in general cases. The NTK framework serves as a crucial bridge between kernel methods and wide neural networks, while the characterization of the eigenspaces of NTK remains an open problem. A deeper understanding of kernel functions and RKHS will facilitate the extension of our approach to other significant problems in more general settings. On the other hand, the optimization problem of narrow networks is also a difficult but important problem which this paper does not involve. We leave these valuable problems to future research.

## Acknowledgements

Zhuo Chen is supported in part by National Natural Science Foundation of China (Grant 12071241). Qian Lin is supported in part by National Natural Science Foundation of China (Grant 92370122, Grant 11971257) and the Beijing Natural Science Foundation (Grant Z190001).

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

# A Smoothness of RKHS

In this section, we prove an important theorem, which verifies the well-definedness of the physics-informed kernel gradient flow estimator (13), the population gradient flow (21) and the space $\bar{\mathcal{H}}$ defined in Definition 3.6.

**Theorem A.1.** (Smoothness of RKHS) *Assume that* Assumption 3.3 *and* Assumption 5.2 *are satisfied. Then any $u \in \mathcal{H}$ is also in $C^2(\bar{\Omega})$;*

*Moreover, if the RKHS expansion of $u \in \mathcal{H}$ is given by $u(x) = \sum_{i=0}^{\infty} u_i \sqrt{\lambda_i} \phi_i(x)$, then $u$ can be twice differentiated term by term; to be more specific,*

$$\nabla^l u(x) = \sum_{i=0}^{\infty} u_i \sqrt{\lambda_i} \nabla^l \phi_i(x), \quad l = 1, 2. \tag{30}$$

The proof of this theorem is divided into the following series of lemmata:

**Lemma A.2.** *The eigenfunctions $\phi_i$ of $K$ are $C^2(\bar{\Omega})$.*

*Proof.* It is a direct consequence of the equality

$$\phi_i(x) = \frac{1}{\lambda_i} \int_{\Omega} K(x, y) \phi_i(y) dy. \tag{31}$$

$\square$

**Lemma A.3.** $\partial_{y_i} K(\cdot, y)$ *and* $\partial^2_{y_i y_j} K(\cdot, y)$ *are in $\mathcal{H}$ for any fixed $y \in \bar{\Omega}$ and $i, j = 1, \ldots, d$.*

*Proof.* Let $v_i = (0, \ldots, 0, 1, 0, \ldots, 0) \in \mathbb{R}^d$ with the $i$-th component to be $1$ and the rest to be $0$. We first show that the difference quotient

$$\Delta_i(h) = \frac{K(\cdot, y + hv_i) - K(\cdot, y)}{h} \tag{32}$$

is a Cauchy sequence in $\mathcal{H}$ as $h \searrow 0$.

Recall that by the definition of RKHS, for any $y_1, y_2 \in \bar{\Omega}$,

$$\langle K(\cdot, y_1), K(\cdot, y_2) \rangle = K(y_1, y_2), \tag{33}$$

hence for any $h_1, h_2 > 0$, we have

$$\begin{aligned}
&\langle \Delta_i(h_1), \Delta_i(h_2) \rangle \\
=& \frac{1}{h_1 h_2} (K(y_1 + h_1 v_i, y_2 + h_2 v_i) - K(y_1 + h_1 v_i, y_2) - K(y_1, y_2 + h_2 v_i) + K(y_1, y_2)) \\
\to& \partial_{x_i} \partial_{y_i} K(y_1, y_2)
\end{aligned} \tag{34}$$

as $h_1, h_2 \searrow 0$. Thus,

$$\|\Delta_i(h_1) - \Delta_i(h_2)\|^2_{\mathcal{H}} = \langle \Delta_i(h_1), \Delta_i(h_1) \rangle - 2\langle \Delta_i(h_1), \Delta_i(h_2) \rangle + \langle \Delta_i(h_2), \Delta_i(h_2) \rangle \to 0 \tag{35}$$

as $h_1, h_2 \searrow 0$. This implies that $\Delta_i(h)$ is a Cauchy sequence in $\mathcal{H}$, hence has a limit $\Delta_i \in \mathcal{H}$. $\Delta_i$ is also the $L^2$ limit of $\Delta_i(h)$ since

$$\|\Delta_i(h) - \Delta_i\|_{L^2} \leq \|\Delta_i(h) - \Delta_i\|_{\mathcal{H}}. \tag{36}$$

However, $\Delta_i(h)$ converges to $\partial_{y_i} K(\cdot, y)$ pointwisely. Thus, $\partial_{y_i} K(\cdot, y) = \Delta_i \in \mathcal{H}$.

The proof for $\partial^2_{y_i y_j} K(\cdot, y)$ is similar. $\square$

**Lemma A.4.** *For any $y \in \bar{\Omega}$, the sequences*

$$\sqrt{\lambda_i} \partial_{y_j} \phi_i(y) \quad \text{and} \quad \sqrt{\lambda_i} \partial^2_{y_j y_k} \phi_i(y) \tag{37}$$

*are in $l^2$.*

*Moreover, let $v_i = (0, \ldots, 0, 1, 0, \ldots, 0) \in \mathbb{R}^d$ with the $i$-th component to be $1$ and the rest to be $0$.*
*Then*

$$\frac{\sqrt{\lambda_i}\phi_i(y + h_j) - \sqrt{\lambda_i}\phi_i(y)}{h} \to \sqrt{\lambda_i}\partial_{y_j}\phi_i(y) \quad in \; l^2 \tag{38}$$

*and*

$$\frac{\sqrt{\lambda_i}\partial_{y_j}\phi_i(y + h_k) - \sqrt{\lambda_i}\partial_{y_j}\phi_i(y)}{h} \to \sqrt{\lambda_i}\partial^2_{y_j y_k}\phi_i(y) \quad in \; l^2. \tag{39}$$

*Proof.* Note that

$$\Delta_j(h) = \frac{K(x, y + hv_j) - K(x, y)}{h} = \sum_{i=0}^{\infty} \frac{\sqrt{\lambda_i}\phi_i(y + h_j) - \sqrt{\lambda_i}\phi_i(y)}{h}\sqrt{\lambda_i}\phi_i(x). \tag{40}$$

By Lemma A.3, $\Delta_j(h)$ converges in $\mathcal{H}$. In other words, the sequence

$$\frac{\sqrt{\lambda_i}\phi_i(y + h_j) - \sqrt{\lambda_i}\phi_i(y)}{h} \tag{41}$$

converges in $l^2$. Since $\sqrt{\lambda_i}\partial_{y_j}$ is the termwise limit of the above sequence, then

$$\frac{\sqrt{\lambda_i}\phi_i(y + h_j) - \sqrt{\lambda_i}\phi_i(y)}{h} \to \sqrt{\lambda_i}\partial_{y_j}\phi_i(y) \quad in \; l^2. \tag{42}$$

Likewise we have

$$\frac{\sqrt{\lambda_i}\partial_{y_j}\phi_i(y + h_k) - \sqrt{\lambda_i}\partial_{y_j}\phi_i(y)}{h} \to \sqrt{\lambda_i}\partial^2_{y_j y_k}\phi_i(y) \quad in \; l^2. \tag{43}$$

$\square$

As direct consequences of the above lemma, we have the following differentiation properties of kernel function $K$ and functions in the RKHS of $K$:

**Corollary A.5.** *For any multi-index $\alpha, \beta$ with $|\alpha|, |\beta| \le 2$,*

$$\partial_x^\alpha \partial_y^\beta K(x, y) = \sum_{i=0}^{\infty} \lambda_i \partial_x^\alpha \phi_i(x) \partial_y^\beta \phi_i(y), \tag{44}$$

*converges absolutely and uniformly.*

**Corollary A.6.** *For any $u \in \mathcal{H}$, $u$ is twice continuously differentiable. Moreover, if the RKHS expansion of $u$ is*

$$u(x) = \sum_{i=0}^{\infty} u_i \sqrt{\lambda_i}\phi_i(x), \quad \{u_i\} \in l^2, \tag{45}$$

*then for any multi-index $\alpha$ with $|\alpha| \le 2$,*

$$\partial_x^\alpha u(x) = \sum_{i=0}^{\infty} u_i \sqrt{\lambda_i}\partial_x^\alpha \phi_i(x) \tag{46}$$

*converges absolutely and uniformly.*

**Corollary A.7.** *For any $u \in \mathcal{H}$,*

$$\|u\|_{C^2(\bar{\Omega})} \le C\|u\|_{\mathcal{H}}. \tag{47}$$

*Proof.* Suppose that

$$u(x) = \sum_{i=0}^{\infty} u_i \sqrt{\lambda_i}\phi_i(x), \quad \{u_i\} \in l^2. \tag{48}$$

By direct computations, for any $x \in \bar{\Omega}$, we have

$$|u(x)| \le \sum_{i=0}^{\infty} \left| u_i \sqrt{\lambda_i}\phi_i(x) \right| \le \|u_i\|_{l^2} \cdot \|\sqrt{\lambda_i}\phi_i(x)\|_{l^2} = \|u\|_{\mathcal{H}} \cdot \sqrt{K(x, x)}, \tag{49}$$

$$|\partial_{x_j} u(x)| \le \sum_{i=0}^{\infty} \left| u_i \sqrt{\lambda_i} \partial_{x_j} \phi_i(x) \right| \le \|u_i\|_{l^2} \cdot \|\sqrt{\lambda_i} \partial_{x_j} \phi_i(x)\|_{l^2} = \|u\|_{\mathcal{H}} \cdot \partial_{x_j} \sqrt{\partial_{y_j} K(x,x)}, \quad (50)$$

and

$$|\partial^2_{x_j x_k} u(x)| \le \sum_{i=0}^{\infty} \left| u_i \sqrt{\lambda_i} \partial^2_{x_j x_k} \phi_i(x) \right|$$
$$\le \|u_i\|_{l^2} \cdot \|\sqrt{\lambda_i} \partial^2_{x_j x_k} \phi_i(x)\|_{l^2} = \|u\|_{\mathcal{H}} \cdot \partial^2_{x_j x_k} \sqrt{\partial^2_{y_j y_k} K(x,x)}. \quad (51)$$

$\square$

The above results complete the proof of Theorem A.1.

## B  Proof of Lemma 4.1

In this section, we analyze the difference between the empirical gradient flow (13)

$$\frac{d}{dt} \hat{u}_t(x) = -\frac{2}{N_u} \sum_{i=1}^{N_u} K(x, X_i) \cdot \hat{u}_t(X_i) - \frac{2}{N_f} \sum_{j=1}^{N_f} \mathcal{L}_y K(x, Y_j) \cdot (\mathcal{L}\hat{u}_t(Y_j) - f(Y_j) - \varepsilon_j) \quad (52)$$

and the population gradient flow (21)

$$\frac{d}{dt} v_t(x) = -2 \int_{\partial\Omega} K(x, \xi) \cdot v_t(\xi) d\xi - 2 \int_{\Omega} \mathcal{L}_y K(x, y) \cdot (\mathcal{L}v_t(y) - f(y)) dy. \quad (53)$$

**Lemma B.1.** *As $N_f \to \infty$, we have*

$$\sup_{x \in \bar{\Omega}, t \in [0,T]} \left| \frac{2}{N_f} \sum_{j=1}^{N_f} \mathcal{L}_y K(x, Y_j) \cdot (\mathcal{L}\hat{u}_t(Y_j) - f(Y_j) - \varepsilon_j) - 2 \int_{\Omega} \mathcal{L}_y K(x, y) \cdot (\mathcal{L}\hat{u}_t(y) - f(y)) dy \right|$$

$$= \mathcal{O}\left( \sqrt{T \frac{\log N_f}{N_f}} \right)$$

(54)

*with high probability $1 - O(N_f^{-1})$, where the conventions $\mathcal{O}$ hide the terms involving $\Omega$, $K$, $f$ and $T$.*

*Proof.* Define

$$U_i(t) = \int_0^t \hat{u}_s(X_i)(s) ds, \quad F_j(t) = \int_0^t (\mathcal{L}\hat{u}_s(Y_j) - f(Y_j) - \varepsilon_j)(s) ds, \quad (55)$$

then

$$\mathcal{L}\hat{u}_t(x) = \int_0^t \frac{d}{ds} \mathcal{L}\hat{u}_s(x) ds = -\frac{2}{N_u} \sum_{i=1}^{N_u} \mathcal{L}_x K(x, X_i) U_i(t) - \frac{2}{N_f} \sum_{j=1}^{N_f} \mathcal{L}_x \mathcal{L}_y K(x, Y_j) F_j(t). \quad (56)$$

We make the following decomposition:

$$-\frac{2}{N_f} \sum_{j=1}^{N_f} \mathcal{L}_y K(x, Y_j) \cdot (\mathcal{L}\hat{u}_t(Y_j) - f(Y_j) - \varepsilon_j) = T_1 + T_2 + T_3 + T_4 + T_5, \quad (57)$$

where

$$T_1 = -\frac{2}{N_u N_f} \sum_{i=1}^{N_u} \sum_{j=1}^{N_f} \mathcal{L}_x K(Y_j, X_i) \mathcal{L}_y K(x, Y_j) U_i, \quad (58)$$

$$T_2 = -\frac{2}{N_f^2} \sum_{j \ne k} \mathcal{L}_y K(x, Y_j) \mathcal{L}_x \mathcal{L}_y K(Y_j, Y_k) F_k, \quad (59)$$

$$T_3 = -\frac{2}{N_f^2} \sum_{j=1}^{N_f} \mathcal{L}_y K(x, Y_j) \mathcal{L}_x \mathcal{L}_y K(Y_j, Y_j) F_j, \tag{60}$$

$$T_4 = -\frac{2}{N_f} \sum_{j=1}^{N_f} \mathcal{L}_y K(x, Y_j) f(Y_j). \tag{61}$$

$$T_5 = -\frac{2}{N_f} \sum_{j=1}^{N_f} \mathcal{L}_y K(x, Y_j) \varepsilon_j. \tag{62}$$

For brevity, we only prove the estimation for $T_4$. The proofs for $T_1$, $T_2$, $T_3$ and $T_5$ are similar. It is easy to see that by strong law of large number, for any fixed $x \in \bar{\Omega}$,

$$T_4 \to -2 \int_\Omega \mathcal{L}_y K(x, y) f(y) dy \tag{63}$$

almost surely as $N_f$ goes to infinity. Next, we show that the convergence is actually uniform with respect to $x \in \bar{\Omega}$.

By assumption, $f \in L^2(\Omega)$ is bounded. And by **Theorem 5.4**, $\mathcal{L}_y K(x, y)$ is Lipschitz for $(x, y) \in \bar{\Omega} \times \bar{\Omega}$. Since $\bar{\Omega}$ is a bounded region, the random variables $\mathcal{L}_y K(x, Y_j) f(Y_j)$ are sub-exponential obviously. Then, by Bernstein's inequality (see Section 2.8 of [39]), we have the following estimation:

$$\mathbb{P}\left( \left| -\frac{2}{N_f} \sum_{j=1}^{N_f} \mathcal{L}_y K(x, Y_j) f(Y_j) + 2 \int_\Omega \mathcal{L}_y K(x, y) f(y) dy \right| > \eta \right) \le 2 e^{-c \min\{N_f \eta^2, N_f \eta\}}, \tag{64}$$

where the constant $c > 0$ depends only on $\Omega$, $k$ and $f$.

Next, we apply the $\varepsilon$-net argument to $x \in \bar{\Omega}$. Without loss of generality, assume that $\Omega$ is contained in the cube $[0, D]^d$. For any $n \in \mathbb{N}$, we define

$$\mathcal{N}_\varepsilon = \{(\varepsilon z_1, \ldots, \varepsilon z_d) : z_i = 0, \ldots, n+1, i = 1, \ldots, d\} \subset [0, D]^d, \tag{65}$$

then the canonical number of $\mathcal{N}_\varepsilon$ is $(n+1)^d$, and

$$\mathbb{P}\left( \max_{x \in \mathcal{N}_\varepsilon} \left| -\frac{2}{N_f} \sum_{j=1}^{N_f} \mathcal{L}_y K(x, Y_j) f(Y_j) + 2 \int_\Omega \mathcal{L}_y K(x, y) f(y) dy \right| \le \eta \right)$$
$$> 1 - 2(n+1)^d e^{-c \min\{N_f \eta^2, N_f \eta\}}. \tag{66}$$

Moreover, for any $x \in \bar{\Omega}$, there exists $x' \in \mathbb{N}_\varepsilon$ such that $|x - x'| < \varepsilon^d$.

For any $x, x' \in \bar{\Omega}$ such that $|x - x'| < \varepsilon^d$, by applying Bernstein's inequality again, we obtain that

$$\mathbb{P}\left( \sup_{|x-x'|<\varepsilon^d} \left| \frac{2}{N_f} \sum_{j=1}^{N_f} \mathcal{L}_y K(x, Y_j) f(Y_j) - \frac{2}{N_f} \sum_{j=1}^{N_f} \mathcal{L}_y K(x', Y_j) f(Y_j) \right| \le \eta \right)$$
$$\ge \mathbb{P}\left( \left| \frac{2c' \varepsilon^d}{N_f} \sum_{j=1}^{N_f} f(Y_j) \right| \le \eta \right)$$
$$\ge \mathbb{P}\left( \left| \frac{2}{N_f} \sum_{j=1}^{N_f} f(Y_j) - 2 \int_\Omega f \right| \le \frac{\eta}{2c' \varepsilon^d}, \quad \left| 2 \int_\Omega f \right| \le \frac{\eta}{2c' \varepsilon^d} \right) \tag{67}$$
$$= \mathbb{P}\left( \left| \frac{2}{N_f} \sum_{j=1}^{N_f} f(Y_j) - 2 \int_\Omega f \right| \le \frac{\eta}{2c' \varepsilon^d} \right)$$
$$\ge 1 - 2 e^{-c'' \min\{N_f \eta^2 / \varepsilon^{2d}, N_f \eta / \varepsilon^d\}}$$

if we select $\eta$, $\varepsilon$ such that $|2 \int_\Omega f| \le \frac{\eta}{2c' \varepsilon^d}$, and the constant $c'$, $c'' > 0$ depends only on $\Omega$, $K$ and $f$.

We also have
$$\left| 2\int_\Omega \mathcal{L}_y K(x,y)f(y)dy - 2\int_\Omega \mathcal{L}_y K(x',y)f(y)dy \right| \le c'''\varepsilon^d \tag{68}$$
where $c''' > 0$ depends only on $\Omega$, $K$ and $f$.

Combining the estimations above together yields
$$\mathbb{P}\left( \sup_{x\in\bar\Omega}\left| -\frac{2}{N_f}\sum_{j=1}^{N_f}\mathcal{L}_y K(x,Y_j)f(Y_j) + 2\int_\Omega \mathcal{L}_y K(x,y)f(y)dy \right| \le (2+c''')\eta \right)$$
$$\ge 1 - \mathbb{P}\left( \max_{x\in\mathcal{N}_\varepsilon}\left| -\frac{2}{N_f}\sum_{j=1}^{N_f}\mathcal{L}_y K(x,Y_j)f(Y_j) + 2\int_\Omega \mathcal{L}_y K(x,y)f(y)dy \right| > \eta \right) \tag{69}$$
$$- \mathbb{P}\left( \sup_{|x-x'|<\varepsilon^d}\left| \frac{2}{N_f}\sum_{j=1}^{N_f}\mathcal{L}_y K(x,Y_j)f(Y_j) - \frac{2}{N_f}\sum_{j=1}^{N_f}\mathcal{L}_y K(x',Y_j)f(Y_j) \right| > \eta \right)$$
$$\ge 1 - 2(D/\varepsilon+1)^d e^{-c\min\{N_f\eta^2,N_f\eta\}} - 2e^{-c''\min\{N_f\eta^2/\varepsilon^{2d},N_f\eta/\varepsilon^d\}}.$$

Finally, we select $\varepsilon^{2d}\asymp \frac{1}{N_f}$ and $\eta^2\asymp \frac{\log N_f}{N_f}$, then with high probability $1-\mathcal{O}(N_f^{-1})$, we have
$$\sup_{x\in\bar\Omega}\left| -\frac{2}{N_f}\sum_{j=1}^{N_f}\mathcal{L}_y K(x,Y_j)f(Y_j) + 2\int_\Omega \mathcal{L}_y K(x,y)f(y)dy \right| = \mathcal{O}\left( \sqrt{\frac{\log N_f}{N_f}} \right). \tag{70}$$

Likewise, we also have the following estimation for $T_1$, $T_2$, $T_3$ and $T_5$:
$$\sup_{x\in\bar\Omega}\left| -\frac{2}{N_f}\sum_{j=1}^{N_f}\mathcal{L}_x K(Y_j,X_i)\mathcal{L}_y K(x,Y_j) + 2\int_\Omega \mathcal{L}_x K(y,X_i)\mathcal{L}_y K(x,y) \right|$$
$$=\mathcal{O}\left( \sqrt{\frac{\log N_f}{N_f}} \right), \tag{71}$$

$$\sup_{x\in\bar\Omega}\left| -\frac{2}{N_f}\sum_{1\le j\le N_f, j\ne k}\mathcal{L}_y K(x,Y_j)\mathcal{L}_x\mathcal{L}_y K(Y_j,Y_k) - \int_\Omega \mathcal{L}_y K(x,y)\mathcal{L}_y K(y,Y_k)dy \right|$$
$$= \mathcal{O}\left( \sqrt{\frac{\log N_f}{N_f}} \right), \quad k=1,2,\ldots,N_f, \tag{72}$$

$$\sup_{x\in\bar\Omega}\left| -\frac{2}{N_f^2}\sum_{j=1}^{N_f} L_y K(x,Y_j)\mathcal{L}_x\mathcal{L}_y K(Y_j,Y_j) \right| = \mathcal{O}\left( \frac{1}{N_f}\sqrt{\frac{\log N_f}{N_f}} \right), \tag{73}$$

$$\sup_{x\in\bar\Omega}\left| -\frac{2}{N_f}\sum_{j=1}^{N_f}\mathcal{L}_y K(x,Y_j)\varepsilon_j \right| = \mathcal{O}\left( \sqrt{\frac{\log N_f}{N_f}} \right) \tag{74}$$

with probability $1-\mathcal{O}(N_f^{-1})$. We also note that
$$\int_\Omega \mathcal{L}_y K(x,y)\cdot(\mathcal{L}\hat u_t(y) - f(y))dy$$
$$= \int_\Omega \mathcal{L}_y K(x,y)\left( -\frac{2}{N_u}\sum_{i=1}^{N_u}\mathcal{L}_x K(y,X_i)U_i - \frac{2}{N_f}\sum_{j=1}^{N_f}\mathcal{L}_x\mathcal{L}_y K(y,Y_j)F_j - f(y) \right)dy, \tag{75}$$

hence

$$-\frac{2}{N_f}\sum_{j=1}^{N_f}\mathcal{L}_y K(x,Y_j)\cdot(\mathcal{L}\hat{u}_t(Y_j)-f(Y_j)-\varepsilon_j)+2\int_\Omega \mathcal{L}_y K(x,y)\cdot(\mathcal{L}\hat{u}_t(y)-f(y))dy$$

$$=\int_\Omega\left(\frac{1}{N_u}\sum_{i=1}^{N_u}U_i\cdot\mathcal{O}\left(\sqrt{\frac{\log N_f}{N_f}}\right)+\frac{1}{N_f}\sum_{j=1}^{N_f}F_j\cdot\mathcal{O}\left(\sqrt{\frac{\log N_f}{N_f}}\right)+\mathcal{O}\left(\sqrt{\frac{\log N_f}{N_f}}\right)\right)dy.$$

(76)

By Cauchy's inequality, and by the definition of $U_i$ and $F_j$, we have

$$\left|-\frac{2}{N_f}\sum_{j=1}^{N_f}\mathcal{L}_y K(x,Y_j)\cdot(\mathcal{L}\hat{u}_t(Y_j)-f(Y_j)-\varepsilon_j)+2\int_\Omega \mathcal{L}_y K(x,y)\cdot(\mathcal{L}\hat{u}_t(y)-f(y))dy\right|^2$$

$$\leq\int_\Omega\int_0^t\mathcal{O}\left(\sqrt{\frac{\log N_f}{N_f}}\right)^2\left(\frac{1}{N_u}\sum_{i=1}^{N_u}\hat{u}_s(X_i)^2+\frac{1}{N_f}\sum_{j=1}^{N_f}(\mathcal{L}\hat{u}_s(Y_j)-f(Y_j)-\varepsilon_j)^2\right)dsdy$$

$$+\mathcal{O}\left(\sqrt{\frac{\log N_f}{N_f}}\right)^2$$

$$=\mathcal{O}\left(\sqrt{\frac{\log N_f}{N_f}}\right)^2\int_0^t\hat{L}(\hat{u}_s)ds+\mathcal{O}\left(\sqrt{\frac{\log N_f}{N_f}}\right)^2.$$

(77)

Since $\hat{L}(u)$ decreases along the gradient flow, then $\hat{L}(u(t))\leq\hat{L}(u(0))$ for any $t\in[0,T]$, hence

$$\int_0^t\hat{L}(\hat{u}_s)ds\leq T\hat{L}(\hat{u}_0).$$

(78)

In conclusion, with probability $1-\mathcal{O}(N_f^{-1})$, we have

$$\sup_{x\in\bar{\Omega},t\in[0,T]}\left|-\frac{2}{N_f}\sum_{j=1}^{N_f}\mathcal{L}_y K(x,Y_j)\cdot(\mathcal{L}\hat{u}_t(Y_j)-f(Y_j)-\varepsilon_j)+2\int_\Omega \mathcal{L}_y K(x,y)\cdot(\mathcal{L}\hat{u}_t(y)-f(y))dy\right|$$

$$=\mathcal{O}\left(\sqrt{T\frac{\log N_f}{N_f}}\right).$$

(79)

$\square$

Likewise, we also have

**Lemma B.2.** *As $N_u\to\infty$, we have*

$$\sup_{x\in\bar{\Omega},t\in[0,T]}\left|-\frac{2}{N_u}\sum_{i=1}^{N_u}K(x,X_i)\cdot\hat{u}_t(X_i)+2\int_{\partial\Omega}K(x,\xi)\cdot\hat{u}_t(\xi)d\xi\right|=\mathcal{O}\left(\sqrt{T\frac{\log N_u}{N_u}}\right)\quad(80)$$

*with high probability $1-\mathcal{O}(N_u^{-1})$.*

*Proof.* The proof is similar with that of **Lemma B.1**. $\square$

**Lemma B.3.** *With high probability $1-\mathcal{O}(N_u^{-1}+N_f^{-1})$, we have*

$$d_{pinn}(\hat{u}_t-v_t)=\mathcal{O}\left(T^{\frac{3}{2}}(\sqrt{\frac{\log N_f}{N_f}}+\sqrt{\frac{\log N_u}{N_u}})\right)\quad(81)$$

*for any $t\in[0,T]$.*

*Proof.* Take $w_t = \hat{u}_t - v_t$. Then by **Lemma B.1** and **Lemma B.2**, we have

$$\frac{d}{dt}w_t(x) = -2\int_{\partial\Omega} K(x,\xi)\cdot w_t(\xi)d\xi - 2\int_{\Omega} \mathcal{L}_y K(x,\eta)\cdot \mathcal{L}w_t(\eta)d\eta + \mathcal{R}, \tag{82}$$

where

$$\mathcal{R} = \mathcal{O}\left(\sqrt{T\frac{\log N_u}{N_u}}\right) + \mathcal{O}\left(\sqrt{T\frac{\log N_f}{N_f}}\right) \tag{83}$$

with probability $1 - \mathcal{O}(N_u^{-1}) - \mathcal{O}(N_f^{-1})$.

Define $W(t) = d_{pinn}(\hat{u}_t, v_t)$, then

$$W(t)^2 = d_{pinn}(\hat{u}_t, v_t)^2 = \int_{\partial\Omega} |w_t(x)|^2 dx + \int_{\Omega} |\mathcal{L}w_t(y)|^2 dy, \tag{84}$$

and

$$\frac{d}{dt}(W(t)^2) = -2\mathcal{Q} + \mathcal{R}\left(\int_{\partial\Omega} w_t(x)dx + \int_{\Omega} \mathcal{L}w_t(y)dy\right), \tag{85}$$

where

$$\mathcal{Q} = \int_{\partial\Omega}\int_{\partial\Omega} K(x,\xi)w_t(x)w_t(\xi)dxd\xi + 2\int_{\Omega}\int_{\partial\Omega} \mathcal{L}_y K(x,\eta)\mathcal{L}w_t(\eta)w_t(x)dxd\eta$$
$$+ \int_{\Omega}\int_{\Omega} \mathcal{L}_x\mathcal{L}_y K(y,\eta)\mathcal{L}w_t(y)\mathcal{L}w_t(\eta)dyd\eta. \tag{86}$$

Assume that the RKHS expansion of $w_t$ is

$$w_t(x) = \sum_{i=0}^{\infty} w_t^i \sqrt{\lambda_i}\phi_i(x), \tag{87}$$

and recall that

$$K(x,y) = \sum_{i=0}^{\infty} \lambda_i\phi_i(x)\phi_i(y), \tag{88}$$

then by computation, we obtain that

$$\mathcal{Q} = \sum_{i=0}^{\infty}\left(\int_{\partial\Omega} w_i^t\lambda_i\phi_i(x)dx + \int_{\Omega} w_i^t\lambda_i\mathcal{L}\phi_i(y)dy\right)^2 \geq 0. \tag{89}$$

Thus, by Cauchy inequality,

$$\frac{d}{dt}(W(t)^2) \leq \mathcal{R}\left(\int_{\partial\Omega} w_t(x)dx + \int_{\Omega} \mathcal{L}w_t(y)dy\right) \leq 2\mathcal{R}W(t), \tag{90}$$

then

$$\frac{d}{dt}W(t) \leq \mathcal{R}. \tag{91}$$

Therefore, for any $t \in [0, T]$, we have

$$W(t) = W(t) - W(0) = \int_0^t \frac{d}{ds}W(s)ds \leq \mathcal{O}\left(T^{\frac{3}{2}}(\sqrt{\frac{\log N_f}{N_f}} + \sqrt{\frac{\log N_u}{N_u}})\right). \tag{92}$$

$\square$

*Proof.* **(of Lemma 4.1)**. Note that

$$\begin{aligned}|L(\hat{u}_t) - L(v_t)| &= |d_{pinn}(\hat{u}_t, u^*)^2 - d_{pinn}(v_t, u^*)^2| \\ &= |(d_{pinn}(\hat{u}_t, u^*) - d_{pinn}(v_t, v^*))^2 \\ &\quad + 2d_{pinn}(v_t, u^*)(d_{pinn}(\hat{u}_t, u^*) - d_{pinn}(v_t, u^*)| \\ &\leq d_{pinn}(\hat{u}_t, v_t)^2 + 2d_{pinn}(v_t, u^*)d_{pinn}(\hat{u}_t, v_t).\end{aligned} \tag{93}$$

Since $L(v_t) = d_{pinn}(v_t, u^*)$ is decreasing, then

$$d_{pinn}(v_t, u^*) \leq d_{pinn}(v_0, u^*) = d_{pinn}(0, u^*) = \sqrt{\int_\Omega |f|^2}. \tag{94}$$

Thus, by **Lemma B.3**, with great probability $1 - \mathcal{O}(N_u^{-1} + N_f^{-1})$, we have

$$|L(\hat{u}_t) - L(v_t)| = \mathcal{O}\left(T^{\frac{3}{2}}(\sqrt{\frac{\log N_f}{N_f}} + \sqrt{\frac{\log N_u}{N_u}})\right) \tag{95}$$

for all $t \in [0, T]$. $\qquad\square$

## C   Proof of Lemma 4.2

In this section, we consider the population gradient flow (21):

$$\frac{d}{dt} v_t(x) = -2 \int_{\partial\Omega} K(x, \xi) \cdot v_t(\xi) d\xi - 2 \int_\Omega \mathcal{L}_y K(x, y) \cdot (\mathcal{L} v_t(y) - f(y)) dy. \tag{96}$$

### C.1   Diagonalized representation of population gradient flow

Assume that the expansion of $v$ in RKHS is

$$v_t = \sum_{i=0}^\infty v_i(t) \sqrt{\lambda_i} \phi_i, \tag{97}$$

then along the gradient flow (21), the evolution ODE of the coefficients $v_i = v_i(t)$ is

$$
\begin{aligned}
\frac{d}{dt} v_i(t) &= -2 \int_{\partial\Omega} \sqrt{\lambda_i} \phi_i \sum_{j=0}^\infty v_j(t) \sqrt{\lambda_j} \phi_j - 2 \int_\Omega \sqrt{\lambda_i} \mathcal{L}\phi_i \left( \sum_{j=0}^\infty v_j(t) \sqrt{\lambda_j} \mathcal{L}\phi_j - f \right) \\
&= -2 \sum_{j=0}^\infty (S_{ij} + T_{ij}) v_j(t) + R_i,
\end{aligned}
\tag{98}
$$

where

$$S_{ij} = \int_{\partial\Omega} \sqrt{\lambda_i \lambda_j} \phi_i \phi_j, \quad T_{ij} = \int_\Omega \sqrt{\lambda_i \lambda_j} \mathcal{L}\phi_i \mathcal{L}\phi_j, \quad R_i = \int_\Omega \sqrt{\lambda_i} \mathcal{L}\phi_i f. \tag{99}$$

The infinite-dimensional symmetric matrices $S_{ij}$ and $T_{ij}$ are the Gram matrices of the following self-adjoint operators on $\mathcal{H}$:

$$S : \mathcal{H} \to \mathcal{H}, \quad Sv = \sum_{i=0}^\infty \left( \sum_{j=0}^\infty S_{ij} v_j \right) \sqrt{\lambda_i} \phi_i, \tag{100}$$

$$T : \mathcal{H} \to \mathcal{H}, \quad Tv = \sum_{i=0}^\infty \left( \sum_{j=0}^\infty T_{ij} v_j \right) \sqrt{\lambda_i} \phi_i. \tag{101}$$

The population gradient flow (21) can be rewritten as

$$\frac{d}{dt} v_t = -2(S + T)(v_t) + R, \tag{102}$$

where

$$R = \sum_{i=0}^\infty R_i \phi_i. \tag{103}$$

**Lemma C.1.** *S and T are compact.*

*Proof.* For any $v \in \mathcal{H}$, let $w = Sv$, then

$$|w_i|^2 = \left| \sum_{j=0}^{\infty} S_{ij} v_j \right|^2 \leq \sum_{j=0}^{\infty} v_j^2 \cdot \sum_{j=0}^{\infty} S_{ij}^2 = \|v\|_{\mathcal{H}}^2 \sum_{j=0}^{\infty} \left( \int_{\partial\Omega} \sqrt{\lambda_i \lambda_j} \phi_i \phi_j \right)^2$$

$$\leq \|v\|_{\mathcal{H}}^2 \sum_{j=0}^{\infty} \int_{\partial\Omega} \lambda_i \phi_i^2 \cdot \int_{\partial\Omega} \lambda_j \phi_j^2 = \|v\|_{\mathcal{H}}^2 \int_{\partial\Omega} \lambda_i \phi_i^2 \cdot \int_{\partial\Omega} \sum_{j=0}^{\infty} \lambda_j \phi_j^2 \qquad (104)$$

$$= \|v\|_{\mathcal{H}}^2 \int_{\partial\Omega} \lambda_i \phi_i^2 \cdot \int_{\partial\Omega} K(x,x) dx.$$

Thus, $S$ maps the infinite-dimensional ball $\{v \in \mathcal{H} : \|v\|_{\mathcal{H}} \leq B\}$ into the infinite-dimensional cube

$$Q_B = \{w \in \mathcal{H} : w_i \leq B a_i\}, \qquad (105)$$

where

$$a_i^2 = \int_{\partial\Omega} K(x,x) dx \cdot \int_{\partial\Omega} \lambda_i \phi_i^2. \qquad (106)$$

Note that

$$\sum_{i=0}^{\infty} a_i^2 = \left( \int_{\partial\Omega} K(x,x) dx \right)^2 < \infty, \qquad (107)$$

hence $\{a_i\} \in l^2$.

It suffices to show that any sequence $\{w^k\}_{k=0}^{\infty} \subset Q_B$ has a convergent sequence in $\mathcal{H}$.

Since $\{w_0^k\}$ is contained in $[-Ba_0, Ba_0]$, it has a subsequence $\{w_0^{k^0(j)}\}$ such that $w_0^{k^0(j)} \to w_0$ as $j \to \infty$. Then we select a subsequence of $\{w_1^{k^0(j)}\}$, denoted as $w_1^{k^1(j)}$, such that $w_1^{k^1(j)} \to w_1$ as $j \to \infty$. Iteratively, we select subsequence $k^l(j)$ from $k^{l-1}(j)$ such that $w_l^{k_l(j)} \to w_l$ as $j \to \infty$. Note that $|w_l| \leq Ba_l$, hence $\{w_l\} \in l^2$. Define

$$w = \sum_{l=0}^{\infty} w_l \sqrt{\lambda_l} \phi_l \in \mathcal{H}. \qquad (108)$$

Then we claim that

$$\|w^{k^l(l)} - w\|_{\mathcal{H}} \to 0 \quad \text{as } l \to \infty. \qquad (109)$$

In fact, for any $\varepsilon > 0$, we can decompose $\|w^{k^l(l)} - w\|\mathcal{H}$ into two parts:

$$\|w^{k^l(l)} - w\|_{\mathcal{H}}^2 = \sum_{i=0}^{N} (w_i^{k^l(l)} - w_i)^2 + \sum_{i=N+1}^{\infty} (w_i^{k^l(l)} - w_i)^2, \qquad (110)$$

where $N \in \mathbb{N}$ is selected such that

$$\sum_{l=N+1}^{\infty} a_i^2 < \varepsilon. \qquad (111)$$

Then we obtain that

$$\sum_{i=N+1}^{\infty} (w_i^{k^l(l)} - w_i)^2 \leq 2 \sum_{i=N+1}^{\infty} ((w_i^{k^l(l)})^2 + w_i^2) \leq 4B^2 \sum_{i=N+1}^{\infty} a_i^2 < 4B^2 \varepsilon. \qquad (112)$$

Thus,

$$\limsup_{l \to \infty} \|w^{k^l(l)} - w\|_{\mathcal{H}} \leq 4B^2 \varepsilon. \qquad (113)$$

Since $\varepsilon$ is arbitrary, we conclude that

$$\lim_{l \to \infty} \|w^{k^l(l)} - w\|_{\mathcal{H}} = 0. \qquad (114)$$

$\square$

Thus, $S+T$ is a self-adjoint compact operator on $\mathcal{H}$. By Riezs-Schauder theory, $S+T$ has eigenvalues $\mu_0 \geq \mu_1 \geq \cdots \geq 0$, $\mu_i \to 0$ as $i \to \infty$, and $\{\psi_i\}$ the corresponding eigenbasis.

Under the new orthonormal basis $\{\psi_i\}$, let $\tilde{v}_i$ be the coefficients of expansion of $v_t$:

$$v_t(x) = \sum_{i=0}^{\infty} \tilde{v}_i(t)\psi_i(x), \tag{115}$$

Then under the basis $\{\psi_i\}$, the gradient flow (102) can be rewritten as

$$\frac{d}{dt}\tilde{v}_i(t) = -2\mu_i\tilde{v}_i(t) + \tilde{R}_i, \tag{116}$$

where $\{\tilde{R}_i\} \in l^2$ is determined by

$$R = \sum_{i=0}^{\infty} R_i\phi_i(x) = \sum_{i=0}^{\infty} \tilde{R}_i\psi_i(x). \tag{117}$$

Thus, for each $i$, $\tilde{v}_i^2$ is increasing along the gradient flow (21).

## C.2  Convex optimization

**Lemma C.2.** $\|\nabla L(v_t)\|_{\mathcal{H}} \to 0$ as $t \to \infty$.

*Proof.* It suffices to show that $L(v_t)$ is a convex function with respect to $t \in [0, \infty)$. The convexity is directly checked by

$$
\begin{aligned}
\frac{d^2}{dt^2}L(v_t) &= -\frac{d}{dt}\sum_{i=0}^{\infty}\left(\frac{\partial L}{\partial v_i}(v_t)\right)^2 = -2\sum_{i=0}^{\infty}\frac{\partial L}{\partial v_i}(v_t) \cdot \frac{\partial^2 L}{\partial t \partial v_i}(v_t) \\
&= -2\sum_{i=0}^{\infty}\frac{\partial L}{\partial v_i}(v_t) \cdot \frac{\partial}{\partial v_i}\left(\frac{\partial L}{\partial t}(v_t)\right) \\
&= -2\sum_{i=0}^{\infty}\frac{\partial L}{\partial v_i}(v_t)\frac{\partial}{\partial v_i}\left(-\sum_{j=0}^{\infty}\left(\frac{\partial L}{\partial v_j}(v_t)\right)^2\right) \\
&= 4\sum_{i=0}^{\infty}\sum_{j=0}^{\infty}\frac{\partial L}{\partial v_i}(v_t) \cdot \frac{\partial^2 L}{\partial v_i \partial v_j}(v_t) \cdot \frac{\partial L}{\partial v_j}(v_t) \\
&\geq 0.
\end{aligned} \tag{118}
$$

$\square$

*Proof.* **(of Lemma 4.2).** Along the gradient flow, $L$ is non-increasing, hence $L$ converges to some constant $A \geq 0$. It suffices to prove that $A = \inf_{v \in \mathcal{H}} L(v)$.

Assume otherwise. Then by the convexity of $L$, for any $t > 0$, the set $\{v \in \mathcal{H} : L(v) < A\}$ must be contained in one side of the tangent hyperplane at $v(t)$:

$$\langle w, \nabla L(v_t)\rangle_{\mathcal{H}} < \langle v_t, \nabla L(v_t)\rangle_{\mathcal{H}}, \quad \forall w \in \{w \in \mathcal{H} : L(w) < A\}. \tag{119}$$

By assumption, $L(v_t) \geq A > \inf_{v \in \mathcal{H}} L(v)$ for any $t$. Then we can fix one element $w \in \{w \in \mathcal{H} : \inf_{v \in \mathcal{H}} L(v) < L(w) < A\}$, and denote its coefficient series with respect to the basis $\{\psi_i\}$ as $\{\tilde{w}_i\}$. Hence

$$\sum_{i=0}^{\infty} \tilde{w}_i \cdot \frac{\partial L}{\partial \tilde{v}_i}(v_t) < \sum_{i=0}^{\infty} \tilde{v}_i \cdot \frac{\partial L}{\partial \tilde{v}_i}(v_t), \tag{120}$$

which implies that

$$\frac{d}{dt}\|v_t - w\|_{\mathcal{H}}^2 = 2\sum_{i=0}^{\infty}(\tilde{v}_i(t) - \tilde{w}_i) \cdot \frac{d\tilde{v}_i}{dt} = -2\sum_{i=0}^{\infty}(\tilde{v}_i(t) - \tilde{w}_i)\frac{\partial L}{\partial \tilde{v}_i}(v_t) \leq 0. \tag{121}$$

Thus, along the gradient flow, the RKHS norm of $v(t)$ is bounded by

$$\|v_t\|_{\mathcal{H}}^2 \leq \|w\|_{\mathcal{H}}^2. \tag{122}$$

The equation (116) implies that $\tilde{v}_i(t)$ is monotonic. Thus, $\|v_t\|_{\mathcal{H}}^2$ increasingly converges to some $C \geq 0$. In other words,

$$\lim_{t \to \infty} \sum_{i=0}^{\infty} \tilde{v}_i(t)^2 = C. \tag{123}$$

Let $\tilde{V}_i = \lim_{t \to \infty} \tilde{v}_i(t)$. By monotone convergence theorem, we have

$$V = \sum_{i=0}^{\infty} \tilde{V}_i \psi_i \in \mathcal{H}, \quad \|V\|_{\mathcal{H}}^2 = \sum_{i=0}^{\infty} \tilde{V}_i^2 = \lim_{t \to \infty} \sum_{i=0}^{\infty} \tilde{v}_i(t)^2 = C < \infty, \tag{124}$$

hence

$$\|v_t - V\|_{\mathcal{H}} \to 0. \tag{125}$$

By **Lemma C.2** and the continuity of $\nabla L$, we obtain that

$$\|\nabla L(V)\|_{\mathcal{H}}^2 = \lim_{t \to \infty} \|\nabla L(v_t)\|_{\mathcal{H}}^2 = 0. \tag{126}$$

Then $V$ is in fact a minimizer of $L$ by convexity, hence $L(v_t) \to L(V) = 0$. This contradicts with the assumption that $L(v_t) \geq A > \inf_{v \in \mathcal{H}} L(v) \geq 0$! This completes our proof. $\qquad \square$

## D   Norm control

*Proof.* **(of Lemma 3.9)**. We decompose $u - u^*$ into $v + w$, where

$$\begin{cases} \mathcal{L}v = \mathcal{L}u - f & \text{in } \Omega; \\ v = 0 & \text{on } \partial\Omega, \end{cases} \qquad \begin{cases} \mathcal{L}w = 0 & \text{in } \Omega; \\ w = u & \text{on } \partial\Omega. \end{cases} \tag{127}$$

By the standard elliptic PDE theory, the operator $\mathcal{L}^{-1} : L^2(\Omega) \to H_0^1(\Omega)$ is bounded, hence

$$\|v\|_{L^2}^2 \leq C\|\mathcal{L}v\|_{L^2}^2 = C\|\mathcal{L}u - f\|_{L^2}, \tag{128}$$

where the constant $C > 0$ depends only on $\Omega$ and $\mathcal{L}$.

For the estimation of $w$, we first apply Green's representation formula to $w$ and obtain

$$w(y) = \int_{\partial\Omega} \mathcal{K}(x, y)u(x)dx, \quad \mathcal{K}(x, y) = \langle A\nabla_x G(x, y), \nu(x) \rangle \tag{129}$$

where $A = (a_{ij})$ is the coefficient matrix of $L$, $G$ is the Green's function of $\mathcal{L}$ on $\Omega$, and $\nu(x)$ is the outer unit normal vector of $\partial\Omega$ at $x$. It is well-known that (see Theorem 1.2.8 of [23] and its remarks for example)

$$\mathcal{K}(x, y) = \langle A\nabla_x G(x, y), \nu(x) \rangle \geq 0, \quad \forall x \in \partial\Omega, \ y \in \Omega \tag{130}$$

and

$$|\nabla_x G(x, y)| \leq C|x - y|^{1-d} \tag{131}$$

for some $C > 0$ depending only on $\Omega$ and $\mathcal{L}$. Therefore,

$$|\mathcal{K}(x, y)| \leq |A\nabla_x G(x, y)| \leq \|A\| \cdot |\nabla_x G(x, y)| \leq C|x - y|^{1-d}, \tag{132}$$

and

$$\int_{\Omega} w(y)^2 dy = \int_{\Omega} \left( \int_{\partial\Omega} \sqrt{\mathcal{K}(x, y)} \cdot \sqrt{\mathcal{K}(x, y)} u(x)dx \right)^2 dy$$
$$\leq \int_{\Omega} \left( \int_{\partial\Omega} \mathcal{K}(x, y)dx \cdot \int_{\partial\Omega} \mathcal{K}(x, y)u(x)^2 dx \right) dy. \tag{133}$$

Note that the function

$$\eta(y) = \int_{\partial\Omega} \mathcal{K}(x, y)dx \tag{134}$$

is the solution to the boundary value problem

$$\begin{cases} \mathcal{L}\eta = 0 & \text{in } \Omega; \\ \eta = 1 & \text{on } \partial\Omega, \end{cases} \tag{135}$$

then by the maximum principle, $|\eta| \leq 1$. Thus,

$$\begin{aligned}
\int_\Omega w(y)^2 dy &\leq \int_\Omega \int_{\partial\Omega} \mathcal{K}(x,y)u(x)^2 dx dy \\
&\leq \int_{\partial\Omega} \left( \int_\Omega C|x-y|^{1-d}dy \right) u(x)^2 dx \\
&\leq C' \int_{\partial\Omega} u(x)^2 dx,
\end{aligned} \tag{136}$$

where $C' > 0$ depends only on $\Omega$ and $\mathcal{L}$.

In conclusion, we have

$$\|u - u^*\|_{L^2(\Omega)} \leq \|v\|_{L^2(\Omega)} + \|w\|_{L^2(\Omega)} \leq C''(\|\mathcal{L}u - f\|_{L^2(\Omega)} + \|u\|_{L^2(\partial\Omega)}^2) = C''L(u) \tag{137}$$

for some $C'' > 0$ depending only on $\Omega$ and $\mathcal{L}$. □

## E    Proof of Theorem 3.12

**Lemma E.1.** *If $u^* \in \mathcal{H}$, then*

$$\|u - u^*\|_{\mathcal{H}} \cdot \|\nabla L(u)\|_{\mathcal{H}} \geq L(u). \tag{138}$$

*Proof.* When $u = u^*$, then $\|\nabla L(u^*)\|_{\mathcal{H}} = L(u) = 0$.

When $u \neq u^*$, define $R = \|u - u^*\|_{\mathcal{H}}$ and $w = (u - u^*)/R$. Let $\psi(r) = L(u^* + rw)$, then $\psi(0) = L(u^*) = 0$, $\psi(R) = L(u)$, and

$$\psi'(r) = \nabla_w L(u^* + rw) = \langle \nabla L(u^* + rw), w \rangle_{\mathcal{H}}. \tag{139}$$

Since $L$ is convex, then $\psi$ is also convex on $[0, R]$. Moreover, since $\psi(r)$ reaches its minimum at $r = 0$, then $\psi'(r)$ is increasing. Thus,

$$L(u) = \psi(R) - \psi(0) = \int_0^R \psi'(r)dr \leq R\psi'(R) = R\langle \nabla L(u), w \rangle_{\mathcal{H}} \leq R\|\nabla L(u)\|_{\mathcal{H}}. \tag{140}$$

□

**Lemma E.2.** *Along the gradient flow $v_t$ defined by (21), we have*

$$L(v_t) \leq \frac{C}{t}, \tag{141}$$

*for some constant $C > 0$ depending only on $\Omega$, $\mathcal{L}$, $f$ and $K$.*

*Proof.* Without loss of generality, assume that $u^* \neq 0$. Let $L(v_0) = L(0) = M$, then $M > 0$ depends on $\Omega$, $\mathcal{L}$ and $f$. Along the gradient flow (21), the loss $L(v_t)$ is decreasing. By Theorem A.1, $u^* \in C^2(\bar{\Omega})$. Then by Lemma A.7, we obtain that

$$\|v_t - u^*\|_{\mathcal{H}} \leq \|v_t - u^*\|_{C^2} \leq C'L(v_t) \leq C'L(v_0) = C'M \tag{142}$$

for some $C' > 0$ depending only on $\Omega$ and $K$.

Finally, by Lemma E.1, we have

$$\begin{aligned}
\frac{d}{dt}L(v_t) &= -\sum_{i=0}^\infty \left( \frac{\partial L}{\partial v_i}(v_t) \right)^2 = -\|\nabla L(v_t)\|_{\mathcal{H}}^2 \\
&\geq -\frac{1}{(C'M)^2}L(u)^2.
\end{aligned} \tag{143}$$

hence

$$\frac{1}{L(v_0)} - \frac{1}{L(v_t)} = \int_0^t \frac{1}{L(v_t)^2} \frac{d}{dt}L(v_t)dt \leq \frac{t}{(C'M)^2}. \tag{144}$$

□

*Proof.* **(of Theorem 3.12)** The conclusion is immediately obtained by combining Lemma 4.1 and Lemma E.2. □

# F   Proof of Theorem 5.4

In the rest of this paper, we will discuss the neural tangent kernel $K_{NT}$, its RKHS $\mathcal{H}_{NTK}$ and the network $\hat{u}_{\theta(t)}$. For brevity and with no ambiguity, in the following context, we denote the neural tangent kernel as $K$ instead of $K_{NT}$, and denote the RKHS of NTK as $\mathcal{H}$ instead of $\mathcal{H}_{NTK}$.

We will first prove Theorem 5.4 for shallow networks with only one hidden layer ($l = 1$) in Section F.1 and F.2, and then provide proofs for deep neural networks in Section F.3.

When $l = 1$, the network (5) has the following form:

$$\hat{u}_\theta(x) = \frac{1}{\sqrt{2m}} \sum_{k=1}^{2m} A_k \sigma(W_k \cdot x + B_k) + D, \tag{145}$$

where the parameters are initialized in the following way: $A_{k+m} = -A_k$, $W_{k+m} = W_k$, $B_{k+m} = B_k$, $D = 0$, and $A_k, W_{k,l}, B_k \sim N(0, 1)$, $k = 1, \ldots, m$, $l = 1, \ldots, d$ are initialized as i.i.d. standard Gaussian random variables.

## F.1   Convergence at initial time

### F.1.1   Pointwise convergence

Since $\Omega$ is bounded domain, we assume that $\bar{\Omega}$ is contained in the cube $Q(R) = [-R, R]^d$.

**Lemma F.1.** *For each $x, y \in Q(R) = [-R, R]^d$, $K_{\theta(0)}(x, y)$ converges to $K(x, y)$ almost surely.*

*Proof.* By direct computation, we have

$$\begin{aligned} K_{\theta(0)}(x, y) =& 1 + \frac{1}{m} \sum_{k=1}^{m} \sigma(W_k \cdot x + B_k) \cdot \sigma(W_k \cdot y + B_k) \\ & + \frac{1}{m} \sum_{k=1}^{m} A_k^2 (1 + \langle x, y \rangle) \sigma'(W_k \cdot x + B_k) \cdot \sigma'(W_k \cdot y + B_k) \end{aligned} \tag{146}$$

with the parameters initialized as i.i.d. standard Gaussian random variables. Thus, by strong law of large number (see Section 2.4 of [12] for example), it converges almost surely to

$$\begin{aligned} K(x, y) =& 1 + \mathrm{E}\left[\sigma(W_1 \cdot x + B_1) \cdot \sigma(W_1 \cdot y + B_1)\right] \\ & + (1 + \langle x, y \rangle) \mathrm{E}\left[A_1^2 \sigma'(W_1 \cdot x + B_1) \cdot \sigma'(W_1 \cdot y + B_1)\right]. \end{aligned} \tag{147}$$

□

**Lemma F.2.** *The limit kernel $K(x, y)$ is $C^2$ with respect to $x \in Q(R)$. Thus, $\mathcal{L}_x K(x, y)$ exists and is continuous for $(x, y) \in Q(R) \times Q(R)$.*

*Proof.* For brevity, we only prove that $\nabla_x K(x, y)$ exists and is continuous with respect to $x, y$. The proofs for the second derivatives $\nabla_x^2 K(x, y)$ are similar.

Note that $K(x, y) = 1 + G(x, y) + H(x, y)$, where

$$G(x, y) = \mathrm{E}\left[\sigma(W_1 \cdot x + B_1) \cdot \sigma(W_1 \cdot y + B_1)\right], \tag{148}$$

$$H(x, y) = (1 + \langle x, y \rangle) \mathrm{E}\left[A_1^2 \sigma'(W_1 \cdot x + B_1) \cdot \sigma'(W_1 \cdot y + B_1)\right]. \tag{149}$$

We first prove that the function $G(x, y)$ is $C^1$ with respect to $x$. Denote $v = (1, 0, \ldots, 0) \in \mathbb{R}^d$. By mean value theorem,

$$\frac{\sigma(W_1 \cdot (x + tv) + B_1) - \sigma(W_1 \cdot x + B_1)}{t} = W_{11} \sigma'(W_1 \cdot (x + sv) + B_1) \tag{150}$$

for some $s \in [0, t]$. Since $\sigma'$ is polynomially bounded, then for fixed $x, y$, the function

$$\left| \frac{\sigma(W_1 \cdot (x + tv) + B_1) - \sigma(W_1 \cdot x + B_1)}{t} \cdot \sigma(W_1 \cdot y + B_1) \right| \leq q(W_1, B_1) \tag{151}$$

for some polynomial $q$, which is integrable. By dominant convergence theorem, we have

$$\lim_{t \to 0} \frac{G(x + tv, y) - G(x, y)}{t}$$

$$= \mathrm{E} \left( \lim_{t \to 0} \frac{\sigma(W_1 \cdot (x + tv) + B_1) - \sigma(W_1 \cdot x + B_1)}{t} \cdot \sigma(W_1 \cdot y + B_1) \right) \tag{152}$$

$$= \mathrm{E}(\frac{\partial}{\partial x_1} \sigma(W_1 \cdot x + B_1) \cdot \sigma(W_1 \cdot y + B_1)),$$

which is continuous with respect to $x, y$. In other words, $G(x, y)$ has a continuous partial derivative in the direction $x_1$. Likewise, continuous partial derivatives in other directions also exist. Thus, $G(x, y)$ is differential with respect to $x$, and

$$\nabla_x G(x, y) = \mathrm{E}(\nabla_x \sigma(W_1 \cdot x + B_1) \cdot \sigma(W_1 \cdot y + B_1)) \tag{153}$$

is continuous with respect to $x, y$.

Likewise, $H(x, y)$ is continuously differentiable for $x$ as well. Thus, $K(x, y)$ is continuously differentiable for $x$. $\qquad \square$

**Lemma F.3.** *For each $x, y \in \bar{\Omega}$, $\nabla_x K_{\theta(0)}(x, y)$, $\nabla_x^2 K_{\theta(0)}(x, y)$ converge in probability to $\nabla_x k(x, y)$ and $\nabla_x^2 K(x, y)$, respectively. Thus, $\mathcal{L}_x K_{\theta(0)}(x, y)$ converges almost surely to $\mathcal{L}_x K(x, y)$.*

*Proof.* By strong law of large number,

$$\nabla_x K_{\theta(0)}(x, y) = 1 + \frac{1}{m} \sum_{k=1}^{m} \nabla_x \sigma(W_k \cdot x + B_k) \cdot \sigma(W_k \cdot y + B_k)$$

$$+ \frac{1}{m} \sum_{k=1}^{m} A_k^2 (1 + \langle x, y \rangle) \nabla_x \sigma'(W_k \cdot x + B_k) \cdot \sigma'(W_k \cdot y + B_k) \tag{154}$$

$$+ \frac{1}{m} \sum_{k=1}^{m} A_k^2 y \cdot \sigma'(W_k \cdot x + B_k) \cdot \sigma'(W_k \cdot y + B_k)$$

converges almost surely to

$$\nabla_x K(x, y) = 1 + \mathrm{E} \left[ \nabla_x \sigma(W_1 \cdot x + B_1) \cdot \sigma(W_1 \cdot y + B_1) \right]$$

$$+ (1 + \langle x, y \rangle) \mathrm{E} \left[ A_1^2 \nabla_x \sigma'(W_1 \cdot x + B_1) \cdot \sigma'(W_1 \cdot y + B_1) \right] \tag{155}$$

$$+ y \mathrm{E} \left[ A_1^2 \sigma'(W_1 \cdot x + B_1) \cdot \sigma'(W_1 \cdot y + B_1) \right].$$

Proofs for the second derivatives are similar. $\qquad \square$

So far, we have finished the prove for the existence of $K(x, y)$, $\mathcal{L}_x K(x, y)$ and the convergence of $K_{\theta(0)}(x, y)$, $\mathcal{L}_x K_{\theta(0)}(x, y)$ to $K(x, y)$, $\mathcal{L}_x K(x, y)$, respectively. For $\mathcal{L}_y K(x, y)$ and $\mathcal{L}_x \mathcal{L}_y K(x, y)$, by following a similar discussion, we can also obtain

**Lemma F.4.** *$\mathcal{L}_y K(x, y)$ and $\mathcal{L}_x \mathcal{L}_y K(x, y)$ both exist and are continuous with respect to $(x, y) \in Q(R) \times Q(R)$. And for fixed $x, y$, $\mathcal{L}_y K_{\theta(0)}(x, y)$ and $\mathcal{L}_x \mathcal{L}_y K_{\theta(0)}(x, y)$ converge almost surely to $\mathcal{L}_y K(x, y)$ and $\mathcal{L}_x \mathcal{L}_y K(x, y)$, respectively.*

### F.1.2 Uniform convergence at initial time

Recall that we assume $\bar{\Omega} \subset Q(R) = [-R, R]^d$. We will show that the convergence of $K_{\theta(0)}(x, y)$ and its derivatives is in fact uniform for $x, y \in Q(R)$. The proof is based on a $\varepsilon$-net argument.

For $N \in \mathbb{N}$, we set $\varepsilon = 2R/N$ and place $N^d$ points in the cube $Q(R) = [-R, R]^d$:

$$\mathcal{N}_\varepsilon = \left\{ \left( R + (-1 + \frac{2i_1}{N}), \dots, R + (-1 + \frac{2i_d}{N}) \right) : \quad i_k = 1, \dots, N, \, k = 1, \dots, d \right\} \tag{156}$$

For any $x \in Q(R)$, there exists some $z \in \mathcal{N}_\varepsilon$ such that $|x - z| \leq \varepsilon\sqrt{d}$.

Again, we denote $K_{\theta(0)}(x, y) = 1 + G_m(x, y) + H_m(x, y)$ and $K(x, y) = 1 + G(x, y) + H(x, y)$, where

$$G_m(x, y) = \frac{1}{m} \sum_{k=1}^{m} \sigma(W_k \cdot x + B_k) \cdot \sigma(W_k \cdot y + B_k), \tag{157}$$

$$G(x, y) = \mathrm{E}\left[\sigma(W_1 \cdot x + B_1) \cdot \sigma(W_1 \cdot y + B_1)\right], \tag{158}$$

$$H_m(x, y) = \frac{1}{m} \sum_{k=1}^{m} A_k^2(1 + \langle x, y \rangle)\sigma'(W_k \cdot x + B_k) \cdot \sigma'(W_k \cdot y + B_k), \tag{159}$$

$$H(x, y) = (1 + \langle x, y \rangle)\mathrm{E}\left[A_1^2\sigma'(W_1 \cdot x + B_1) \cdot \sigma'(W_1 \cdot y + B_1)\right]. \tag{160}$$

For any $x, y \in Q(R)$, we select $z, w \in \mathcal{N}_\varepsilon$ such that $|x - z| \leq \varepsilon\sqrt{d}, |y - w| \leq \varepsilon\sqrt{d}$. Using triangular inequality, we obtain that

$$|G_m(x, y) - G(x, y)| \leq |G_m(x, y) - G_m(z, w)| + |G(z, w) - G(x, y)| + |G_m(z, w) - G(z, w)|, \tag{161}$$

$$|H_m(x, y) - H(x, y)| \leq |H_m(x, y) - H_m(z, w)| + |H(z, w) - H(x, y)| + |H_m(z, w) - H(z, w)|. \tag{162}$$

The following lemmata provide the estimations for the terms on the right hand side.

**Lemma F.5.** *Define the event*

$$\mathcal{B} = \{|A_k|, |W_{k,l}|, |B_k| \leq M, \ k = 1, \ldots, m, \ l = 1, \ldots, d\}, \tag{163}$$

*where $M = \sqrt{3 \log m}$. Then $\mathbb{P}(\mathcal{B}) \geq 1 - (d + 2)m^{-1/2}$.*

*Proof.* If $Z \sim N(0, 1)$, then a classical Gaussian tail bound gives

$$\mathbb{P}(|Z| \geq M) \leq \frac{2e^{-M^2/2}}{\sqrt{2\pi}M} \leq m^{-3/2}, \tag{164}$$

hence

$$\mathbb{P}(\mathcal{B}) \geq 1 - m(d + 2)\mathbb{P}(|Z| \geq M) \geq 1 - (d + 2)m^{-1/2} \tag{165}$$

$\square$

**Lemma F.6.** *Conditioning on the event $\mathcal{B}$, we have*

$$|G_m(x, y) - G_m(z, w)| < C_1(\log m)^{p+\frac{1}{2}}\varepsilon, \tag{166}$$

$$|H_m(x, y) - H_m(z, w)| < C_1(\log m)^{p+\frac{3}{2}}\varepsilon, \tag{167}$$

*where the constant $C_1 > 0$ depends only on $d$, $R$ and $\sigma$.*

*Proof.* Recall that we assume $\bar{\Omega} \subset Q(R) = [-R, R]^d$. Conditioning on $\mathcal{B}$, for any $x \in Q(R)$, we have

$$|W_k \cdot x + B_k| \leq \sum_{l=1}^{d} |W_{k,l}| \cdot |x_l| + |B_k| \leq dRM + M. \tag{168}$$

By **Assumption 5.2**, $|\sigma(r)| \leq C'(1 + |r|^p)$, $|\sigma'(r)| \leq C'(1 + |r|^p)$ for some $C' > 0$. Then

$$|\sigma(W_k \cdot x + B_k)| \leq C'((1 + (dR + 1)M)^p), \tag{169}$$

hence

$$\|\nabla_x\sigma(W_k \cdot x + B_k)\| = \|W_k \cdot \sigma'(W_k \cdot x + B_k)\| \leq \|W_k\| \cdot |\sigma'(W_k \cdot x + B_k)|$$
$$\leq \sqrt{d}M \cdot C'(1 + ((dR + 1)M)^p) \leq C''M^{p+1}. \tag{170}$$

where $C''$ depends only on $\sigma$, $R$ and $d$. Thus, the Lipschitz constant of $G_m$ is bounded by

$$
\begin{aligned}
\sup_{x,y \in Q(R)} \|\nabla G_m(x,y)\| = \sup_{x,y} \|\nabla \frac{1}{m} \sum_{k=1}^{m} \sigma(W_k \cdot x + B_k)\sigma(W_k \cdot y + B_k)\| \\
\leq \sup_{x,y,k} \|\nabla(\sigma(W_k \cdot x + B_k)\sigma(W_k \cdot y + B_k))\| \\
\leq \sqrt{2} \sup_{x,y,k} \|\nabla_x(\sigma(W_k \cdot x + B_k)\sigma(W_k \cdot y + B_k))\| \\
\leq \sqrt{2} \sup_{x,k} \|\nabla_x \sigma(W_k \cdot x + B_k)\| \cdot \sup_{y,k} |\sigma(W_k \cdot y + B_k)| \\
\leq C'' M^{p+1} \cdot C'(1 + ((dR+1)M)^p) \\
\leq C''' M^{2p+1}
\end{aligned}
\tag{171}
$$

for some $C''' > 0$ depending only on $d$, $R$ and $\sigma$. This yields the conclusion.

The proof for $H_m$ is similar. $\qquad\square$

**Lemma F.7.** *We have*

$$|G(z,w) - G(x,y)| \leq C_2 \varepsilon, \tag{172}$$

$$|H(z,w) - H(x,y)| \leq C_2 \varepsilon, \tag{173}$$

*where $C_2 > 0$ depends only on $d$ and $\sigma$.*

*Proof.* We have proved in **Lemma F.3** that $\nabla_x G(x,y)$ and $\nabla_y G(x,y)$ is continuous, hence $G(x,y)$ is Lipschitz on $Q(R)$, and by the explicit expression of $\nabla G$, the Lipschitz constant $L_G$ depends only on $\sigma$. Since $|x - z| \leq \sqrt{d}\varepsilon$, $|y - w| \leq \sqrt{d}\varepsilon$, we have

$$|G(z,w) - G(x,y)| \leq |G(z,w) - G(z,y)| + |G(z,y) - G(x,y)| \leq 2L_G \sqrt{d}\varepsilon. \tag{174}$$

The proof for $H(x,y)$ is similar. $\qquad\square$

**Lemma F.8.** *We have*

$$
\mathbb{P}\left( \sup_{z,w \in \mathcal{N}_\varepsilon} |G_m(z,w) - G(z,w)| \leq C_3 \sqrt{\frac{\log m}{m}} \right) \geq 1 - 2\left(\frac{2R}{\varepsilon}\right)^{2d} m^{-\frac{1}{2} - 2d}, \tag{175}
$$

$$
\mathbb{P}\left( \sup_{z,w \in \mathcal{N}_\varepsilon} |H_m(z,w) - H(z,w)| \leq C_3 \sqrt{\frac{\log m}{m}} \right) \geq 1 - 2\left(\frac{2R}{\varepsilon}\right)^{2d} m^{-\frac{1}{2} - 2d}, \tag{176}
$$

*where $C_3 > 0$ depends only on $d$, $R$ and $\sigma$.*

*Proof.* For $x,y \in Q(R)$, the random variable $\sigma(W_1 \cdot x + B_1) \cdot \sigma(W_1 \cdot y + B_1)$ is polynomially bounded:

$$|\sigma(W_1 \cdot x + B_1) \cdot \sigma(W_1 \cdot y + B_1)| \leq (1 + (R \sum_{l=1}^{d} |W_{1,l}| + B_1)^p)^2 \tag{177}$$

hence it is clearly sub-exponential, and its sub-exponential norm has an upper bound $c' > 0$ depending only on $\sigma$ and $d$. Thus, by Bernstein's inequality (Section 2.8 of [39]), we have

$$\mathbb{P}(|G_m(x,y) - G(x,y)| \geq r) \leq 2\exp\left(-c'' m \cdot \min\{\frac{t^2}{r'^2}, \frac{r}{c'}\}\right), \tag{178}$$

where $c'' > 0$ is a universal constant.

Note that $|\mathcal{N}_\varepsilon| = (2R/\varepsilon)^d$, hence

$$\mathbb{P}\left( \sup_{z,w \in \mathcal{N}_\varepsilon} |G_m(z,w) - G(z,w)| < r \right) \geq 1 - 2\left(\frac{2R}{\varepsilon}\right)^{2d} \exp\left(-c'' m \cdot \min\{\frac{r^2}{c'^2}, \frac{r}{c'}\}\right). \tag{179}$$

By taking $r = \sqrt{\frac{(1+4d)c'^2 \log m}{2c''m}}$, we obtain

$$\mathbb{P}\left(\sup_{z,w \in \mathcal{N}_\varepsilon} |G_m(z,w) - G(z,w)| < C_3 \sqrt{\frac{\log m}{m}}\right) \geq 1 - 2\left(\frac{2R}{\varepsilon}\right)^{2d} m^{-\frac{1}{2}-2d}. \tag{180}$$

The proof for $H_m$ and $H$ is similar, and we also have

$$\mathbb{P}\left(\sup_{z,w \in \mathcal{N}_\varepsilon} |H_m(z,w) - H(z,w)| < C_3 \sqrt{\frac{\log m}{m}}\right) \geq 1 - 2\left(\frac{2R}{\varepsilon}\right)^{2d} m^{-\frac{1}{2}-2d}. \tag{181}$$

$\square$

Combining the above lemmata together, we obtain

$$\sup_{x,y \in Q(R)} |G_m(x,y) - G(x,y)| \leq C_1 (\log m)^{p+\frac{1}{2}} \varepsilon + C_2 \varepsilon + C_3 \sqrt{\frac{\log m}{m}} \tag{182}$$

$$\sup_{x,y \in Q(R)} |H_m(x,y) - H(x,y)| \leq C_1 (\log m)^{p+\frac{3}{2}} \varepsilon + C_2 \varepsilon + C_3 \sqrt{\frac{\log m}{m}} \tag{183}$$

with probability $1 - 2\left(\frac{2R}{\varepsilon}\right)^{2d} m^{-1/2-2d} - (d+2)m^{-1/2}$. By taking $\varepsilon = 1/m$, we have

$$\sup_{x,y \in Q(R)} |G_m(x,y) - G(x,y)| \leq C \sqrt{\frac{\log m}{m}}, \tag{184}$$

$$\sup_{x,y \in Q(R)} |H_m(x,y) - H(x,y)| \leq C \sqrt{\frac{\log m}{m}} \tag{185}$$

with probability $1 - cm^{-1/2}$, where the constants $C$ and $c$ depend only on $d$, $R$ and $\sigma$.
Since $\bar{\Omega} \subset Q(R)$, we obtain

**Lemma F.9.**

$$\sup_{x,y \in \bar{\Omega}} |K_{\theta(0)}(x,y) - K(x,y)| \leq C \sqrt{\frac{\log m}{m}} \tag{186}$$

*for some constant $C > 0$ depending only on $\Omega$ and $\sigma$, with probability $1 - \mathcal{O}(m^{-1/2})$.*

The proof for the derivatives of $K_{\theta(0)}$ and $K$ is similar. Following a similar discussion as above, it is easy to prove

**Lemma F.10.**

$$\sup_{x,y \in \bar{\Omega}} |\mathcal{L}_x K_{\theta(0)}(x,y) - \mathcal{L}_x K(x,y)| \leq C \sqrt{\frac{\log m}{m}} \tag{187}$$

$$\sup_{x,y \in \bar{\Omega}} |\mathcal{L}_y K_{\theta(0)}(x,y) - \mathcal{L}_y K(x,y)| \leq C \sqrt{\frac{\log m}{m}} \tag{188}$$

$$\sup_{x,y \in \bar{\Omega}} |\mathcal{L}_x \mathcal{L}_y K_{\theta(0)}(x,y) - \mathcal{L}_x \mathcal{L}_y K(x,y)| \leq C \sqrt{\frac{\log m}{m}} \tag{189}$$

*for some constant depending only on $\Omega$ and $\sigma$, with probability $1 - \mathcal{O}(m^{-1/2})$.*

## F.2 Convergence during training

**Lemma F.11.** *Conditioning on the event* $\mathcal{B}$ *defined in* **Definition F.5**, *along the training procedure, the parameters of the network satisfies:*

$$\sup_j \sup_{t \in [0,T]} |\theta_j(t) - \theta_j(0)| = \mathcal{O}(\frac{(\log m)^{2p}}{\sqrt{m}}) \tag{190}$$

*for* $\theta_j = A_k$, $W_{k,l}$ *or* $B_k$, *and the constant* $p \geq 1$ *is defined in* **Assumption 5.2**. *Here, the convention* $\mathcal{O}$ *hides the constants depending on* $d$, $T$, $\Omega$, $\sigma$, $f$ *and* $\mathcal{L}$.

*Proof.* Recall that the network is set to be

$$\hat{u}_{\theta(t)}(x) = \frac{1}{\sqrt{2m}} \sum_{k=1}^{2m} A_k(t)\sigma(\alpha_k(t,x)) + D(t), \tag{191}$$

where $\alpha_k(t,x) = W_k(t) \cdot x + B_k(t)$. Then

$$
\begin{aligned}
\mathcal{L}\hat{u}_{\theta(t)}(x) &= \frac{1}{\sqrt{2m}} \sum_{k=1}^{2m} A_k \mathcal{L}\sigma(\alpha_k(t,x)) \\
&= \frac{1}{\sqrt{2m}} \sum_{k=1}^{2m} A_k \left( \sum_{i,j=1}^{d} a_{ij} W_{k,i} W_{k,j} \sigma''(\alpha_k(t,x)) \right) \\
&\quad + \frac{1}{\sqrt{2m}} \sum_{k=1}^{2m} A_k \left( \sum_{j=1}^{d} \bar{b}_j W_{k,j} \sigma'(\alpha_k(t,x)) + c\sigma(\alpha_k(t,x)) \right),
\end{aligned} \tag{192}
$$

where $\bar{b}_j = b_j + \sum_{i=1}^{d} \partial_i a_{ij}$.

The gradient flow (9) can be explicitly computed as follows:

$$
\begin{aligned}
\frac{d}{dt} A_k &= -\frac{2}{N_u} \sum_{i=1}^{N_u} \frac{1}{\sqrt{2m}} \sigma(\alpha_k(t, X_i)) \cdot \hat{u}_{\theta(t)}(X_i) \\
&\quad - \frac{2}{N_f} \sum_{j=1}^{N_f} \frac{1}{\sqrt{2m}} \mathcal{L}\sigma(\alpha_k(t, Y_j)) \cdot (\mathcal{L}\hat{u}_{\theta(t)}(Y_j) - f(Y_j)),
\end{aligned} \tag{193}
$$

$$
\begin{aligned}
\frac{d}{dt} W_{k,l} &= -\frac{2}{N_u} \sum_{i=1}^{N_u} \frac{1}{\sqrt{2m}} A_k X_{i,l} \sigma(\alpha_k(t, X_i)) \cdot \hat{u}_{\theta(t)}(X_i) \\
&\quad - \frac{2}{N_f} \sum_{j=1}^{N_f} \frac{1}{\sqrt{2m}} A_k \left( 2\sum_{i=1}^{d} a_{il} W_{k,i} \cdot \sigma''(\alpha_k(t,x)) + \bar{b}_l \sigma'(\alpha_k(t,x)) \right) \cdot (\mathcal{L}\hat{u}_{\theta(t)}(Y_j) - f(Y_j)) \\
&\quad - \frac{2}{N_f} \sum_{j=1}^{N_f} \frac{1}{\sqrt{2m}} A_k \left( \sum_{m,n=1}^{d} a_{mn} W_{k,m} W_{k,n} \cdot Y_{j,l} \sigma'''(\alpha_k(t, Y_j)) \right) \cdot (\mathcal{L}\hat{u}_{\theta(t)}(Y_j) - f(Y_j)) \\
&\quad - \frac{2}{N_f} \sum_{j=1}^{N_f} \frac{1}{\sqrt{2m}} A_k \left( \sum_{m=1}^{d} \bar{b}_m W_{k,m} \cdot Y_{j,l} \sigma''(\alpha_k(t, Y_j)) \right) \cdot (\mathcal{L}\hat{u}_{\theta(t)}(Y_j) - f(Y_j)) \\
&\quad - \frac{2}{N_f} \sum_{j=1}^{N_f} \frac{1}{\sqrt{2m}} A_k \left( cY_{j,l} \sigma'(\alpha_k(t, Y_j)) \right) \cdot (\mathcal{L}\hat{u}_{\theta(t)}(Y_j) - f(Y_j)),
\end{aligned} \tag{194}
$$

$$\frac{d}{dt} B_k = -\frac{2}{N_u} \sum_{i=1}^{N_u} \frac{1}{\sqrt{2m}} A_k \sigma(\alpha_k(t, X_i)) \cdot \hat{u}_{\theta(t)}(X_i)$$

$$-\frac{2}{N_f} \sum_{j=1}^{N_f} \frac{1}{\sqrt{2m}} A_k \left( \sum_{m,n=1}^{d} a_{mn} W_{k,m} W_{k,n} \sigma'''(\alpha_k(t, Y_j)) \right) \cdot (\mathcal{L}\hat{u}_{\theta(t)}(Y_j) - f(Y_j))$$

$$-\frac{2}{N_f} \sum_{j=1}^{N_f} \frac{1}{\sqrt{2m}} A_k \left( \sum_{m=1}^{d} \bar{b}_m W_{k,m} \cdot \sigma''(\alpha_k(t, Y_j)) + c\sigma'(\alpha_k(t, Y_j)) \right) \cdot (\mathcal{L}\hat{u}_{\theta(t)}(Y_j) - f(Y_j)).$$

$$(195)$$

If $x \in Q(R) = [-R, R]^d$, then we have the following estimation:

$$|\alpha_k(t, x)| = |W_k(t) \cdot x + B_k(t)| \le R \sum_{l=1}^{d} |W_{k,l}(t)| + |B_k(t)|, \qquad (196)$$

hence by **Assumption 5.2** and Hölder's inequality, for some $p \ge 1$ depending on $\sigma$, we have

$$|\sigma(\alpha_k(t, x))| \lesssim 1 + |\alpha_k(t, x)|^p \lesssim 1 + \sum_{l=1}^{d} |W_{k,l}(t)|^p + |B_k(t)|^p, \qquad (197)$$

$$|\sigma'(\alpha_k(t, x))| \lesssim 1 + \sum_{l=1}^{d} |W_{k,l}(t)|^p + |B_k(t)|^p, \qquad (198)$$

$$|\sigma''(\alpha_k(t, x))| \lesssim 1 + \sum_{l=1}^{d} |W_{k,l}(t)|^p + |B_k(t)|^p, \qquad (199)$$

and

$$|\mathcal{L}\sigma(\alpha_k(t, x))|$$

$$= \left| \sum_{i,j=1}^{d} a_{ij} W_{k,i}(t) W_{k,j}(t) \sigma''(\alpha_k(t, x)) + \sum_{j=1}^{d} \bar{b}_j W_{k,j}(t) \sigma'(\alpha_k(t, x)) + c\sigma(\alpha_k(t, x)) \right|$$

$$\lesssim \sum_{i,j=1}^{d} \left( |W_{k,i}(t)|^3 + |W_{k,j}(t)|^3 + |\sigma''(\alpha_k(t, x))|^3 \right)$$

$$+ \sum_{j=1}^{d} \left( |W_{k,j}(t)|^2 + |\sigma'(\alpha_k(t, x))|^2 \right) + |\sigma(\alpha_k(t, x))|$$

$$\lesssim 1 + \sum_{l=1}^{d} |W_{k,l}|^{3p} + |B_k|^{3p}.$$

$$(200)$$

for some $p \geq 1$. Here, we use the mean value inequality $x_1 \ldots x_n \lesssim x_1^n + \cdots + x_n^n$ for $x_1, \ldots, x_n \geq 0$. Then, by Cauchy's inequality, we obtain that

$$
\left| \frac{d}{dt} |A_k(t) - A_k(0)| \right| \leq \left| \frac{d}{dt} (A_k(t) - A_k(0)) \right|
$$

$$
\leq \sqrt{\frac{2}{m}} \left| \frac{1}{N_u} \sum_{i=1}^{N_u} \sigma(\alpha(t, X_i)) \cdot \hat{u}_{\theta(t)}(X_i) + \frac{1}{N_f} \sum_{j=1}^{N_f} \mathcal{L}\sigma(\alpha_k(t, Y_j)) \cdot (\mathcal{L}\hat{u}_{\theta(t)}(Y_j) - f(Y_j)) \right|
$$

$$
\leq \sqrt{\frac{2}{m}} \sqrt{\frac{1}{N_u} \sum_{i=1}^{N_u} |\sigma(\alpha(t, X_i))|^2} \sqrt{\frac{1}{N_u} \sum_{i=1}^{N_u} |\hat{u}_{\theta(t)}(X_i)|^2}
$$

$$
+ \sqrt{\frac{2}{m}} \sqrt{\frac{1}{N_u} \sum_{j=1}^{N_f} |\mathcal{L}\sigma(\alpha(t, Y_j))|^2} \sqrt{\frac{1}{N_u} \sum_{j=1}^{N_f} |\mathcal{L}\hat{u}_{\theta(t)}(Y_j) - f(Y_j)|^2}
$$

$$
\lesssim \sqrt{\frac{2}{m}} \left( 1 + \sum_{l=1}^{d} |W_{k,l}(t)|^p + |B_k(t)|^p + 1 + \sum_{l=1}^{d} |W_{k,l}(t)|^{3p} + |B_k(t)|^{3p} \right) \cdot \sqrt{\hat{L}(\hat{u}_{\theta(t)})}
$$

$$
\lesssim \sqrt{\frac{2}{m}} \left( 1 + \sum_{l=1}^{d} |W_{k,l}(t)|^{3p} + |B_k(t)|^{3p} \right) \cdot \sqrt{\hat{L}(\hat{u}_{\theta(t)})}.
$$

$\qquad(201)$

Likewise, we also have

$$
\left| \frac{d}{dt} |W_{k,l}(t) - W_{k,l}(0)| \right| \lesssim \frac{1}{\sqrt{m}} \left( 1 + |A_k(t)|^{4p} + \sum_{l=1}^{d} |W_{k,l}(t)|^{4p} + |B_k(t)|^{4p} \right) \cdot \sqrt{\hat{L}(\hat{u}_{\theta(t)})},
$$

$\qquad(202)$

$$
\left| \frac{d}{dt} |B_k(t) - B_k(0)| \right| \lesssim \frac{1}{\sqrt{m}} \left( 1 + |A_k(t)|^{4p} + \sum_{l=1}^{d} |W_{k,l}(t)|^{4p} + |B_k(t)|^{4p} \right) \cdot \sqrt{\hat{L}(\hat{u}_{\theta(t)})}. \quad(203)
$$

Thus, if we define

$$
F_k(t) = |A_k(t) - A_k(0)| + \sum_{l=1}^{d} |W_{k,l}(t) - W_{k,l}(0)| + |B_k(t) - B_k(0)|, \qquad(204)
$$

which is a Lipschitz function, then

$$
|A_k(t)|^{4p} + \sum_{l=1}^{d} |W_{k,l}(t)|^{4p} + |B_k(t)|^{4p} \lesssim F_k(t)^{4p} + |A_k(0)|^{4p} + \sum_{l=1}^{d} |W_{k,l}(0)|^{4p} + |B_k(0)|^{4p}.
$$

$\qquad(205)$

Conditioning on the event $\mathcal{B}$ which is defined in **Lemma F.5**, we have

$$
|A_k(0)|^{4p} + \sum_{l=1}^{d} |W_{k,l}(0)|^{4p} + |B_k(0)|^{4p} \lesssim (\log m)^{2p}, \qquad(206)
$$

hence

$$
\left| \frac{d}{dt} F_k(t) \right| \leq \frac{(\log m)^{2p}}{\sqrt{m}} (1 + F_k(t)^{4p}) \cdot \sqrt{\hat{L}(\hat{u}_{\theta(t)})}, \qquad(207)
$$

where $C$ depends only on $d$, $\Omega$, $\mathcal{L}$, $\sigma$ and $\hat{L}(0)$. Since $\hat{L}(\hat{u}_{\theta(t)})$ is decreasing along the gradient flow, then

$$
\left| \frac{d}{dt} F_k(t) \right| \leq \frac{(\log m)^{2p}}{\sqrt{m}} (1 + F_k(t)^{4p}) \cdot \sqrt{\hat{L}(u_{\theta(0)})} \lesssim \frac{(\log m)^{2p}}{\sqrt{m}} (1 + F_k(t)^{4p}). \qquad(208)
$$

Therefore, conditioning on $\mathcal{B}$, for $m$ sufficiently great, we have

$$
F(t) = |F(t) - F(0)| \lesssim \frac{T(\log m)^{2p}}{\sqrt{m}} \qquad(209)
$$

for $t \in [0, T]$. $\qquad\qquad\square$

**Lemma F.12.** *Conditioning on the event $\mathcal{B}$, we have*

$$\sup_{x,y\in\bar{\Omega},\, t\in[0,T]} \left|K_{\theta(t)}(x,y) - K_{\theta(0)}(x,y)\right| = \mathcal{O}\left(\frac{(\log m)^{\frac{5p+3}{2}}}{\sqrt{m}}\right). \tag{210}$$

*Proof.* By definition, we have

$$K_{\theta(t)}(x,y) = 1 + G_m(t,x,y) + H_m(t,x,y) \tag{211}$$

where

$$G_m(t,x,y) = \frac{1}{2m}\sum_{k=1}^{2m}\sigma(\alpha_k(t,x))\cdot\sigma(\alpha_k(t,y)), \tag{212}$$

$$H_m(t,x,y) = \frac{1}{2m}\sum_{k=1}^{2m}A_k(t)^2(1+\langle x,y\rangle)\sigma'(\alpha_k(t,x))\cdot\sigma'(\alpha_k(t,y)). \tag{213}$$

Denote $\xi_k(t) = \sigma(\alpha_k(t,x))$, $\eta_k(t) = \sigma(\alpha_k(t,y))$, $\alpha_k(t,x) = W_k\cdot x + B_k$. Conditioning on the event $\mathcal{B}$, we have

$$|\alpha_k(0,x)| \lesssim \sqrt{\log m} \tag{214}$$

and

$$|\alpha_k(t,x) - \alpha_k(0,x)| = |(W_k(t) - W_k(0))\cdot x + B_k(t) - B_k(0))| \lesssim \frac{(\log m)^{2p}}{\sqrt{m}}, \tag{215}$$

then by mean value theorem,

$$\begin{aligned}
|\xi_k(t) - \xi_k(0)| &= |\sigma(\alpha_k(t,x)) - \sigma(\alpha_k(0,x))| \\
&= |\sigma'(\zeta)|\cdot|\alpha_k(t,x) - \alpha_k(0,x)| \\
&\lesssim (1+\zeta^p)\frac{(\log m)^{2p}}{\sqrt{m}}
\end{aligned} \tag{216}$$

for some value $\zeta$ between $\alpha_k(t,x)$ and $\alpha_k(0,x)$. Then $\zeta = \mathcal{O}(\sqrt{\log m})$, and

$$|\xi_k(t) - \xi_k(0)| \lesssim (1+(\sqrt{\log m})^p)\frac{(\log m)^{2p}}{\sqrt{m}}. \tag{217}$$

The same estimation also holds for $|\eta_k(t) - \eta_k(0)|$. Thus,

$$\begin{aligned}
&|\xi_k(t)\eta_k(t) - \xi_k(0)\eta_k(0)| \\
&\leq |\xi_k(t) - \xi_k(0)|\cdot|\eta_k(t) - \eta_k(0)| + |\xi_k(t) - \xi_k(0)|\cdot|\eta_k(0)| + |\xi_k(0)|\cdot|\eta_k(t) - \eta_k(0)| \\
&\lesssim \left((1+(\sqrt{\log m})^p)\frac{(\log m)^{2p}}{\sqrt{m}}\right)^2 + 2(1+(\sqrt{\log m})^p)\frac{(\log m)^{2p}}{\sqrt{m}}\cdot\sqrt{\log m} \\
&\lesssim \frac{(\log m)^{\frac{5p+1}{2}}}{\sqrt{m}}.
\end{aligned} \tag{218}$$

In other words,

$$|\sigma(\alpha_k(t,x))\cdot\sigma(\alpha_k(t,y)) - \sigma(\alpha_k(0,x))\cdot\sigma(\alpha_k(0,y))| \lesssim \frac{(\log m)^{\frac{5p+1}{2}}}{\sqrt{m}}. \tag{219}$$

Recall that

$$G_m(t,x,y) = \frac{1}{2m}\sum_{k=1}^{2m}\sigma(\alpha_k(t,x))\cdot\sigma(\alpha_k(t,y)), \tag{220}$$

then

$$|G_m(t,x,y) - G_m(0,x,y)| \lesssim \frac{(\log m)^{\frac{5p+1}{2}}}{\sqrt{m}}. \tag{221}$$

Similarly with (219), we have

$$|\sigma'(\alpha_k(t,x)) \cdot \sigma'(\alpha_k(t,y)) - \sigma'(\alpha_k(0,x)) \cdot \sigma'(\alpha_k(0,y))| \lesssim \frac{(\log m)^{\frac{5p+1}{2}}}{\sqrt{m}}, \qquad (222)$$

hence

$$
\begin{aligned}
&|H_m(t,x,y) - H_m(0,x,y)| \\
&\leq \frac{1}{2m} \sum_{k=1}^{2m} |A_k(t)|^2 |1 + \langle x,y \rangle| \cdot |\sigma'(\alpha_k(t,x)) \cdot \sigma'(\alpha_k(t,y)) - \sigma(\alpha_k(0,x)) \cdot \sigma(\alpha_k(0,y))| \\
&\lesssim \frac{1}{2m} \sum_{k=1}^{2m} M^2 (1 + dR^2) \cdot \frac{(\log m)^{\frac{5p+1}{2}}}{\sqrt{m}} \\
&\lesssim \frac{(\log m)^{\frac{5p+3}{2}}}{\sqrt{m}}.
\end{aligned}
\qquad (223)
$$

Combining (221) and (223), we conclude that

$$\sup_{x,y \in \bar{\Omega},\, t \in [0,T]} \left| K_{\theta(t)}(x,y) - K_{\theta(0)}(x,y) \right| = \mathcal{O}\left( \frac{(\log m)^{\frac{5p+3}{2}}}{\sqrt{m}} \right). \qquad (224)$$

$\square$

**Lemma F.13.** *Conditioning on $\mathcal{B}$, we have*

$$\sup_{x,y \in \bar{\Omega},\, t \in [0,T]} \left| \mathcal{L}_x K_{\theta(t)}(x,y) - \mathcal{L}_x K_{\theta(0)}(x,y) \right| = \mathcal{O}\left( \frac{(\log m)^{\frac{5p+5}{2}}}{\sqrt{m}} \right), \qquad (225)$$

$$\sup_{x,y \in \bar{\Omega},\, t \in [0,T]} \left| \mathcal{L}_y K_{\theta(t)}(x,y) - \mathcal{L}_y K_{\theta(0)}(x,y) \right| = \mathcal{O}\left( \frac{(\log m)^{\frac{5p+5}{2}}}{\sqrt{m}} \right), \qquad (226)$$

$$\sup_{x,y \in \bar{\Omega},\, t \in [0,T]} \left| \mathcal{L}_x \mathcal{L}_y K_{\theta(t)}(x,y) - \mathcal{L}_x \mathcal{L}_y K_{\theta(0)}(x,y) \right| = \mathcal{O}\left( \frac{(\log m)^{\frac{5p+7}{2}}}{\sqrt{m}} \right). \qquad (227)$$

*Proof.* For brevity, we only provide the proof for $\mathcal{L}_x K_{\theta(t)}$ here. The proof for $\mathcal{L}_y K_{\theta(t)}$ and $\mathcal{L}_x \mathcal{L}_y K_{\theta(t)}$ are exactly the same.

Note that

$$\mathcal{L}_x K_{\theta(t)}(x,y) = 1 + \mathcal{L}_x G_m(t,x,y) + H_m(t,x,y), \qquad (228)$$

where

$$G_m(t,x,y) = \frac{1}{2m} \sum_{k=1}^{2m} \mathcal{L}_x \sigma(\alpha_k(t,x)) \cdot \sigma(\alpha_k(t,y)), \qquad (229)$$

$$H_m(t,x,y) = \frac{1}{2m} \sum_{k=1}^{2m} A_k(t)^2 \mathcal{L}_x[(1 + \langle x,y \rangle)\sigma'(\alpha_k(t,x))] \cdot \sigma'(\alpha_k(t,y)). \qquad (230)$$

By computation, we have

$$\mathcal{L}_x \sigma(\alpha_k(t,x)) = \sum_{i,j=1}^{d} a_{ij} W_{k,i}(t) W_{k,j}(t) \sigma''(\alpha_k(t,x)) + \sum_{j=1}^{d} \bar{b}_j W_{k,j}(t) \sigma'(\alpha_k(t,x)) + c\sigma(\alpha_k(t,x)), \qquad (231)$$

where $\bar{b}_j = b_j + \sum_{i=1}^{d} \partial_i a_{ij}$. Similarly with (219), for any $x,y \in \bar{\Omega}$, $t \in [0,T]$, conditioning on $\mathcal{B}$, we have

$$|\sigma'(\alpha_k(t,x)) \cdot \sigma(\alpha_k(t,y)) - \sigma'(\alpha_k(0,x)) \cdot \sigma(\alpha_k(0,y))| \lesssim \frac{(\log m)^{\frac{5p+1}{2}}}{\sqrt{m}}, \qquad (232)$$

$$|\sigma''(\alpha_k(t,x)) \cdot \sigma(\alpha_k(t,y)) - \sigma''(\alpha_k(0,x)) \cdot \sigma(\alpha_k(0,y))| \lesssim \frac{(\log m)^{\frac{5p+1}{2}}}{\sqrt{m}}, \qquad (233)$$

hence

$$
\begin{aligned}
&|\mathcal{L}_x\sigma(\alpha_k(t,x))\cdot\sigma(\alpha_k(t,y)) - \mathcal{L}_x\sigma(\alpha_k(0,x))\cdot\sigma(\alpha_k(0,y))| \\
&\leq \sum_{i,j=1}^{d} |a_{ij}||W_{k,i}(t)||W_{k,j}(t)||\sigma''(\alpha_k(t,x))\sigma(\alpha_k(t,y)) - \sigma''(\alpha_k(0,x))\sigma(\alpha_k(0,y))| \\
&\quad + \sum_{j=1}^{d} |\bar{b}_j||W_{k,j}(t)||\sigma'(\alpha_k(t,x))\sigma(\alpha_k(t,y)) - \sigma'(\alpha_k(0,x))\sigma(\alpha_k(0,y))| \\
&\quad + |c||\sigma(\alpha_k(t,x))\sigma(\alpha_k(t,y)) - \sigma(\alpha_k(0,x))\sigma(\alpha_k(0,y))| \\
&\lesssim \sum_{i,j=1}^{d}(\sqrt{\log m})^2 \frac{(\log m)^{\frac{5p+1}{2}}}{\sqrt{m}} + \sum_{j=1}^{d}\sqrt{\log m}\frac{(\log m)^{\frac{5p+1}{2}}}{\sqrt{m}} + \frac{(\log m)^{\frac{5p+1}{2}}}{\sqrt{m}} \\
&\lesssim \frac{(\log m)^{\frac{5p+3}{2}}}{\sqrt{m}}.
\end{aligned}
\tag{234}
$$

Thus,

$$
\begin{aligned}
&|\mathcal{L}_x G_m(t,x,y) - \mathcal{L}_x G_m(0,x,y)| \\
&= \frac{1}{2m}\sum_{k=1}^{2m} |\mathcal{L}_x\sigma(\alpha_k(t,x))\cdot\sigma(\alpha_k(t,y)) - \mathcal{L}_x\sigma(\alpha_k(0,x))\cdot\sigma(\alpha_k(0,y))| \\
&\lesssim \frac{(\log m)^{\frac{5p+3}{2}}}{\sqrt{m}}.
\end{aligned}
\tag{235}
$$

Likewise, we have

$$
|\mathcal{L}_x H_m(t,x,y) - \mathcal{L}_x H_m(0,x,y)| \lesssim \frac{(\log m)^{\frac{5p+5}{2}}}{\sqrt{m}}.
\tag{236}
$$

In conclusion, we have

$$
\sup_{x,y\in\bar{\Omega},\, t\in[0,T]} \left|\mathcal{L}_x K_{\theta(t)}(x,y) - \mathcal{L}_x K_{\theta(0)}(x,y)\right| = \mathcal{O}\left(\frac{(\log m)^{\frac{5p+5}{2}}}{\sqrt{m}}\right).
\tag{237}
$$

$\square$

Now we begin to prove **Theorem 5.4**.

*Proof.* (of Theorem 5.4). By **Lemma F.9** and **Lemma F.12**, we have

$$
\sup_{x,y\in\bar{\Omega},t\in[0,T]} |K_{\theta(t)}(x,y) - K(x,y)| = \mathcal{O}\left(\sqrt{\frac{\log m}{m}} + \frac{(\log m)^{\frac{5p+3}{2}}}{\sqrt{m}}\right)
\tag{238}
$$

with probability $1 - \mathcal{O}(m^{-\frac{1}{2}})$. By **Lemma F.10** and **Lemma F.13**, we have

$$
\sup_{x,y\in\bar{\Omega},\, t\in[0,T]} \left|\mathcal{L}_x K_{\theta(t)}(x,y) - \mathcal{L}_x K(x,y)\right| = \mathcal{O}\left(\sqrt{\frac{\log m}{m}} + \frac{(\log m)^{\frac{5p+5}{2}}}{\sqrt{m}}\right),
\tag{239}
$$

$$
\sup_{x,y\in\bar{\Omega},\, t\in[0,T]} \left|\mathcal{L}_y K_{\theta(t)}(x,y) - \mathcal{L}_y K(x,y)\right| = \mathcal{O}\left(\sqrt{\frac{\log m}{m}} + \frac{(\log m)^{\frac{5p+5}{2}}}{\sqrt{m}}\right),
\tag{240}
$$

$$
\sup_{x,y\in\bar{\Omega},\, t\in[0,T]} \left|\mathcal{L}_x\mathcal{L}_y K_{\theta(t)}(x,y) - \mathcal{L}_x\mathcal{L}_y K(x,y)\right| = \mathcal{O}\left(\sqrt{\frac{\log m}{m}} + \frac{(\log m)^{\frac{5p+7}{2}}}{\sqrt{m}}\right)
\tag{241}
$$

with probability $1 - \mathcal{O}(m^{-\frac{1}{2}})$. $\square$

### F.3 Proofs for deep neural networks

In this section, we prove Theorem 5.4 for deep neural networks, while may omit some tedious computations in the proof.

In this section, the activation function of the network is set to be $\sigma = \tanh$, the depth of the network is $L \geq 9$, and the domain $\Omega$ is assumed to be convex.

It is easy to see that the NNK of $\hat{u}_\theta(x)$ coincides with that of $\hat{u}'_\theta(x)$ defined by

$$z^1(x) = W^0 x + b^0, \quad z^{l+1}(x) = \frac{1}{\sqrt{m_l}} W^l \sigma(z^l(x)) + b^l, \quad \hat{u}'_\theta(x) = z^{L+1}(x), \tag{242}$$

where $m_0 = d, m_1, \ldots, m_L, m_{L+1} = 1$ are the widths of the layers of $\hat{u}'_\theta$, and $W^l \in \mathbb{R}^{m_l \times m_{l+1}}, b^l \in \mathbb{R}^{m_{l+1}}$ are initialized as independent standard Gaussian random variable. We also recall that

$$cm \leq m_1, \ldots, m_{L-1} \leq Cm$$

for some $0 < c < C$.

The neural network kernel (NNK) of the $l$-th layer $z^l$ is defined by

$$K^l_{\theta, ij}(x, y) = \nabla_{\theta_l} z^l_i(x) \cdot \nabla_{\theta_l} z^l_j(y), \tag{243}$$

where $\theta_l$ denotes the parameters of the first $l$ layers. As an analogue of (146), the NNK of $z^{l+1}(x)$ is computed by

$$K^{l+1}_{\theta, ij}(x, y) = \delta_{ij} + \frac{\delta_{ij}}{m_l} \sum_{k=1}^{m_l} \sigma(z^l_k(x)) \sigma(z^l_k(y)) + \frac{1}{m_l} \sum_{p,q=1}^{m_l} W^l_{ip} W^l_{jq} \sigma'(z^l_p(x)) \sigma'(z^l_q(y)) K^l_{\theta, pq}(x, y). \tag{244}$$

Our proof for the convergence of NNK to NTK at initial time (Lemma F.14 and Theorem F.15)follows the discussions in Section 3.1 of [17].

**Lemma F.14.** *As $m \to \infty$, the $l$-th layer $z^l(x)$ converges weakly in $C^2$ to $g^l(x)$, where $g^l(x)$ is a Gaussian process with i.i.d. rows and covariance function $G^l$ given by*

$$\begin{aligned} G^1(x, y) &= 1 + \langle x, y \rangle, \\ G^{l+1}(x, y) &= 1 + \mathrm{E}_{f \sim N(0, G^l)}[\sigma(f(x)) \sigma(f(y))]. \end{aligned} \tag{245}$$

*Proof.* The smoothness of $z^l$ and $g^l$ arises from the smoothness of $\sigma = \tanh$.

Our proof is by induction. The proof for $l = 1$ is trivial, since

$$\mathrm{Cov}(z^1_i(x), z^1_i(y)) = \mathrm{Cov}(W^0_i x + b^0, W^0_i y + b^0) = 1 + \langle x, y \rangle, \quad i = 1, \ldots, m_1 \tag{246}$$

and $z^1_i(x), z^1_j(x)$ are independent for $i \neq j$.

Assume that the statement of this lemma holds for $1, \ldots, l - 1$. For $l \geq 2$, it is well-known (see Proposition 2.1 of [20] for example) that for every fixed $x \in \bar{\Omega}$, $z^l(x)$ converges weakly to $g^l(x)$. By Lemma B.1 of [17], in order to complete the proof, it suffices to show that for any $\delta \in (0, 1)$, there exists some constant $C_l$ depending on $l$, $\delta$, $\sigma$, $m_l$ and $\Omega$ such that

$$\|z^l(x)\|_{C^{2,1}} \leq C_l. \tag{247}$$

Define $\varphi^h(x) = \frac{1}{\sqrt{m_h}} \sigma(W^{h-1} x + b^{h-1})$ and $\varphi^l(x) = W^{l-1} x + b^{l-1}$. Then

$$z^l(x) = \varphi^l \circ \varphi^{l-1} \circ \cdots \circ \varphi^1(x). \tag{248}$$

By Lemma B.4 of [17], there exists a constant $C_1$ depending only on $\Omega$, $\sigma$ and $\delta$ such that

$$\mathbb{P}(A_1) > 1 - \frac{\delta}{l}, \quad \text{where } A_1 = \{\|\varphi^1(x)\|_{C^{2,1}} \leq C_1\}. \tag{249}$$

Then, by Lemma B.4 and 2.1 of [17], and using the fact that $W^0$ and $W^1$ are independent, we obtain that there exists some constant $C_2$ depending only on $\Omega$, $\sigma$ and $\delta$ such that

$$
\begin{aligned}
\mathbb{P}(\|\varphi^2 \circ \varphi^1\|_{C^{2,1}} \geq C_2) &\geq \mathbb{P}(\|\varphi^2 \circ \varphi^1\|_{C^{2,1}} \leq C_2, A_1) \\
&= \mathrm{E}(I_{A_1} \mathbb{P}(\|\varphi^2 \circ \varphi^1\|_{C^{2,1}} \leq C_2 | A_1)) \\
&\geq P(A_1) \cdot (1 - \frac{\delta}{l}) \\
&\geq 1 - \frac{2}{l}\delta.
\end{aligned}
\tag{250}
$$

Then, repeating this procedure, we can obtain the conclusion. $\qquad\square$

**Theorem F.15.** *For any $l = 1, \ldots, L+1$ and any multi-indices $\alpha$ and $\beta$ with $|\alpha|, |\beta| \leq 2$, with $m_l$ fixed, as $m \to \infty$, we have*

$$
\nabla_x^\alpha \nabla_y^\beta K_{\theta,ij}^l(x,y) \xrightarrow{p} \delta_{ij} \nabla_x^\alpha \nabla_y^\beta K^l(x,y)
\tag{251}
$$

*in $C^0$, where the neural tangent kernel $K^l(x,y)$ is determined by*

$$
\begin{aligned}
K^1(x,y) &= 1 + \langle x, y \rangle, \\
K^{l+1}(x,y) &= 1 + G^{l+1}(x,y) + K^l(x,y)\dot{G}^{l+1}(x,y),
\end{aligned}
\tag{252}
$$

*and*

$$
\dot{G}^{l+1}(x,y) = \mathrm{E}_{f \sim N(0,G^l)}[\sigma'(f(x))\sigma'(f(y))].
\tag{253}
$$

*Proof.* The smoothness of $K_{\theta,ij}^l$ and $K^l$ arises from the smoothness of $\sigma = \tanh$.

The proof is by induction. For $l = 1$, the proof is trivial:

$$
K_{\theta,ij}^1(x,y) = \delta_{ij}(1 + \langle x, y \rangle) = \delta_{ij}K^1(x,y).
\tag{254}
$$

Assume that the statement is true for $1, \ldots, l-1$. For $l \geq 2$, the following pointwise convergence is well-known (we refer to [1, 43] for details):

$$
K_{\theta,ij}^l(x,y) \xrightarrow{w} \delta_{ij}K^l(x,y), \quad x,y \in \bar{\Omega}.
\tag{255}
$$

In order to extend this pointwise convergence to the uniform convergence of kernel functions and their derivatives, by Lemma B.2 of [17], it suffices to verify that for any $\delta \in (0,1)$, there exists some constant $C_l$ depending only on $l$, $\delta$ and $\Omega$ such that

$$
\mathbb{P}(\|K_{l,ij}^\theta(x,y)\|_{C^{(2,1),(2,1)}(\bar{\Omega} \times \bar{\Omega})} \leq C_l) > 1 - \delta,
\tag{256}
$$

where $C^{(2,1),(2,1)}(\bar{\Omega} \times \bar{\Omega})$ denotes the set of all the functions $k(x,y)$, $x,y \in \bar{\Omega}$ such that $k(x,y)$ is $C^{2,1}$ for $x$ with $y$ fixed, and is also $C^{2,1}$ for $y$ with $x$ fixed.

By basic inequality $2ab \leq a^2 + b^2$ and Proposition B.7 of [17], for any multi indices $\alpha, \beta$ with $|\alpha|, |\beta| \leq 3$, we have

$$
\mathrm{E}\left[ \sup_{x,y \in \bar{\Omega}} \left| D_x^\alpha D_y^\alpha \sigma(g_k^l(x))\sigma(g_k^l(y)) \right| \right] < C < \infty
\tag{257}
$$

for some constant $C > 0$ depending only on $\Omega$ and $l$, where $g^l$ is the Gaussian process defined in Lemma F.14. Then, by Lemma F.14 and the induction hypothesis, we obtain that for any $M > 0$, for any $m_l$ fixed and as $cm < m_1, \ldots, m_{l-1} < Cm$, $m \to \infty$, we have

$$
\begin{aligned}
&\lim_{m \to \infty} \mathrm{E}\left[ \sup_{x,y \in \bar{\Omega}} \left| \frac{\delta_{ij}}{m_l} \sum_{k=1}^{m_l} D_x^\alpha D_y^\beta \sigma(z_k^l(x))\sigma(z_k^l(y)) \right| \wedge M \right] \\
&\leq \lim_{m \to \infty} \mathrm{E}\left[ \sup_{x,y \in \bar{\Omega}} \frac{1}{m_l} \sum_{k=1}^{m_l} \left| D_x^\alpha D_y^\beta \sigma(z_k^l(x))\sigma(z_k^l(y)) \right| \wedge M \right] \\
&= \mathrm{E}\left[ \sup_{x,y \in \bar{\Omega}} \frac{1}{m_l} \sum_{k=1}^{m_l} \left| D_x^\alpha D_y^\alpha \sigma(g_k^l(x))\sigma(g_k^l(y)) \right| \wedge M \right] \\
&\leq \mathrm{E}\left[ \sup_{x,y \in \bar{\Omega}} \frac{1}{m_l} \sum_{k=1}^{m_l} \left| D_x^\alpha D_y^\alpha \sigma(g_k^l(x))\sigma(g_k^l(y)) \right| \right] < C < \infty,
\end{aligned}
\tag{258}
$$

hence, if $m$ is sufficiently great, then there exists a constant $C' > 0$ depending only on $l$ and $\Omega$ such that

$$\mathbb{P}\left(\sup_{x,y\in\bar\Omega}\left|\frac{\delta_{ij}}{m_l}\sum_{k=1}^{m_l}D_x^\alpha D_y^\beta\sigma(z_k^l(x))\sigma(z_k^l(y))\right| > M\right)$$
$$\leq\frac{1}{M}\mathrm{E}\left[\sup_{x,y\in\bar\Omega}\left|\frac{\delta_{ij}}{m_l}\sum_{k=1}^{m_l}D_x^\alpha D_y^\beta\sigma(z_k^l(x))\sigma(z_k^l(y))\right|\wedge M\right]\leq\frac{C'}{M}. \tag{259}$$

Thus, we can choose $M$ sufficiently great (depending on $\delta$ and $C'$) such that

$$\mathbb{P}\left(\sup_{x,y\in\bar\Omega}\left|D_x^\alpha D_y^\beta\left(\frac{\delta_{ij}}{m_l}\sum_{k=1}^{m_l}\sigma(z_k^l(x))\sigma(z_k^l(y))\right)\right| > M\right) < \delta. \tag{260}$$

Next, for the estimation of the remaining terms of $K_{\theta,ij}^l(x,y)$ in (244), we make the following decomposition:

$$\frac{1}{m_l}\sum_{p,q=1}^{m_l}W_{ip}^lW_{jq}^l\sigma'(z_p^l(x))\sigma'(z_q^l(y))K_{\theta,pq}^l(x,y)$$
$$=\frac{1}{m_l}\sum_{p=1}^{m_l}W_{ip}^lW_{jp}^l\sigma'(z_p^l(x))\sigma'(z_p^l(y))K_{\theta,pp}^l(x,y) \tag{261}$$
$$+\frac{1}{m_l}\sum_{p\neq q}W_{ip}^lW_{jq}^l\sigma'(z_p^l(x))\sigma'(z_q^l(y))K_{\theta,pq}^l(x,y)$$

Similarly with (260, we can also prove that for $m$ sufficiently great, we have

$$\mathbb{P}\left(\left\|\frac{1}{m_l}\sum_{p=1}^{m_l}W_{ip}^lW_{jp}^l\sigma'(z_p^l(x))\sigma'(z_p^l(y))K_{\theta,pp}^l(x,y)\right\|_{C^{3,3}(\bar\Omega\times\bar\Omega)} > M\right) < \delta. \tag{262}$$

For the second term on the right-hand side of (261), we first note that there exists some constant $C''$ depending only on $l$ and $\Omega$ such that

$$\mathbb{P}\left(\sup_{x,y\in\bar\Omega}\frac{1}{m_l^2}\sum_{p\neq q}\left(\nabla_x^\alpha\nabla_y^\beta\sigma'(z_p^l(x))\sigma'(z_q^l(y))K_{\theta,pq}^l(x,y)\right)^2 > M\right) \leq \frac{C''}{M}, \tag{263}$$

for any $M > 0$ (the proof is similar with (260)). In other words,

$$\mathbb{P}(\|F(x,y)\|_{C^{3,3}(\bar\Omega\times\bar\Omega)} > M) < \frac{C''}{M}, \tag{264}$$

where

$$F(x,y):\bar\Omega\times\bar\Omega\to\mathbb{R}^{m_l(m_l-1)},\quad F_{pq}(x,y)=\frac{1}{m_l}\sigma'(z_p^l(x))\sigma'(z_q^l(y)). \tag{265}$$

We also note that, by the tail estimation for Gaussian distributions, the map

$$\varphi:\mathbb{R}^{m_l(m_l-1)}\to\mathbb{R},\quad \varphi(x)=\sum_{p\neq q}W_{ip}^lW_{jq}^lx_{pq} \tag{266}$$

satisfies

$$\mathbb{P}(\|\varphi\|_{C^3} > C\log N)\leq N^{-10},\quad \forall N > 0 \tag{267}$$

for some universal constant $C > 0$. Thus, by selecting appropriate $M$ and $N$, we can prove that there exists some constant $C'''$ depending only on $\delta$, $\Omega$ and $l$ such that

$$\mathbb{P}(\|\varphi\circ F(x,y)\|_{C^{(2,1),(2,1)}(\bar\Omega\times\bar\Omega)} > C''') \leq \delta, \tag{268}$$

in other words,

$$\mathbb{P}\left(\left\|\frac{1}{m_l}\sum_{p,q=1}^{m_l}W_{ip}^lW_{jq}^l\sigma'(z_p^l(x))\sigma'(z_q^l(y))K_{\theta,pq}^l(x,y)\right\|_{C^{3,3}(\bar\Omega\times\bar\Omega)} > M\right) < \delta. \tag{269}$$

Combining (260), (269) and (262) together, we obtain (256), hence complete the proof of this theorem. □

In the following results (Lemma F.16, Theorem F.17 and Theorem F.18), we establish convergence of NNK during training. Since the network $\hat{u}_\theta(x) = \frac{1}{\sqrt{2}}(z_1^{L+1}(x) - z_2^{L+1}(x))$ consists of two independent parts $z_1^{L+1}$ and $z_2^{L+1}$, we only need to prove for the two parts separately. Therefore, for brevity of computations, we prove the results for the fully-connected network $\hat{u}_\theta'(x)$ defined in (242) instead, and the proofs for $\hat{u}_\theta$ are almost the same.

**Lemma F.16.** *Define $\mathcal{B}'$ to be the event that all the parameters $\theta_k = W_{ij}^l$ or $b_i^l$ are bounded by $M = c\sqrt{\log m}$ at the initial time $t = 0$:*

$$\mathcal{B}' = \{|W_{ij}^l|, |b_i^l| \leq M\}, \tag{270}$$

*then for some constant $c, C > 0$, we have $\mathbb{P}(\mathcal{B}) \geq 1 - Cm^{-1/2}$.*

*Proof.* The proof is similar with that of Lemma F.5. □

**Theorem F.17.** *Conditioning on the event $\mathcal{B}'$, along the training procedure, for any $T > 0$ fixed, the parameters $\theta_k = W_{ij}^l$ or $b_i^l$ of the network satisfies*

$$\sup_k \sup_{t \in [0,T]} |\theta_k(t) - \theta_k(0)| = \mathcal{O}(\sqrt{\frac{\log m}{m}}). \tag{271}$$

*Proof.* This Lemma is an analogue of Lemma F.11. Recall that the evolution equation of $\theta_k$ is

$$\frac{d}{dt}\theta_k = -\frac{2}{N_u}\sum_{i=1}^{N_u}\frac{\partial z^{L+1}(X_i)}{\partial \theta_k} \cdot z^{L+1}(x_i) - \frac{2}{N_f}\sum_{j=1}^{N_f}\frac{\partial \mathcal{L}z^{L+1}(Y_j)}{\partial \theta_k} \cdot (\mathcal{L}z^{L+1}(Y_j) - f(Y_j)). \tag{272}$$

For any $n = 1, \ldots, d$, by direct computations we have

$$\frac{\partial}{\partial x_n}z^{L+1}(x) = \frac{1}{\sqrt{m_1 \cdots m_L}}W^L \cdot \mathrm{diag}(\dot{\sigma}(z^L(x))) \cdots W^1 \cdot \mathrm{diag}(\dot{\sigma}(z^1(x))) \cdot W_{\cdot,n}^0, \tag{273}$$

where $W_{\cdot,n}^0$ is the $n$-th column of $W^0$. Next, we compute the derivatives of $\frac{\partial}{\partial x_n}z^{L+1}(x)$ with respect to $W_{ij}^l$. Note that

$$\frac{\partial}{\partial W_{1k}^L}\left(\frac{\partial}{\partial x_n}z^{L+1}(x)\right) = \frac{1}{\sqrt{m_1 \cdots m_L}}\dot{\sigma}(z_k^L(x)) \cdot W_{k,\cdot}^{L-1} \cdots W^1 \cdot \mathrm{diag}(\dot{\sigma}(z^1(x))) \cdot W_{\cdot,n}^0. \tag{274}$$

Define the matrix norm $\|\cdot\|$ by $\|M\| = \sum_{i,j}|M_{ij}|$ for $M = (M_{ij})$. It is well-known that $\|MN\| \leq \|M\| \cdot \|N\|$. Also note that $\sigma = \tanh$ has bounded derivative $|\dot{\sigma}| \leq C$. Then we have

$$\begin{aligned}\left|\frac{\partial}{\partial W_{1k}^L}\left(\frac{\partial}{\partial x_n}z^{L+1}(x)\right)\right| &\leq \frac{\|W_{k,\cdot}^{L-1}\| \cdots \|W_{\cdot,n}^0\|}{\sqrt{m_1 \cdots m_L}} \\ &\leq \frac{\|W^{L-1}\| \cdots \|W^0\|}{\sqrt{m_1 \cdots m_L}} \\ &\leq \frac{C}{\sqrt{m_1 \cdots m_L}}(1 + \sum_{l,i,j}|W_{ij}^l|)^L\end{aligned} \tag{275}$$

Likewise, for other parameters $W_{ij}^l$, $l = 0, \ldots, L-1$, we can also prove that (we omit the tedious computations here)

$$\left|\frac{\partial}{\partial W_{ij}^l}\left(\frac{\partial}{\partial x_n}z^{L+1}(x)\right)\right| \leq \frac{C}{\sqrt{m_1 \cdots m_L}}(1 + \sum_{l,i,j}|W_{ij}^l|)^\alpha, \tag{276}$$

$$\left|\frac{\partial}{\partial W_{ij}^l}\left(\frac{\partial^2}{\partial x_n \partial x_m}z^{L+1}(x)\right)\right| \leq \frac{C}{\sqrt{m_1 \cdots m_L}}(1 + \sum_{l,i,j}|W_{ij}^l|)^\alpha \tag{277}$$

for some $\alpha$ depending only on $L$. Combining (276), (277) with (272), we obtain that

$$\left| \frac{\partial}{\partial t} \sum_{l,i,j} |W_{ij}^l(t) - W_{ij}^l(0)| \right|$$
$$\leq \frac{C(m_L + m_L m_L - 1 + \cdots + m_1 m_0 + m_0 d)}{\sqrt{m_1 \cdots m_L}} (1 + \sum_{l,i,j} |W_{ij}^l(t)|)^\alpha. \tag{278}$$

Recall that $L \geq 9$ and $cm \leq m_1, \ldots, m_L \leq Cm$, hence

$$\left| \frac{\partial}{\partial t} \sum_{l,i,j} |W_{ij}^l(t) - W_{ij}^l(0)| \right| \leq \frac{C}{m^{5/2}} (1 + \sum_{l,i,j} |W_{ij}^l(t)|)^\alpha. \tag{279}$$

We also note that conditioning on $\mathcal{B}'$, we have

$$\sum_{l,i,j} |W_{ij}^l(0)| \leq Cm^2 \sqrt{\log m}. \tag{280}$$

Finally, following the same procedure from (204) to (208) in the proof of Lemma F.11, we can prove that

$$F(t) \lesssim T \frac{\sqrt{\log m}}{\sqrt{m}} \tag{281}$$

for $m$ sufficiently great, where

$$F(t) = \sum_{l,i,j} |W_{ij}^l(t) - W_{ij}^l(0)|. \tag{282}$$

This completes the proof. $\qquad \square$

Since $m_{L+1} = 1$, we denote $K_\theta(x,y) = K_{\theta,11}^{L+1}(x,y)$ for simplicity.

**Theorem F.18.** *Conditioning on $\mathcal{B}'$, for $l = 1, \ldots, L+1$, we have*

$$\sup_{x,y \in \bar{\Omega}, t \in [0,T]} |K_{\theta(t)}(x,y) - K_{\theta(0)}(x,y)| \to 0 \tag{283}$$

*as $m \to \infty$.*

*Proof.* We start by estimating the perturbation of $z^l(\theta, x)$. Note that for $l = 1$, we have

$$z^l(\theta(t), x) - z^l(\theta(0), x) = (W^0(t) - W^0(0))x + (b^0(t) - b^0(0)). \tag{284}$$

The result (271) in the proof of Lemma F.17 in fact implies that $\|W^0(t) - W^0(0)\|_2^2 \lesssim \frac{\log m}{m}$ and $|b^0(t) - b^0(0)|^2 \lesssim \frac{\log m}{m}$, where $\|\cdot\|_2$ is the matrix 2-norm and $|\cdot|$ is the Euclidean norm. Thus, we obtain that

$$\sup_x |z^l(\theta(t), x) - z^l(\theta(0), x)|^2 \lesssim \frac{\log m}{m}. \tag{285}$$

Then, for $l = 2$, we note that

$$z^2(\theta(t), x) - z^2(\theta(0), x) = \frac{1}{\sqrt{m_1}} (W^1(t) - W^1(0)) \sigma(z^1(\theta(t), x))$$
$$+ \frac{1}{\sqrt{m_1}} W^1(0) (\sigma(z^1(\theta(t), x)) - \sigma(z^1(\theta(0), x))) \tag{286}$$
$$+ (b^1(t) - b^1(0)).$$

Again, by (271), we have $\|W^1(t) - W^1(0)\|_2^2 \lesssim \frac{\log m}{m}$ and $|b^0(t) - b^0(0)|^2 \lesssim \frac{\log m}{m}$. Using the fact that $|\sigma'|$ is bounded, we also have

$$|\sigma(z^1(\theta(t), x)) - \sigma(z^1(\theta(0), x))|^2 \leq C|z^l(\theta(t), x) - z^l(\theta(0), x)|^2 \lesssim \frac{\log m}{m}. \tag{287}$$

We also note that $\|\frac{1}{\sqrt{m_l}}W^1(0)\|_2^2 \lesssim \log m$ conditioning on $\mathcal{B}'$. Therefore, we obtain that

$$\sup_x |\sigma(z^1(\theta(t),x)) - \sigma(z^1(\theta(0),x))|^2 \lesssim \frac{(\log m)^2}{m}. \tag{288}$$

Repeating this procedure, we can prove that for any $l = 1, \ldots, L$, we have

$$\sup_x |\sigma(z^l(\theta(t),x)) - \sigma(z^l(\theta(0),x))|^2 \lesssim \frac{(\log m)^l}{m}. \tag{289}$$

Next, we estimate the perturbation of $D_\theta z^l = (\frac{\partial z_i^l}{\partial \theta_j})$. Denote $\theta^l$ to be the parameters of the first $l$ layers.

For $l = 1$, since $z^l(\theta,x)$ is linear with respect to $\theta$, we have

$$\left\|D_{\theta^1} z_i^1(\theta(t),x) - D_{\theta^1} z^1(\theta(0),x)\right\|_2^2 = 0, \quad i = 1, \ldots, m_1. \tag{290}$$

Moreover, it is easy to see that

$$\left\|D_{\theta^1} z^1(\theta(0),x)\right\|_2^2 \leq C < \infty, \tag{291}$$

where $C > 0$ depends only on $\Omega$.

Then, for $l = 2$, note that

$$\frac{\partial z_i^2(\theta,x)}{\partial W_{ij}^1} = \frac{1}{\sqrt{m_1}}\sigma(z_j^1(\theta,x)), \tag{292}$$

$$D_{\theta^1} z_i^2(\theta,x) = \frac{1}{\sqrt{m_1}}\sum_{j=1}^{m_1} W_{ij}^1 D_{\theta^1}\sigma(z_j^1(\theta,x)). \tag{293}$$

Thus, using (292), (291) and the fact that $\|\frac{1}{m_1}W_{ij}^1\|_2^2 \lesssim \log m$, it is easy to prove that (with some tedious computations omitted) conditioning on $\mathcal{B}'$, we have

$$\|D_{\theta^2} z_i^2(\theta(t),x) - D_{\theta^2} z_i^2(\theta(0),x)\|_2^2 \lesssim \frac{\log m}{m}, \tag{294}$$

$$\|D_{\theta^2} z^2(\theta(0),x)\|_2^2 \lesssim \log m. \tag{295}$$

Repeating this procedure for $l = 1, \ldots, L+1$, we conclude that

$$|D_\theta z^{L+1}(\theta(0),x)|^2 \lesssim (\log m)^\alpha, \tag{296}$$

$$|D_\theta z^{L+1}(\theta(t),x) - D_\theta z^{L+1}(\theta(0),x)|^2 \lesssim \frac{(\log m)^\alpha}{m}, \tag{297}$$

where $\alpha > 0$ is a constant depending only on $L$.

Finally, since

$$K_\theta(x,y) = \langle D_\theta z^{L+1}(\theta,x), D_\theta z^{L+1}(\theta,x)\rangle, \tag{298}$$

by (296), (297) and Cauchy-Schwarz inequality, we conclude that conditioning on $\mathcal{B}'$, we have

$$\sup_{x,y\in\bar{\Omega},t\in[0,T]} |K_{\theta(t)}(x,y) - K_{\theta(0)}(x,y)| \lesssim \frac{(\log m)^\alpha}{\sqrt{m}}. \tag{299}$$

$\square$

Similarly with Theorem F.18, we also have

**Theorem F.19.** *Conditioning on $\mathcal{B}'$, for $l = 1, \ldots, L+1$, we have*

$$\sup_{x,y\in\bar{\Omega},t\in[0,T]} |\mathcal{L}_x K_{\theta(t)}(x,y) - \mathcal{L}_x K_{\theta(0)}(x,y)| \to 0, \tag{300}$$

$$\sup_{x,y\in\bar{\Omega},t\in[0,T]} |\mathcal{L}_y K_{\theta(t)}(x,y) - \mathcal{L}_y K_{\theta(0)}(x,y)| \to 0, \tag{301}$$

$$\sup_{x,y\in\bar{\Omega},t\in[0,T]} |\mathcal{L}_x\mathcal{L}_y K_{\theta(t)}(x,y) - \mathcal{L}_x\mathcal{L}_y K_{\theta(0)}(x,y)| \to 0, \tag{302}$$

*as $m \to \infty$.*

*Proof.* The proof is similar with that of Theorem F.18. □

Combining Theorem F.15, F.18 and F.19 together, we complete the proof for Theorem 5.4 (II). We note that, as discussed earlier in this section, although Theorem F.15, F.18 and F.19 are proved for $\hat{u}'_\theta$, the proofs for $\hat{u}_\theta$ are almost the same. This is because the network $\hat{u}_\theta(x) = \frac{1}{\sqrt{2}}(z_1^{L+1}(x) - z_2^{L+1}(x))$ consists of two independent parts $z_1^{L+1}$ and $z_2^{L+1}$, both of which are fully-connected networks in the same form of $\hat{u}'_\theta$, and we only need to prove for the two parts separately.

# G Proof of Lemma 5.6

Since $\varepsilon_j$ is sub-Gaussian, we have the following tail estimation for $\varepsilon_j$:

$$\mathbb{P}(|\varepsilon_j| > r) \leq 2e^{-ct^2} \tag{303}$$

for some $c > 0$. Thus, for any $n, \beta > 0$, with great probability $1 - \mathcal{O}(n^{-\beta})$, we have

$$|Z_j| \lesssim \log n, \quad \text{where } Z_j = f(Y_j) + \varepsilon_j. \tag{304}$$

Consider the gradient flow (10):

$$\frac{d}{dt}\hat{u}_{\theta(t)}(x) = -\frac{2}{N_u}\sum_{i=1}^{N_u} K_{\theta(t)}(x, X_i) \cdot \hat{u}_{\theta(t)}(X_i)$$
$$-\frac{2}{N_f}\sum_{j=1}^{N_f} \mathcal{L}_y K_{\theta(t)}(x, Y_j) \cdot (\mathcal{L}\hat{u}_{\theta(t)}(Y_j) - Z_j), \tag{305}$$

which are both PDEs. To avoid thorny PDE problems, we add the gradient flow of $\mathcal{L}\hat{u}_\theta$ into consideration:

$$\frac{d}{dt}\mathcal{L}\hat{u}_{\theta(t)}(x) = -\frac{2}{N_u}\sum_{i=1}^{N_u} \mathcal{L}_x K_{\theta(t)}(x, X_i) \cdot \hat{u}_{\theta(t)}(X_i)$$
$$-\frac{2}{N_f}\sum_{j=1}^{N_f} \mathcal{L}_x \mathcal{L}_y K_{\theta(t)}(x, Y_j) \cdot (\mathcal{L}\hat{u}_{\theta(t)}(Y_j) - Z_j). \tag{306}$$

We also recall the gradient flow (11):

$$\frac{d}{dt}\hat{u}_t(x) = -\frac{2}{N_u}\sum_{i=1}^{N_u} K(x, X_i) \cdot \hat{u}_t(X_i) - \frac{2}{N_f}\sum_{j=1}^{N_f} \mathcal{L}_y K(x, Y_j) \cdot (\mathcal{L}\hat{u}_t(Y_j) - Z_j), \tag{307}$$

$$\frac{d}{dt}\mathcal{L}\hat{u}_t(x) = -\frac{2}{N_u}\sum_{i=1}^{N_u} \mathcal{L}_x K(x, X_i) \cdot \hat{u}_t(X_i) - \frac{2}{N_f}\sum_{j=1}^{N_f} \mathcal{L}_x \mathcal{L}_y K(x, Y_j) \cdot (\mathcal{L}\hat{u}_t(Y_j) - Z_j). \tag{308}$$

Note that the gradient flows of the following $2 + N_u + N_f$ functions

$$\hat{u}_{\theta(t)}(x), \mathcal{L}\hat{u}_{\theta(t)}(x), \hat{u}_{\theta(t)}(X_1), \ldots, \hat{u}_{\theta(t)}(X_{N_u}), \mathcal{L}\hat{u}_{\theta(t)}(Y_1), \ldots, \mathcal{L}\hat{u}_{\theta(t)}(Y_{N_f}) \tag{309}$$

form a linear ODE system, and the gradient flow (10) is completely determined by this system. Thus, by the uniform convergence of kernel function (**Lemma F.12** and **Lemma F.13**) and the continuous dependence theorem of ODE (**Lemma G.3**), we obtain

**Lemma G.1.** *Given $T > 0$, $N_u$, $N_f$ fixed, we have*

$$\sup_{t\in[0,T], x\in\bar{\Omega}} \sup_{X_i, Y_j} |\hat{u}_{\theta(t)}(x) - \hat{u}_t(x)| \to 0, \tag{310}$$

$$\sup_{t\in[0,T], x\in\bar{\Omega}} \sup_{X_i, Y_j} |\mathcal{L}\hat{u}_{\theta(t)}(x) - \mathcal{L}\hat{u}_t(x)| \to 0 \tag{311}$$

*in probability. The randomness arises from the network initialization.*

*Proof.* We apply **Lemma G.3** by setting $k = 2 + N_u + N_f$ and

$$A(t) = \begin{pmatrix} \mathbf{0}_{1\times 2} & -\frac{2}{N_u}K_{\theta(t)}(x,X) & -\frac{2}{N_f}\mathcal{L}_y K_{\theta(t)}(x,Y) \\ \mathbf{0}_{1\times 2} & -\frac{2}{N_u}\mathcal{L}_x K_{\theta(t)}(x,X) & -\frac{2}{N_f}\mathcal{L}_x\mathcal{L}_y K_{\theta(t)}(x,Y) \\ \mathbf{0}_{N_u\times 2} & -\frac{2}{N_u}K_{\theta(t)}(X,X) & -\frac{2}{N_f}\mathcal{L}_y K_{\theta(t)}(X,Y) \\ \mathbf{0}_{N_f\times 2} & -\frac{2}{N_u}\mathcal{L}_x K_{\theta(t)}(Y,X) & -\frac{2}{N_f}\mathcal{L}_x\mathcal{L}_y K_{\theta(t)}(Y,Y) \end{pmatrix}, \tag{312}$$

$$B(t) = \begin{pmatrix} \frac{2}{N_f}\mathcal{L}_y K_{\theta(t)}(x,Y)\cdot Z \\ \frac{2}{N_f}\mathcal{L}_x\mathcal{L}_y K_{\theta(t)}(x,Y)\cdot Z \\ \frac{2}{N_f}\mathcal{L}_y K_{\theta(t)}(X,Y)\cdot Z \\ \frac{2}{N_f}\mathcal{L}_x\mathcal{L}_y K_{\theta(t)}(Y,Y)\cdot Z \end{pmatrix}, \tag{313}$$

$$A = \begin{pmatrix} \mathbf{0}_{1\times 2} & -\frac{2}{N_u}K(x,X) & -\frac{2}{N_f}\mathcal{L}_y K(x,Y) \\ \mathbf{0}_{1\times 2} & -\frac{2}{N_u}\mathcal{L}_x K(x,X) & -\frac{2}{N_f}\mathcal{L}_x\mathcal{L}_y K(x,Y) \\ \mathbf{0}_{N_u\times 2} & -\frac{2}{N_u}K(X,X) & -\frac{2}{N_f}\mathcal{L}_y K(X,Y) \\ \mathbf{0}_{N_f\times 2} & -\frac{2}{N_u}\mathcal{L}_x K(Y,X) & -\frac{2}{N_f}\mathcal{L}_x\mathcal{L}_y K(Y,Y) \end{pmatrix}, \tag{314}$$

$$B = \begin{pmatrix} \frac{2}{N_f}\mathcal{L}_y K(x,Y)\cdot Z \\ \frac{2}{N_f}\mathcal{L}_x\mathcal{L}_y K(x,Y)\cdot Z \\ \frac{2}{N_f}\mathcal{L}_y K(X,Y)\cdot Z \\ \frac{2}{N_f}\mathcal{L}_x\mathcal{L}_y K(Y,Y)\cdot Z \end{pmatrix}, \tag{315}$$

where $\mathbf{0}_{m\times n}$ denotes the $m \times n$ zero matrix, and

$$K_{\theta(t)}(X,X) = (K_{\theta(t)}(X_i,X_j))_{N_u\times N_u}, \quad \mathcal{L}_y K_{\theta(t)}(X,Y) = (K_{\theta(t)}(X_i,Y_j))_{N_u\times N_f}, \tag{316}$$

$$\mathcal{L}_x K_{\theta(t)}(Y,X) = (K_{\theta(t)}(Y_i,X_j))_{N_f\times N_u}, \quad \mathcal{L}_x\mathcal{L}_y K_{\theta(t)}(Y,Y) = (K_{\theta(t)}(Y_i,Y_j))_{N_f\times N_f}, \tag{317}$$

$$K(X,X) = (K(X_i,X_j))_{N_u\times N_u}, \quad \mathcal{L}_y K(X,Y) = (K(X_i,Y_j))_{N_u\times N_f}, \tag{318}$$

$$\mathcal{L}_x K(Y,X) = (K(Y_i,X_j))_{N_f\times N_u}, \quad \mathcal{L}_x\mathcal{L}_y K(Y,Y) = (K(Y_i,Y_j))_{N_f\times N_f}, \tag{319}$$

$$K_{\theta(t)}(x,X) = (K_{\theta(t)}(x,X_j))_{1\times N_u}, \quad \mathcal{L}_y K_{\theta(t)}(x,Y) = (K_{\theta(t)}(x,Y_j))_{1\times N_f}, \tag{320}$$

$$\mathcal{L}_x K_{\theta(t)}(x,X) = (K_{\theta(t)}(x,X_j))_{x\times N_u}, \quad \mathcal{L}_x\mathcal{L}_y K_{\theta(t)}(x,Y) = (K_{\theta(t)}(x,Y_j))_{1\times N_f}, \tag{321}$$

$$K(x,X) = (K(x,X_j))_{1\times N_u}, \quad \mathcal{L}_y K(x,Y) = (K(x,Y_j))_{1\times N_f}, \tag{322}$$

$$\mathcal{L}_x K(x,X) = (K(x,X_j))_{1\times N_u}, \quad \mathcal{L}_x\mathcal{L}_y K(x,Y) = (K(Y_i,Y_j))_{1\times N_f}, \tag{323}$$

$$Z = (Z_1,\ldots,Z_{N_f})^T. \tag{324}$$

Note that

$$|K(x,y)|, |\mathcal{L}_y K(x,y)|, |\mathcal{L}_x K(x,y)|, |\mathcal{L}_x\mathcal{L}_y K(x,y)|, |f(y)| \le M_K \tag{325}$$

for $x,y \in \bar{\Omega}$, where the bound $M_K > 0$ depends only on $\sigma$.

Denote as $\|\cdot\|_\infty$ the maximum norm: $\|(a_{ij})\|_\infty = \max_{i,j}|a_{ij}|$ for a matrix $(a_{ij})$. Then

$$\|A\|_\infty \le 2\left(\frac{1}{N_u} + \frac{1}{N_f}\right)M_K. \tag{326}$$

By (304), we also have

$$\|B\|_\infty \lesssim \log m. \tag{327}$$

with great probability $1 - \mathcal{O}(m^{-1/2})$.

For shallow networks, by **Lemma F.12** and **Lemma F.13**, we have

$$\max_{t\in[0,T]} \|A(t) - A\|_\infty = \left(\frac{1}{N_u} + \frac{1}{N_f}\right)\mathcal{O}\left(\frac{(\log m)^{\frac{5p+7}{2}}}{\sqrt{m}}\right), \tag{328}$$

$$\max_{t\in[0,T]} \|B(t) - B\|_\infty = \frac{1}{N_f}\mathcal{O}\left(\frac{(\log m)^{\frac{5p+7}{2}}}{\sqrt{m}}\right) \tag{329}$$

with probability $\mathbb{P}(\mathcal{B}) \geq 1 - \mathcal{O}(m^{-1/2})$.

Finally, by **Lemma G.3**, we conclude that

$$
\begin{aligned}
\sup_{t\in[0,T],x\in\bar\Omega}\sup_{X_i,Y_j}|\hat{u}_{\theta(t)}(x)-\hat{u}_t(x)| &\leq Q(m),\\
\sup_{t\in[0,T],x\in\bar\Omega}\sup_{X_i,Y_j}|\mathcal{L}\hat{u}_{\theta(t)}(x)-\mathcal{L}\hat{u}_t(x)| &\leq Q(m),
\end{aligned}
\tag{330}
$$

with probability $1 - \mathcal{O}(m^{-1/2})$, where

$$
Q(m) = D(T, N_u, N_f)\frac{(\log m)^{\frac{5p+7}{2}}}{\sqrt{m}}
\tag{331}
$$

for some constant $C > 0$, and

$$
D(T, N_u, N_f) = CT(2+N_u+N_f)(1+T(2+N_u+N_f)e^{2T(2+N_u+N_f)(\frac{1}{N_u}+\frac{1}{N_f})M_K})(\frac{1}{N_u}+\frac{1}{N_f}).
\tag{332}
$$

For deep networks, using Theorem F.15, F.18 and F.19, we can also complete the proof similarly with the above procedure from (328) to (332). $\qquad\square$

## G.1 Continuous dependence of ODE

**Lemma G.2.** *Assume that $X(t) = (x_1(t), \ldots, x_k(t))$ is the solution to the following $k$-order ODE system:*

$$
\frac{d}{dt}X(t) = A(t)X(t) + B(t),
\tag{333}
$$

*where $A(t)$ and $B(t)$ are continuous with respect to $t \in [0,T]$. If $X(0) = 0$, then*

$$
\max_{t\in[0,T]}\|X(t)\|_\infty \leq Tk\exp\left(Tk\max_{t\in[0,T]}\|A(t)\|_\infty\right)\cdot\max_{t\in[0,T]}\|B(t)\|_\infty.
\tag{334}
$$

*Proof.* Note that

$$
\left|\frac{d}{dt}|x_i(t)|\right| = \left|\frac{d}{dt}x_i(t)\right| \leq \sum_{j=1}^k|A_{ij}(t)|\cdot|x_j(t)|+|B_i(t)| \leq \|A(t)\|_\infty\sum_{j=1}^k|x_j(t)|+\|B(t)\|_\infty.
\tag{335}
$$

Let $M(t) = \sum_{i=1}^k|x_i(t)|$, then

$$
\left|\frac{d}{dt}M(t)\right| \leq k\|A(t)\|_\infty M(t) + k\|B(t)\|_\infty,
\tag{336}
$$

hence by Grönwall's inequality,

$$
\max_{t\in[0,T]}\|X(t)\|_\infty \leq \max_{t\in[0,T]}M(t) \leq Tk\exp\left(Tk\max_{t\in[0,T]}\|A(t)\|_\infty\right)\cdot\max_{t\in[0,T]}\|B(t)\|_\infty.
\tag{337}
$$

$\qquad\square$

**Lemma G.3.** *Assume that $X(t) = (x_1(t), \ldots, x_k(t))$ and $Y(t) = (y_1(t), \ldots, y_k(t))$ are solutions of the $k$-order ODE systems*

$$
\frac{d}{dt}X(t) = A(t)X(t) + B(t), \quad \frac{d}{dt}Y(t) = AY(t) + B,
\tag{338}
$$

*respectively. If $X(0) = Y(0) = 0$, $A(t), B(t)$ are continuous with respect to $t \in [0,T]$, $A$ is a fixed $k \times k$ matrix, $B$ is a constant $k$-dimensional vector and*

$$
\max_{t\in[0,T]}\|A(t) - A\|_\infty < \varepsilon, \quad \max_{t\in[0,T]}\|B(t) - B\|_\infty < \varepsilon
\tag{339}
$$

*for some $\varepsilon > 0$, then*

$$
\max_{t\in[0,T]}\|X(t) - Y(t)\|_\infty \leq Tk\left(1 + Tk\|B\|_\infty e^{Tk\|A\|_\infty}\right)(\varepsilon + \|A\|_\infty)\varepsilon
\tag{340}
$$

*Proof.* Denote $Z(t) = X(t) - Y(t)$. Note that

$$\frac{d}{dt}Z(t) = A(t)Z(t) + R(t), \tag{341}$$

where

$$R(t) = (A(t) - A)Y(t) + B(t) - B. \tag{342}$$

By **Lemma G.2**, we have

$$\max_{t \in [0,T]} \|Y(t)\|_\infty \le Tk\|B\|_\infty e^{Tk\|A\|_\infty} \tag{343}$$

hence

$$\max_{t \in [0,T]} \|R(t)\|_\infty \le \left(1 + Tk\|B\|_\infty e^{Tk\|A\|_\infty}\right)\varepsilon. \tag{344}$$

Again, by **Lemma G.2**, we have

$$\max_{t \in [0,T]} \|Z(t)\|_\infty \le Tk\left(\varepsilon + \|A\|_\infty\right)\left(1 + Tk\|B\|_\infty e^{Tk\|A\|_\infty}\right)\varepsilon. \tag{345}$$

$\square$

# H Sphere and Torus

Now we prove **Theorem 5.9**.

*Proof.* Let $0 = \mu_0 < \mu_1 < \dots$ be the eigenvalues of $(-\Delta)^{-1}$, and $\Phi_{i,k}$, $k = 0, \dots, N(i)$ be the eigenfunctions of $\mu_i$:

$$\Delta\Phi_{i,k} = -\mu_i\Phi_{i,k}. \tag{346}$$

Here, $N(i)$ is the dimension of the eigenspace of $\mu_i$. If $M = \mathbb{S}^d$, then $\lambda_i = i(d + i - 1)$, and $\Phi_{i,k}$ are the spherical harmonics; If $M = \mathbb{T}^d$, then $\Phi_{i,k}$ are products of trigonometric functions. Note that $\{\Phi_{i,k}\}$ forms an orthonormal basis of $L^2(\mathbb{S}^d)$, and if the Fourier expansion of $f$ is

$$f = \sum_{i=1}^{\infty}\sum_{k=1}^{N(i)} f_{i,k}Y_{i,k}, \tag{347}$$

then the unique solution to (25) is

$$u = \tilde{u}_p - \sum_{i=1}^{\infty}\sum_{k=1}^{N(i)} \frac{f_{i,k}}{\mu_i}Y_{i,k}, \tag{348}$$

where the constant $\tilde{u}_p$ is selected such that $u(p) = u_p$.

Note that neural tangent kernel $K(x, y)$ is an inner product kernel on $M$: $K(x, y) = F(\langle x, y \rangle)$ for some function $F : [-1, 1] \to \mathbb{R}$, where $\langle \cdot, \cdot \rangle$ is the inner product in $\mathbb{R}^{d+1}$ if $M = \mathbb{S}^d \subset \mathbb{R}^{d+1}$, or the inner product in $\mathbb{R}^{2d}$ if $M = \mathbb{T}^d \subset \mathbb{R}^{2d}$. Then by Funk-Hecke formula (see Section 1.8 of [16] for example), the eigenspaces of $K$ coincide with the eigenspaces of $(-\Delta)^{-1}$, and the Mercer decomposition of $K$ becomes

$$K(x, y) = \sum_{i=0}^{\infty}\lambda_i\sum_{k=1}^{N(i)} \Phi_{i,k}(x)\Phi_{i,k}(y), \tag{349}$$

where $\lambda_0 \ge \lambda_1 \ge \cdots$, $\lambda_i \to 0$ as $i \to \infty$.

For $l \in \mathbb{N}$, let

$$u_l = \tilde{u}_{p,l} - \sum_{i=1}^{l}\sum_{k=l}^{N(i)} \frac{f_{i,k}}{\mu_i}Y_{i,k}, \tag{350}$$

where the constant $\tilde{u}_{p,l}$ is selected such that $u_l(p) = u_p$. Then $u_l \in \mathcal{H}$, and

$$L(u_l) = d(u_l, u^*) = |u_l(p) - u^*(p)|^2 + \int_M |\Delta u_l - \Delta u_p|^2 = \sum_{i=l+1}^{\infty} \sum_{k=1}^{N(i)} |f_{i,k}|^2 \to 0 \quad (351)$$

as $l \to \infty$. Thus,

$$\inf_{u \in \mathcal{H}} d(u, u^*) = 0, \quad (352)$$

which implies that $u^* \in \bar{\mathcal{H}}$. $\qquad\square$

## H.1 Proof of the negative example

*Proof.* Recall that $u^*(e^{i\pi t}) = t^2 - 1$ and $\Delta u^*(e^{i\pi t}) = 2$ for $t \in (-1, 1)$. For any $v \in \mathcal{H}$, let $a_k$ and $b_k$ be the Fourier series of $v$:

$$v(e^{i\pi t}) = a_0 + \sum_{k=1}^{\infty}(a_k \cos(k\pi t) + b_k \sin(k\pi t)). \quad (353)$$

Since $v$ is in the RKHS, then by **Theorem 14**, we can differentiate $v$ term by term:

$$\Delta v(e^{i\pi t}) = \sum_{k=1}^{\infty}(-a_k k^2 \cos(k\pi t) - b_k k^2 \sin(k\pi t)). \quad (354)$$

The PINN distance between $v$ and $u^*$ is

$$L(v) = d(v, u^*)^2 = |v(p) - u^*(p)|^2 + \int_{\mathbb{S}^1} |\Delta v - \Delta u^*|^2$$

$$= |v(p)|^2 + \frac{1}{2}\int_{-1}^{1}\left|\sum_{k=1}^{\infty}(-a_k k^2 \cos(k\pi t) - b_k k^2 \sin(k\pi t)) - 2\right|^2 dt \quad (355)$$

$$= |v(p)|^2 + \sum_{k=1}^{\infty} k^4(a_k^2 + b_k^2) + 4 \geq 4,$$

and the equality holds if and only if $v = 0$. Thus, 0 is the unique minimizer of the loss function, and the population gradient flow is solved by $v_t \equiv 0$, $t \in [0, \infty)$. $\qquad\square$

## H.2 An extra example

In this section, we provide an example where the physics-informed kernel gradient flow is consistent even though the non-homogeneous term $f$ in the problem (25) is not bounded. It is worth noting that in this example, the convergence speed of the PINN loss $L$ can be arbitrarily low.

Consider a special case of the equation (25):

$$\Delta u = f^* \quad \text{on } \mathbb{S}^1, \quad u(p) = 0, \quad (356)$$

where $p = (-1, 0) = e^{-i\pi}$. If $f^*(\theta) = \sum_{q=1}^{\infty} f_q \sin q\pi\theta$, then the solution to (356) is $u^*(\theta) = -\sum_{q=1}^{\infty} f_q q^{-2} \sin q\pi\theta$.

Consider the physics-informed kernel gradient flow estimator $\hat{u}_t$ of an inner-product kernel $k(x, y)$ in the form

$$k(\theta, \varphi) = a_0 + \sum_{q=1}^{\infty}(a_q \cos q\pi\theta \cos q\pi\varphi + b_q \sin q\pi\theta \sin q\pi\varphi. \quad (357)$$

Note that the NTK on $\mathbb{S}^1$ will be in this form. In addition, we assume that $b_q \asymp q^{-\beta}$ for some $\beta > 5$, in order to ensure that $\Delta_\theta \Delta_\varphi k(\theta, \varphi)$ is well-defined.

In this case, the evolution equation of $\hat{u}_t$ is

$$\frac{d}{dt}\Delta\hat{u}_t(\theta) = -2\sum_{j=1}^{N_u} \tilde{k}(\theta, Y_j)(\Delta\hat{u}_t(Y_j) - f^*(Y_j) - \varepsilon_j), \quad (358)$$

where we set the noise $\varepsilon_j$ to be Gaussian ($\varepsilon_j \sim N(0, \sigma^2)$ for some $\sigma^2 > 0$), and

$$\tilde{k}(\theta, \varphi) = \sum_{q=1}^{\infty} \mu_q \sin q\pi\theta \sin q\pi\varphi, \quad \mu_q = q^4 b_q \asymp \gamma, \quad \gamma = \beta - 4 > 1. \tag{359}$$

(We choose $\beta$ sufficiently great such that $k(\theta, \varphi)$ is twice-differentiable and $\tilde{k}(\theta, \varphi)$ is well-defined.) This is exactly the kernel gradient flow equation for classic regression problem (see Example 2 of [45]). Along the gradient flow, $\hat{u}_t(p) = 0$ for any $t = 0$, where $p = e^{-i\pi}$, hence

$$L(\hat{u}_t) = \|\Delta \hat{u}_t - f^*\|_{L^2}^2. \tag{360}$$

It is shown in Section 4 of [45] that the embedding index of $\tilde{k}$ is $1/\gamma$. For any $s > 0$, let $[\mathcal{H}]^s$ be the $s$-interpolation space of the RKHS of $\tilde{k}$. We refer to Section 2.1 of [45] for more information about RKHS, interpolation spaces and relative smoothness.

Finally, by Theorem 1 of [45], the kernel gradient flow is consistent for any $f^* \in [\mathcal{H}]^s$. We note that in this circumstance, $f^*$ is not necessarily bounded.

Moreover, by Theorem 2 of [45], for any $\delta \in (0, 1)$, when $N_f$ is sufficiently great, there exists a function $f^* \in [\mathcal{H}]^s$ such that

$$L(\hat{u}_t) \gtrsim N_f^{-\frac{s\gamma}{s\gamma+1}} \tag{361}$$

with probability at least $1 - \delta$. Since $s > 0$ can be arbitrarily small, we conclude that the convergence speed of $L(\hat{u}_t)$ can be arbitrarily low.

# I   Experimental Results

Although the results of this paper are fully theoretical, we still provide several experimental results to verify the results. Our small experiments were implemented on a personal computer (Legion Y9000P 2023 with CPU i9-13900HX, RAM of 16GB and GPU RTX-4060 Laptop 8GB). The codes were written in Python, and the Pytorch package was used. The execution time was less than an hour.

## I.1   Experiments on one-dimensional interval

In our first experiment, we focus on the 1-dimensional problem:

$$\begin{cases} u''(x) = 2, & x \in (-1, 1); \\ u(-1) = u(1) = 0, \end{cases} \tag{362}$$

whose solution is

$$u^*(x) = x^2 - 1, \quad x \in [-1, 1]. \tag{363}$$

We select the training sample set to be

$$X_1 = -1, \quad X_2 = 1; \tag{364}$$

$$Y_1 = -0.8, \quad Y_2 = -0.6, \quad \dots \quad Y_8 = 0.6, \quad Y_9 = 0.8. \tag{365}$$

Hence the sample size is $N_u = 2$, $N_f = 9$.

Then we train a network $\hat{u}_\theta^m(x)$ in the form (145) depth $L = 1$ and width $m_1 = m$ and activation function $\sigma = \tanh$ to minimize the empirical loss (8). We use the discrete gradient descent to approximate the continuous gradient flow. The training time is set to be $T = 10$, and the learning rate is set to be $10^{-4}$, hence the number of iteration of gradient descent is $10^5$.

Denote as $K_{\theta(t)}^m(x, y)$ the NNK of $\hat{u}_{\theta(t)}^m(x)$, and recall that $K(x, y)$ is the NTK. Our goal is to compare $K_{\theta(t)}(x, y)$ with $K(x, y)$, and compare $\hat{u}_{\theta(t)}(x)$ with the gradient flow $\hat{u}_t(x)$ defined in (11) on the test point sets, which is defined by

$$\mathcal{Z}_{test} = \{Z_1 = -0.9, Z_2 = -0.8, \dots, Z_{18} = 0.8, Z_{19} = 0.9\}, \tag{366}$$

$$\mathcal{Z}_{test}^2 = \mathcal{Z}_{test} \times \mathcal{Z}_{test} = \{(Z_i, Z_j) : i, j = 1, \dots, 19\}, \tag{367}$$

$$\mathcal{T}_{test} = \{t_1 = 0, t_2 = 0.1, t_3 = 0.2, \quad , t_{100} = 9.9, t_{101} = 1\}. \tag{368}$$

By our numerical experiment, if we set $m = M_0 = 10485760$, then the NNK function of $\hat{u}_{\theta(t)}^{M_0}(x)$ satisfies

$$\frac{1}{|\mathcal{Z}_{test}^2|} \sum_{(x,y)\in\mathcal{Z}_{test}^2} |K_{\theta(t)}^{M_0}(x,y) - K(x,y)|^2 \leq 10^{-9}, \tag{369}$$

$$\frac{1}{|\mathcal{Z}_{test}^2|} \sum_{(x,y)\in\mathcal{Z}_{test}^2} |\mathcal{L}_x K_{\theta(t)}^{M_0}(x,y) - \mathcal{L}_x K(x,y)|^2 \leq 10^{-9}, \tag{370}$$

$$\frac{1}{|\mathcal{Z}_{test}^2|} \sum_{(x,y)\in\mathcal{Z}_{test}^2} |\mathcal{L}_x\mathcal{L}_y K_{\theta(t)}^{M_0}(x,y) - \mathcal{L}_x\mathcal{L}_y K(x,y)|^2 \leq 10^{-9}, \tag{371}$$

for any $t \in \mathcal{T}_{test}$, hence it is reasonable to use $\hat{u}_{\theta(t)}^{M_0}(x)$ as an approximation of $\hat{u}_t(x)$:

$$K_{\theta(t)}^{M_0}(x,y) \approx K(x,y), \quad \hat{u}_{\theta(t)}^{M_0}(x) \approx \hat{u}_t(x). \tag{372}$$

For

$$m = 40, 80, 160, 320, 640, 1280, 2560, 5120, 10240, \tag{373}$$

the figure of the mean square distance between $(K_{\theta(t)}^m(x,y), \mathcal{L}_x K_{\theta(t)}^m(x,y), \mathcal{L}_x\mathcal{L}_y K_{\theta(t)}^m(x,y))$ and $(K_{\theta(t)}^{M_0}(x,y), \mathcal{L}_x K_{\theta(t)}^{M_0}(x,y), \mathcal{L}_x\mathcal{L}_y K_{\theta(t)}^{M_0}(x,y))$, that is,

$$\begin{aligned}
t \mapsto &\frac{1}{3|\mathcal{Z}_{test}^2|} \sum_{(x,y)\in\mathcal{Z}_{test}^2} |K_{\theta(t)}^m(x,y) - K_{\theta(t)}^{M_0}(x,y)|^2 \\
&+ \frac{1}{3|\mathcal{Z}_{test}^2|} \sum_{(x,y)\in\mathcal{Z}_{test}^2} |\mathcal{L}_x K_{\theta(t)}^m(x,y) - \mathcal{L}_x K_{\theta(t)}^{M_0}(x,y)|^2 \\
&+ \frac{1}{3|\mathcal{Z}_{test}^2|} \sum_{(x,y)\in\mathcal{Z}_{test}^2} |\mathcal{L}_x\mathcal{L}_y K_{\theta(t)}^m(x,y) - \mathcal{L}_x\mathcal{L}_y K_{\theta(t)}^{M_0}(x,y)|^2
\end{aligned} \tag{374}$$

for $t \in \mathcal{T}_{test}$ is shown in Figure 1;

The plot of the mean square distance between $\hat{u}_{\theta(t)}^m(x)$ and $\hat{u}_{\theta(t)}^{M_0}(x)$, that is,

$$t \mapsto \frac{1}{|\mathcal{Z}_{test}|} \sum_{x\in\mathcal{Z}_{test}} |\hat{u}_{\theta(t)}^m(x) - \hat{u}_{\theta(t)}^{M_0}(x)|^2, \quad t \in \mathcal{T}_{test}, \tag{375}$$

is shown in Figure 2;

The plot of the mean square distance between $\mathcal{L}\hat{u}_{\theta(t)}^m(x)$ and $\mathcal{L}\hat{u}_{\theta(t)}^{M_0}(x)$, that is,

$$t \mapsto \frac{1}{|\mathcal{Z}_{test}|} \sum_{x\in\mathcal{Z}_{test}} |\mathcal{L}\hat{u}_{\theta(t)}^m(x) - \mathcal{L}\hat{u}_{\theta(t)}^{M_0}(x)|^2, \quad t \in \mathcal{T}_{test}, \tag{376}$$

is shown in Figure 3.

## I.2    More experiments on one-dimensional interval for deep networks

Although we only provide proof of Theorem 5.4 (II) for $L \geq 9$, in this section we offer experimental results that verify 5.4 (II) for $L < 9$.

In particular, we set $L = 4$, and take the network to be $\hat{u}_\theta$ defined in (145) with activation function $\sigma = \tanh$, depth $L = 4$ and width $m_1 = m_2 = m_3 = m_4 = m$ for

$$m = 80, 160, 240, 320. \tag{377}$$

The training time is set to be $T = 4$, and the learning rate is set to be $10^{-4}$, hence the number of iteration of gradient descent is $4 \times 10^4$. The network is still trained to solve the problem (362) in the previous subsection.

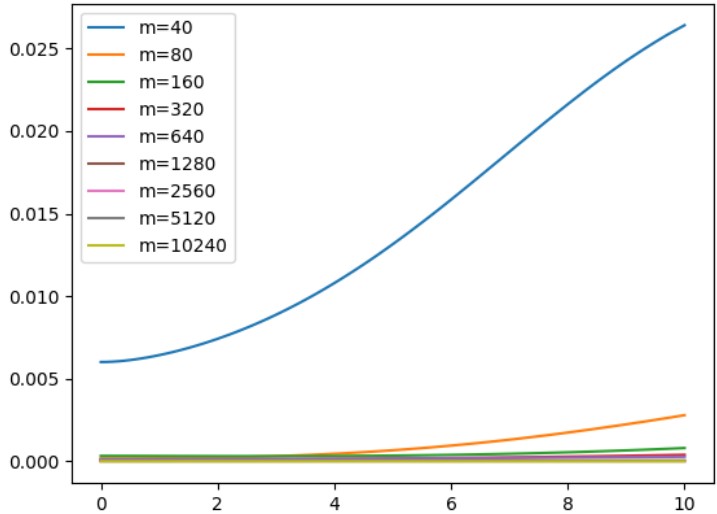

Figure 1: Illustration of Theorem 5.4: The evolution of the distance between $K_{\theta(t)}(x,y)$ and $K_t(x,y)$.
See Appendix I.1 for details.
(x-axis: training time $t$; y-axis: mean square distance between
$(K^m_{\theta(t)},\ \mathcal{L}_x K^m_{\theta(t)},\ \mathcal{L}_x \mathcal{L}_y K^m_{\theta(t)})$ and $(K,\ \mathcal{L}_x K,\ \mathcal{L}_x \mathcal{L}_y K)$ on the test point set)

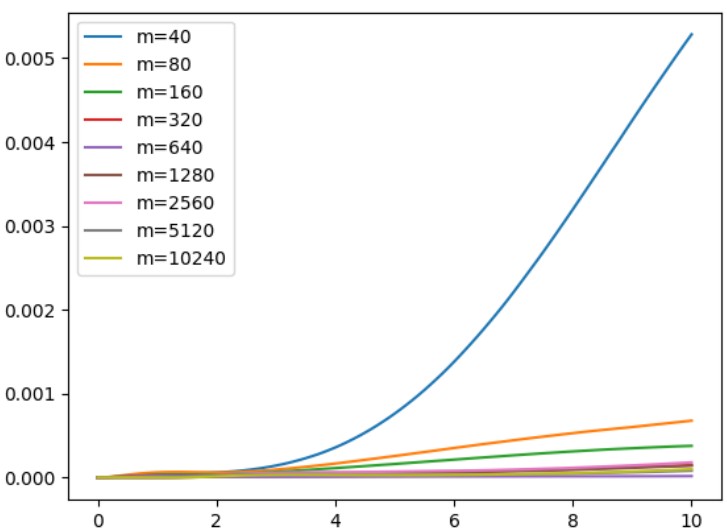

Figure 2: Illustration of Lemma 5.6: The evolution of the distance between $\hat{u}_{\theta(t)}(\cdot)$ and $\hat{u}_t(\cdot)$. See Appendix I.1 for details.
(x-axis: training time $t$; y-axis: mean square distance between $\hat{u}_{\theta(t)}(\cdot)$ and $\hat{u}_t(\cdot)$ on the test point set)

Denote as $K^m_{\theta(t)}(x,y)$ the NNK of $\hat{u}_{\theta(t)}(x)$, and recall that $K(x,y)$ is the NTK. Our goal is to compare $K^m_{\theta(t)}(x,y)$ with $K(x,y)$ on the test point sets, which is defined by

$$\mathcal{Z}_{test} = \{Z_1 = -0.9,\ Z_2 = -0.8, \ldots, Z_{18} = 0.8,\ Z_{19} = 0.9\}, \tag{378}$$

$$\mathcal{Z}^2_{test} = \mathcal{Z}_{test} \times \mathcal{Z}_{test} = \{(Z_i, Z_j) : i, j = 1, \ldots, 19\}, \tag{379}$$

$$\mathcal{T}_{test} = \{t_1 = 0,\ t_2 = 0.1,\ t_3 = 0.2,\quad , t_{100} = 9.9,\ t_{101} = 1\}. \tag{380}$$

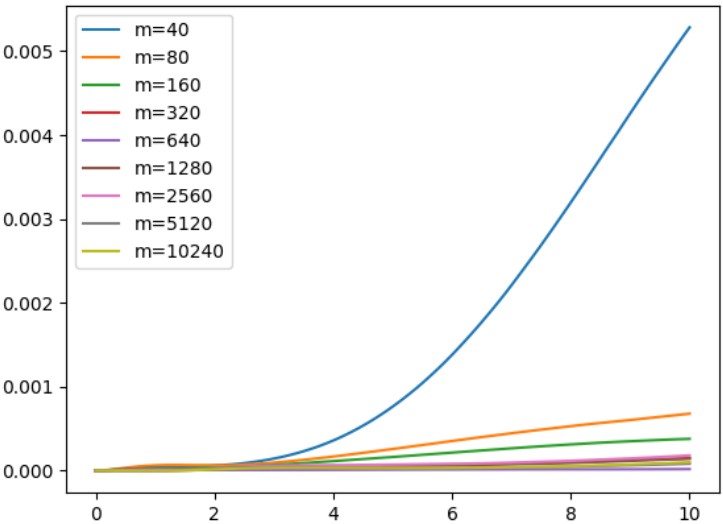

Figure 3: Illustration of Lemma 5.6: The evolution of the distance between $\mathcal{L}\hat{u}_{\theta(t)}(\cdot)$ and $\mathcal{L}\hat{u}_t(\cdot)$. See Appendix I.1 for details.
(x-axis: training time $t$; y-axis: mean square distance between $\mathcal{L}\hat{u}_{\theta(t)}(\cdot)$ and $\mathcal{L}\hat{u}_t(\cdot)$ on the test point set)

A simple numerical simulation reveals that for $M_1 = 10240$,

$$\frac{1}{|\mathcal{Z}_{test}^2|} \sum_{(x,y)\in\mathcal{Z}_{test}^2} |K_{\theta(t)}^{M_1}(x,y) - K(x,y)|^2 \lesssim 10^{-4}, \tag{381}$$

$$\frac{1}{|\mathcal{Z}_{test}^2|} \sum_{(x,y)\in\mathcal{Z}_{test}^2} |\mathcal{L}_x K_{\theta(t)}^{M_1}(x,y) - \mathcal{L}_x K(x,y)|^2 \lesssim 10^{-4}, \tag{382}$$

$$\frac{1}{|\mathcal{Z}_{test}^2|} \sum_{(x,y)\in\mathcal{Z}_{test}^2} |\mathcal{L}_x \mathcal{L}_y K_{\theta(t)}^{M_1}(x,y) - \mathcal{L}_x \mathcal{L}_y K(x,y)|^2 \lesssim 10^{-4}, \tag{383}$$

for any $t \in \mathcal{T}_{test}$, hence it is reasonable to use $K_{\theta(t)}^{M_1}(x,y)$ as an approximation of $K(x,y)$ in the numerical simulations:

$$K(x,y) \approx K_{\theta(t)}^{M_1}(x,y). \tag{384}$$

For $m = 80, 160, 240$ and $320$, the figure of the mean square distance between $(K_{\theta(t)}^m(x,y), \mathcal{L}_x K_{\theta(t)}^m(x,y), \mathcal{L}_x \mathcal{L}_y K_{\theta(t)}^m(x,y))$ and $(K(x,y), \mathcal{L}_x K(x,y), \mathcal{L}_x \mathcal{L}_y K(x,y))$, that is,

$$\begin{aligned}
t \mapsto \frac{1}{3|\mathcal{Z}_{test}^2|} &\sum_{(x,y)\in\mathcal{Z}_{test}^2} |K_{\theta(t)}^m(x,y) - K(x,y)|^2 \\
&+ \frac{1}{3|\mathcal{Z}_{test}^2|} \sum_{(x,y)\in\mathcal{Z}_{test}^2} |\mathcal{L}_x K_{\theta(t)}^m(x,y) - \mathcal{L}_x K(x,y)|^2 \\
&+ \frac{1}{3|\mathcal{Z}_{test}^2|} \sum_{(x,y)\in\mathcal{Z}_{test}^2} |\mathcal{L}_x \mathcal{L}_y K_{\theta(t)}^m(x,y) - \mathcal{L}_x \mathcal{L}_y K(x,y)|^2
\end{aligned} \tag{385}$$

for $t \in \mathcal{T}_{test}$ is provided in Figure 4.

As shown in Figure 4, as $m$ increases, the distance between $K_{\theta(t)}^m(x,y)$ and $K(x,y)(x,y)$ decreases approximately to zero.

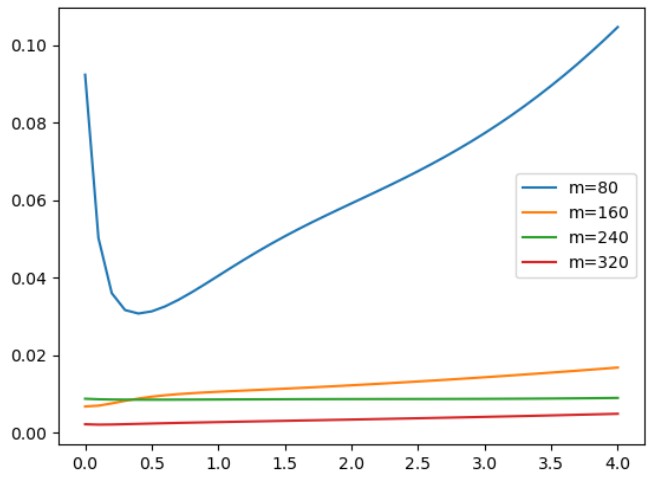

Figure 4: Illustration of Theorem 5.4: The evolution of the distance between $K_{\theta(t)}(x, y)$ and $K(x, y)$. See Appendix I.1 for details.
(x-axis: training time $t$; y-axis: mean square distance between
$(K_{\theta(t)}^m, \mathcal{L}_x K_{\theta(t)}^m, \mathcal{L}_x \mathcal{L}_y K_{\theta(t)}^m)$ and $(K, \mathcal{L}_x K, \mathcal{L}_x \mathcal{L}_y K)$ on the test point set)

### I.3 Experiments on ring

In our second experiment, we consider the following Poisson equation on $\mathbb{S}^1$:

$$\begin{cases} \Delta u(e^{i\pi r}) = r^3 - r, & r \in [-1, 1]; \\ u(p) = 0, & p = e^{-i\pi} = (-1, 0). \end{cases} \tag{386}$$

whose solution is

$$u^*(e^{i\pi r}) = \frac{1}{20}r^5 - \frac{1}{6}r^3 + \frac{7}{60}r, \quad r \in [-1, 1]. \tag{387}$$

The network $\hat{u}_\theta$ is set to be in the form (145) with depth $L = 1$ and width $2m = 3200$ fixed and activation function $\sigma = \tanh$. The learning rate is set to be $0.01$, and the training time is set to be $T = 1000$, hence the number of iteration is $10^5$.

We train the network with different sample size. For sample size $N$, the boundary sample set contains only one element $X_1 = p = e^{-i\pi}$, while the inside sample set contains $N - 1$ elements:

$$Y_j = \exp\left(i\pi(-1 + \frac{2j}{N})\right), \quad j = 1, 2, \ldots, N - 1, \tag{388}$$

hence $N_u = 1$ and $N_f = N - 1$. The test point set is set to be

$$\mathcal{Z}_{test} = \left\{ \exp\left(i\pi(-1 + \frac{j}{10})\right), j = 1, \ldots, 20 \right\}. \tag{389}$$

For sample size $N = 10, 20, 40, 80, 160$, the plot of the networks on test point set after training, that is,

$$r \mapsto \hat{u}_{\theta(t)}(e^{i\pi r}), \quad r \in \mathcal{Z}_{test} \tag{390}$$

is shown in Figure 5.

It is worth noting that in this example, since $\tanh$ is a smooth function whose high-order derivatives are all bounded, then $\mathcal{H}$ is contained in $C^\infty(\mathbb{S}^1)$ (see Corollary 4.36 of [36] or Theorem A.1). On the other hand, the solution $u^*$ is not in $C^3(\mathbb{S}^1)$, hence is not contained in $\mathcal{H}$. However, $u^*$ lies indeed in the larger space $\bar{\mathcal{H}}$, and the model can still fit the solution well, as is shown in Figure 5.

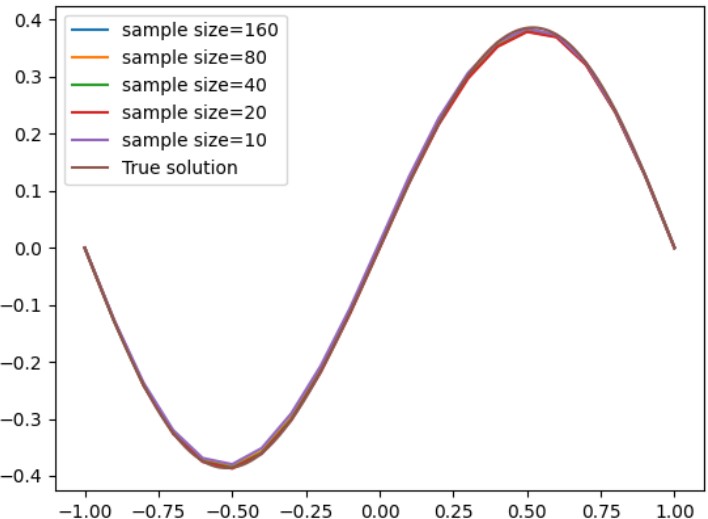

Figure 5: Illustration of Theorem 5.9: The result of PINN trained for Poisson equation problem. (x-axis: $r \in [-1, 1]$; y-axis: the PINN estimator $\hat{u}_{\theta(T)}(e^{i\pi r})$.)

It is revealed in Figure 5 that the sufficiently wide PINN estimator can approximate the true solution $u^* \in \overline{\mathcal{H}}$, even if $u^*$ does not lie in the RKHS of the NTK.

The third experiment is the same with the second experiment, except that we substitute $f(r) = r^3 - r$ with $g(r) = 2$ in the problem setting (386). This is a numerical simulation of the negative example discussed in Subsection 5.1. The result is shown in Figure 6.

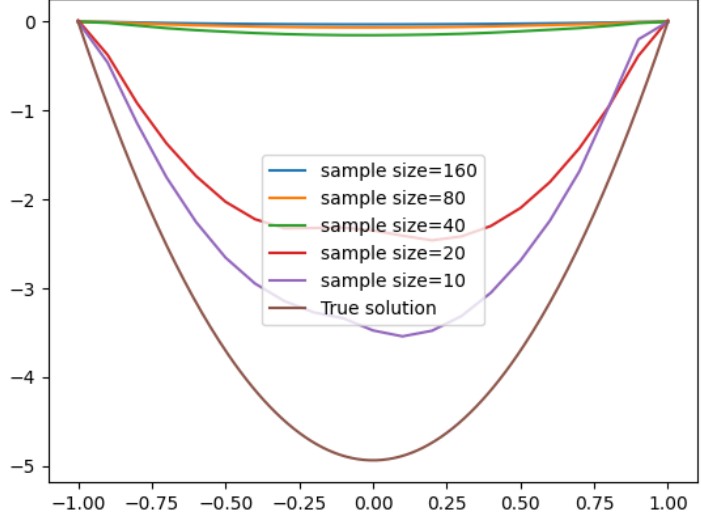

Figure 6: Illustration of Proposition 29: The result of PINN trained for the negative example. As is shown in the figure, the estimator converges to 0 as sample size increases, hence keeps distant away from the true solution. See Appendix I.3 for details.
(x-axis: $r \in [-1, 1]$; y-axis: $\hat{u}_{\theta(T)}(e^{i\pi r})$.)

Figure 6 reveals that when the network width is sufficiently great ($2m = 3200$), as the sample size increases, the PINN estimator gradually converges to 0 and fails to approximate the true solution successfully.

### I.4 Extra experiments on multi-dimensional torus

In this subsection, we provide extra experiments on multi-dimensional torus that illustrate Theorem 5.9.

Let $\mathbb{T}^3 \subset \mathbb{R}^6$ be the three-dimensional torus:

$$\mathbb{T}^3 = \{(e^{i\pi\theta_1}, e^{i\pi\theta_2}, e^{i\pi\theta_3}) : \theta_i \in [-1, 1), \, i = 1, 2, 3\}. \tag{391}$$

Consider the following Poisson equation on $\mathbb{T}^3$:

$$\begin{cases} \Delta u = f & \text{on } \mathbb{T}^3, \\ u(p) = 0 & \text{at } p = (e^{-i\pi}, e^{-i\pi}, e^{-i\pi}), \end{cases} \tag{392}$$

where

$$f(\theta_1, \theta_2, \theta_3) = \sum_{i=1}^{3} (\theta_i^3 - \theta_i). \tag{393}$$

The solution to this problem is

$$u^*(\theta_1, \theta_2, \theta_3) = \sum_{i=1}^{3} \left( \frac{1}{20}\theta_i^5 - \frac{1}{6}\theta_i^3 + \frac{7}{60}\theta_i \right). \tag{394}$$

The network $\hat{u}_\theta$ is set to be in the form (145) with depth $L = 1$ and width $2m = 3200$ fixed, and the activation function is $\sigma = \tanh$. Note that $\tanh$ is smooth, then the NTK function is also smooth. Since the solution $u^*$ is not smooth, then $u^*$ does not lie in the RKHS of NTK by Theorem A.1.

Then we train the network $\hat{u}_\theta$ to learn the solution $u^*$. We repeat this experiment for $50$ times. In each time, we draw $1000$ samples from the uniform distribution of $\mathbb{T}^4$, the learning rate is set to be $0.01$, and the training time is set to be $100$, hence the total number of gradient descent iteration is $10^4$.

Finally, we present the relationship between the training time $t$ and the average PINN loss $L(\hat{u}_{\theta(t)})$ of the $50$ experiments, as well as the region within two standard deviations, as shown in Figure 7.

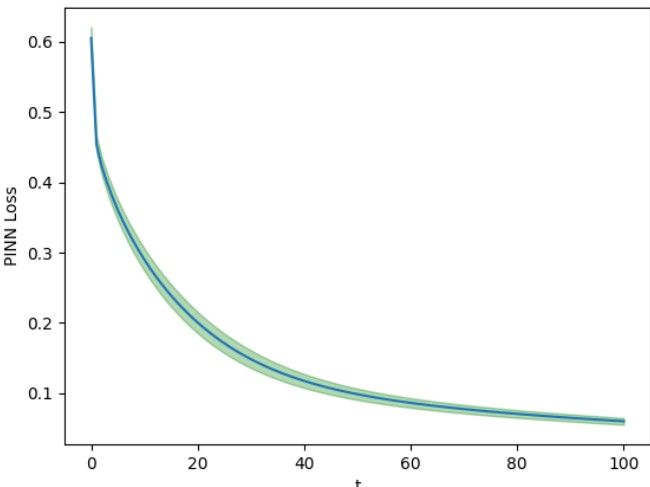

Figure 7: Illustration of Theorem 5.9: The result of PINN trained for the problem (392).
(x-axis: training time $t \in [0, 100]$; y-axis: the PINN loss $L(\hat{u}_{\theta(t)})$)

It is revealed in Figure 7 that PINN can estimate the solution $u^*$ to the problem well, as long as the network width and the sample size are sufficiently great, even though the solution $u^*$ does not lie in the RKHS of the NTK.

