# OpenReview forum: "Consistency of Physics-Informed Neural Networks for Second-Order Elliptic Equations"
_NeurIPS.cc/2025/Conference — NeurIPS 2025 poster_

### Official Review · Reviewer_nmHw · 2025-06-16

**Clarity:** 4
**Significance:** 3
**Originality:** 3
**Rating:** 5
**Confidence:** 4

**Summary:**

This paper consider the consistency of PINNs when applied to second order linear elliptic equations with Dirichlet boundary conditions. The authors establish the sufficient and necessary conditions for the consistency of physics informed kernel gradient flow and sufficiently wide PINN in general cases. The authors also construct a negtive example based on the theoretical results, highlights the risks of directly applying PINNs to specific problems without theoretical assessment of reliability.

**Questions:**

1. Can the results in this paper be extended to nonlinear elliptic equations, or time-dependent PDEs?

2. Can the results be extended to discrete gradient flow? Notice that the convergence of continuous gradient flow and discrete gradient flow have been stated in [1] and related works.
[1] Du S S, Zhai X, Poczos B, et al. Gradient Descent Provably Optimizes Over-parameterized Neural Networks[C]//International Conference on Learning Representations. 2018.

3. What's the essential difference between this work and previous works? The authors give a short comparison in line 38-57, focusing on the training dynamics of networks, but how it affects the convergence result of PINN compared to previous works, especially [27] and [33] in your paper? A table comparison on the assumption, PDEs, convergence results ,etc, is suggested.

4. What about the influence of initialization of neural networks? The paper suggest the parameters are initialized in a particular way in line 100-104, how about the commonly used Xavier initialization in codes, does it affect the results of this work?

5. I have some doubts about the loss (2) $L$ and (3) $\hat{L}$. While I think the main results should be about the emprical loss $\hat{L}$, the continuous loss $L$ is frequently used in this paper. What comes to happen if we replace $L$ by $\hat{L}$ in the main Theorem 3.7 and Lemma 4.1?

**Ethical Concerns:**

["NO or VERY MINOR ethics concerns only"]

**Final Justification:**

All my comments were well addressed. I will keep my positive score.

**Limitations:**

The authors addressed some limitations in the conclusion section.

**Quality:**

4

**Strengths And Weaknesses:**

Strengths:

This paper study the dynamics of the training process and analyzing the convergence of PINNs with large network width.
The theoretical results is intresting and promising, show that the PINN solution depends on the true solution's smoothness and regularity.
This paper is clearly written, well organized, and a pleasant to read, contribute to the theoretical understanding of PINN community.

Weakness:

1. The results are constrained in seccond order linear elliptic equations with Dirichlet boundary conditions, limiting the theoretical guidence of PINN training for different kinds PDEs with different initial/boundary conditions.

2. The concept of consistency and convergence are not well described for non-mathematicians. For AI conference, I suggest using the concept of convergence directly.

3. The results are based on a continuous gradient flow of network training and neural tangent kernel. Practical gradient descent optimizers  are based on discrete gradient flow. However, this work didn't present results for discrete gradient flow.

---

> ### Author Rebuttal · Authors · 2025-07-31
>
> We thank Reviewer nmHw sincerely for your feedback. We greatly appreciate the time and effort you have devoted to reviewing our paper. In response to your questions and concerns, we would like to provide you with the following answers and clarifications:
>
> 1. Concerning other types of PDEs:
>
> We thank the reviewer for this question and would like to provide you with some clarifications on this topic:
>
> 1.1. We are confident that the main results can be extended to other types of linear differential equations, including time-dependent PDEs such as parabolic equations.
>
> The main results include Theorem 3.7, its application to network (Theorem 5.8) and the convergence of NTK (Theorem 5.4). The core derivations in their proofs remain valid for general linear equations.
>
> 1.2. Extension to nonlinear equations is less immediate, because the loss $L$ and the empirical loss $\hat{L}$ are no longer guaranteed to be convex with respect to the RKHS coefficients. This presents an interesting and important direction, which we intend to explore in future work.
>
>
> 2. Concerning discrete gradient flow:
>
> We thank the reviewer for raising this concern. Please let us take this oppotunity to offer some clarifications:
>
> 2.1. The elegant results presented in [1] do not directly adapt to the PINN settings of our work, primarily for the following reasons:
>
> First, the training loss used in PINNs has a non-standard form that differs markedly from the classical mean-square loss, which gives rise to the unusual form of the training gradient flow of PINN (9). As we can see, the equation (9) is extremely different from the evolution equation of network in [1], because (9) requires computing derivatives of the kernel function and handling two distinct sample sets drawn from the boundary and the interior, respectively. Consequently, basic assumptions in [1] such as Assumption 3.1 (positive eigenvalue assumption for kernel matrix) is no longer readily verifiable for PINNs.
>
> Second, the convergence result in [1] is established with respect to the empirical loss, whereas our work is concerned with the convergence of the continuous loss (population loss).
>
> 2.2. In this work, we primarily focus on the continuous gradient flow, whose convergence relies on a delicate analysis of the ill-posed infinite-dimensional convex optimization problem. Whether an analogous statement holds for the discrete scheme is an interesting and very important problem. and we are actively investigating this direction with considerable interest.
>
> [1] Du S S, Zhai X, Poczos B, et al. Gradient Descent Provably Optimizes Over-parameterized Neural Networks[C]//International Conference on Learning Representations. 2018.
>
>
> 3. Concerning comparison with previous works:
>
> We thank the reviewer for this advice, and will add a table comparison of these references. More related references will also be supplied.
>
> In particular, regarding the distinctions between our work and the references [33] and [27]—both of which the reviewer has kindly highlighted—please allow us to clarify:
>
> [27] investigates the convergence of the empirical minimizer of the PINN loss with H\"older regularization term for both second-order elliptic PDEs and second-order parabolic PDEs. The distinctions between [27] and our work are that: First, [27] only studies the empirical risk minimizer, and does not involve analysis of network training, while our work put the network training into consideration; second, the PINN loss in [27] is augmented with a H\"older regularization term, while our work uses the PINN loss without regularization.
>
> [33] investigates the spectral bias of PINN by examining the spectrum of the NTK of two-layer PINN for linear PDE problems with boundary condition (The term "spectral bias" refers to a well-known pathology that prevents deep fully-connected networks from learning high-frequency functions, as discussed in section 5 of [1]). The distinction between [33] and our work is that, [33] does not contain convergence result for PINNs, while our work provides the sufficient and necessary condition for the consistency of PINNs.
>
>
> 4. Concerning initialization:
>
> We thank the reviewer for raising up this question. This is indeed an interesting topic, and we would like to take this opportunity to offer some clarifications.
>
> 4.1. The only purpose we use the special initialization (in line 100-104) is to ensure that the limit gradient flow estimator given by (10) lies in the RKHS of NTK at initial time $t=0$, so that the solution to the kernel gradient flow  (10) is well-defined.
>
> 4.2. The choice of initialization is a delicate matter that needs careful attention in NTK theory.
>
> The special initialization in line 100-104 is suggested by [1], and is popular in NTK related works. [1] also proved that the infinitely wide network will not lie in the RKHS of NTK at initial time, if all parameters are initialized as i.i.d. standard Gaussian random variables, in which case the NTK kernel gradient flow (10) will no longer be well-defined.
>
> 4.3, Therefore, how the popular Xavier initialization has impacts on NTK theory is an interesting topic we are curious about and will perform study on. Thank you again for this valuable question.
>
> [1] Chen G, Li Y, Lin Q. On the impacts of the random initialization in the neural tangent kernel theory[J]. Advances in Neural Information Processing Systems, 2024, 37: 35909-35944.
>
>
> 5. Concerning the loss functions:
>
> We thank the reviewer for this question. We would like to offer some clarifications to address any potential misunderstandings.
>
> 5.1. The empirical loss (or the training loss) $\hat{L}$ is used solely to guide the training of network, whereas the population loss $L(\hat{u})=d_{pinn}(\hat{u},u^* )$ serves to evaluate the distance between the estimator $\hat{u}$ and the true solution $u^*$ in the function space (The metric $d_{pinn}(\cdot,\cdot)$ is defined in (13) in the paper).
>
> 5.2. It is indeed possible to establish a convergence result for $\hat{L}$, similar to Theorem 3.7 and Lemma 4.1, while it is a valuable but different story:
>
> On the one hand, the convergence of the training loss $\hat{L}$ to $0$ indicates that the network can be efficiently trained;
>
> On the other hand, the convergence of $L$ to $0$ means that the estimator converges to the true solution in the function space.
>
> Thank you for raising this point. We hope these clarifications help address your concerns on this topic.
>
>
> 6. Concerning "convergence" and "consistency":
>
> We thank the reviewer for this advice. We will take greater care in using the two terms in the revised version of the paper. Please allow us to offer some clarifications on our use of terminology:
>
> 6.1. We have chosen to use the term “consistency” rather than "convergence" mainly due to the following considerations:
>
> First, our present work focuses on the consistency of PINNs, while we plan to investigate their convergence rate in future research. Thus, we adopt the term "consistency" from learning theory in order to distinguish this work from the existing and forthcoming results that address convergence rates.
>
> Second, the term "consistency" emphasize more on the convergence of the population loss $L$, and we hope to distinguish the convergence results of the population loss in this work from the convergence results of the training loss $\hat{L}$.
>
> 6.2. We will replace many instances of the term "consistency" with "convergence" throughout this paper, including the title, to avoid potential ambiguity and confusion. We will also provide clearer and more precise explanations for the term "consistency" in the revised version of paper. Thank you again for this valuable advice.
>
>
> We hope these clarifications help address your concerns. Thank you once again for taking the time to read our response and for your thoughtful feedback.

---

> > ### Comment · Reviewer_nmHw · 2025-08-01
> >
> > Thanks for responses to my concerns. Most of my comments were well addressed. I will keep my score.
> >
> > For the discrete gradient flow, since this work has involved the training dynamics, I think it is natural and necessary to consider the discrete gradient flow in the future work. Please note that the positive eigenvalue assumption has been solved in PINNs, like [a], [b], [c] and related works.
> >
> > [a] Gradient descent finds the global optima of two-layer physics-informed neural networks. ICML2023
> >
> > [b] Implicit stochastic gradient descent for training physics-informed neural networks. AAAI2023
> >
> > [c] Convergence of implicit gradient descent for training two-layer physics-informed neural networks. arXiv preprint arXiv:2407.02827, 2024

---

> ### Author Response · Authors · 2025-08-01
>
> Thank you very much for your response. We greatly appreciate the references [a,b,c] you kindly offered. These works have solved important problems on the convergence of the empirical loss of PINNs trained via continuous and discrete gradient flows, and they help clarify several of the concerns we initially had.
>
> In particular, Lemmas 3.2 and 3.3 in [a], as well as Lemma 3 in [c], provide theoretical proofs of the positive eigenvalue assumption for PINNs, which we find especially relevant and insightful.
>
> We will properly cite these important contributions in the revised version of our paper. Thank you again for your kind response.

---

### Official Review · Reviewer_Aooh · 2025-06-23

**Clarity:** 3
**Significance:** 2
**Originality:** 3
**Rating:** 4
**Confidence:** 3

**Summary:**

This paper establishes a necessary and sufficient condition for Physics-Informed Neural Network (PINN) consistency in solving second-order elliptic PDEs: convergence occurs if and only if the true solution $u^*$ lies in the closure $\bar{\mathcal{H}}$ of the NTK's RKHS under the PINN metric. The theory is proven via kernel gradient flow and NTK analysis, with non-asymptotic bounds derived under stronger assumptions. A pathological counterexample demonstrates PINN failure when the solution is not in $\bar{\mathcal{H}}$.

**Questions:**

1. In Corollary 3.11, is there a typo in $\hat{u}_{\theta(T)}$? I think it should be $\hat{u}_T$.

2. Is it possible to make Lemma 5.6 more quantitative?

**Ethical Concerns:**

["NO or VERY MINOR ethics concerns only"]

**Limitations:**

The limitations are clearly discussed in Section 5.1 and Section 6.

**Quality:**

3

**Strengths And Weaknesses:**

**Strengths:**
- Establishes necessary and sufficient conditions for PINN convergence ($u^* \in \bar{\mathcal{H}}_{NTK}$).
- Provides non-asymptotic error bounds under stronger assumptions.
- Pathological counterexample reveals fundamental PINN limitations.

**Weaknesses:**
- Verifying $u^* \in \bar{\mathcal{H}}_{NTK}$ is intractable (requires NTK eigenfunctions).
- The experiments are only on a small scale.
- Theory only applies to "infinitely wide networks" (NTK regime).

---

> ### Author Rebuttal · Authors · 2025-07-31
>
> We thank Reviewer Aooh sincerely for your feedback. We greatly appreciate the time and effort you have devoted to reviewing our paper. In response to your questions and concerns, we would like to provide you with the following answers and clarifications:
>
> 1. Concerning the typo in Corollary 3.11:
>
> Yes, it should indeed be $\hat{u}_T$. We are very sorry for the typo, and will correct it in the revision.
>
> 2. Concerning Lemma 5.6:
>
> Yes, it can be reorganized into a more quantitative version. We thank the reviewer for this advice, and will include the quantitative version of Lemma 5.6 in the revision.
>
> For your convenience, we provide a sketch of this quantitative version below.
>
> It will have the following form: With probability $1-\mathcal{O}(m^{\frac{1}{2}})$, the two terms in (23) is in fact less than $Q$, where the quantity $Q$ depends on $m$, $N_u$, $N_f$ and $T$. The explicit computation of $Q$ is given by the right-hand side of (289) in Lemma H.3, and the terms $\|A\|_\infty$, $\|B\|_\infty$ and $\varepsilon$ required in (289) can be found in (279), (280) and (281). Detailed computational results will be updated in the revised version of the paper.
>
> 3. Concerning the intractability of $\bar{\mathcal{H}}$:
>
> We thank the reviewer for raising this question.
>
> We fully agree that the characterization of $\bar{\mathcal{H}}$ is crucial in the analysis of PINN. In this paper, we discuss several situations where the elements of $\bar{\mathcal{H}}$ can be completely characterized.
>
> These situations, which might be of your interest, include: 1. The Sobolev RKHS $\mathcal{H}=H^\beta$, whose closure $\bar{\mathcal{H}}$ is precisely $H^2$ if $\beta>2$; 2. The RKHS of the NTK on spheres or tori, whose closure $\bar{\mathcal{H}}$ is the Sobolev space $H^2$, as is shown in Theorem 5.9.
>
> 4. Concerning the infinitely wide networks:
>
> We thank the reviewer for raising up this question, and we fully understand your concern. Please let us provide you with some clarifications on this topic:
>
> Understanding the dynamics of neural network training is a fascinating and important topic, which we are currently exploring. The training process of networks can be viewed as a form of feature learning. In particular, training infinitely wide neural networks can be interpreted as performing feature learning with fixed, time-invariant features. We believe that investigating and gaining insights from the simplified setting of infinitely-wide networks may serve as a helpful first step toward understanding the more intricate behavior of feature learning for general networks.
>
> With this motivation, our current work focuses on the infinite-width regime, in the hope that it may provide useful intuition and inspire further investigation into the training dynamics of PINNs in more general and practical settings.
>
> 5. Concerning the small scale of experiments:
>
> We thank the reviewer for the insightful advice and fully understand the concerns raised. In the revised version, we will include additional experimental results, particularly for PINNs applied to multi-dimensional PDE problems with more complex solution structures.
>
> We hope these clarifications help address your concerns. Thank you once again for taking the time to read our response and for your thoughtful feedback.

---

### Official Review · Reviewer_yJxF · 2025-07-01

**Clarity:** 3
**Significance:** 3
**Originality:** 3
**Rating:** 5
**Confidence:** 3

**Summary:**

The paper investigates the consistency (convergence property) requirement of PINN when applying to second order elliptic PDEs. Notably $f$ is not required to be in $\mathcal{H}$, but rather in the closure of $\mathcal{H}$ in $H_2(\Omega)$, which is also the necessary and sufficient condition for consistency. Based on the condition, the authors provide a notable pathological example where $f \notin \mathcal{H}$, which makes PINN inconsistent. It is able to deliver a non-asymptotic convergence bound if $f \in \mathcal{H}$.

**Questions:**

- In line 47, it is mentioned that "we provide a different perspective, a new mathematics framework known as NTK ...", actually using NTK to understand theoretical properties of PINN is not new and has been conducted in many works.
- It is mentioned that $f$ does not need to be in $\mathcal{H}$, but only needs to be in its closure w.r.t. $\mathcal{H}^2(\Omega)$, known as $\overline{\mathcal{H}}$, so what are the elements that are in $\overline{\mathcal{H}}$, but not in $\mathcal{H}$? Are there more negative examples can be introduced in section 5.2?
- For the positive example, the Poisson equation is provided which satisfies the codiagonalization assumption, does the theoretical analysis in this paper requires this assumption too?
- Is there any intuition why the theoretical analysis can get rid of the standard capacity and source condition? What aspects make it distinguish from previous work?
- Is there a regime where the equation is "consistent" to solve, but the convergence speed is very low which makes it unsolvable in practice?
- What happens if we conduct analysis on first-order or higher-order PDEs? Some discussions on the necessity of conducting analysis on second-order elliptic PDEs would be beneficial. Can we bring the similar insights to other PDE problems?

**Ethical Concerns:**

["NO or VERY MINOR ethics concerns only"]

**Final Justification:**

The authors have addressed my concerns and I would like to keep my positive score.

**Limitations:**

See weaknesses.

**Quality:**

3

**Strengths And Weaknesses:**

Strengths:
- The authors introduce notion of consistency to area of PINN, which is a weaker notion compared to previous works which establishes non-asymptotic bounds. They also derive necessary and sufficient condition, which is novel and interesting.
- The proofs do not require on standard capacity and source condition, which is standard in previous literature. This is pretty novel in theoretical sense.
- The negative example is quite intriguing and help understanding the properties of PINN.

Weaknesses:
- The analysis is very restrictive since it is only conducted on the second order elliptic PDE.

---

> ### Author Rebuttal · Authors · 2025-07-31
>
> We sincerely thank Reviewer yJxF for your feedback. We greatly appreciate the time and effort you have devoted to reviewing our paper. In response to your questions and concerns, we would like to offer the following clarifications and explanations:
>
> ~
>
> 1. Concerning “a different perspective...”:
>
> We appreciate the reviewer’s attention to this point and apologize for the ambiguity in line 47. Please let us clarify some possible mis-communications.
>
> 1.1. Our original intent was to highlight the distinctions between our work and the previously mentioned references in line 38--45 that focus primarily on analyzing empirical risk minimizers without considering network training dynamics.
>
> For example, [16] establishes convergence rates for statistical and approximation errors of the empirical minimizer of PINN loss for second-order elliptic equations; [27] studies the convergence of the empirical minimizer of the PINN loss with H\"older regularization term for both second-order elliptic PDEs and second-order parabolic PDEs; [8] analyzes the convergence of the empirical minimizer of the PINN loss with Sobolev regularization term for a general linear PDEs. This sequence of works concentrates on convergence analysis of empirical minimizers of the PINN loss (with or without regularization).
>
> In the contrast, compared with these works, our work adopts a different direction by employing NTK theory to incorporate the dynamics of network training into the convergence analysis of wide PINNs. We sincerely hope that this perspective may offer a constructive complement to the existing literature on PINN convergence.
>
> 1.2. We fully agree with the reviewer that there are many related works using NTK to analyze PINN.
>
> For example, [1] investigates the spectral bias of PINN by examining the spectrum of the NTK of two-layer PINN for linear PDE problems with boundary condition (The term "spectral bias" refers to a well-known pathology that prevents deep fully-connected networks from learning high-frequency functions, as discussed in section 5 of [1]); [2] also studies the spectral bias of PINN, while the discussions in [2] were based on analyzing the NTK of multi-layer PINNs for linear PDE problems without boundary condition, concluding that the differential operator in the loss function of PINN does not intensify spectral bias if the boundary condition is absent; [3] extends NTK analysis to PINNs for non-linear PDEs, and suggests that the second-order optimization methods may alleviate spectral bias in training PINNs for non-linear problems. [3] also uses the NTK theory to establish convergence guarantees for the empirical loss of two-layer PINNs under specific assumptions on the network training.
>
> We notice that a relatively small number of the existing NTK-related studies have directly provided consistency results for PINNs. This is where we hope our work could make a meaningful contribution.
>
> 1.3. We appreciate the reviewer’s suggestion and will revise the paper accordingly. Specifically, we will remove the sentence in question, add more related works on NTK and PINN into the references, and include a table comparison of these references.
>
> [1] Wang S, Yu X, Perdikaris P. When and why PINNs fail to train: A neural tangent kernel perspective[J]. Journal of Computational Physics, 2022, 449: 110768.
>
> [2] Gan W, Li Y, Lin Q, et al. Neural Tangent Kernel of Neural Networks with Loss Informed by Differential Operators[J]. arXiv preprint arXiv:2503.11029, 2025.
>
> [3] Bonfanti A, Bruno G, Cipriani C. The challenges of the nonlinear regime for physics-informed neural networks[J]. Advances in Neural Information Processing Systems, 2024, 37: 41852-41881.
>
> 2. Concerning elements in $\bar{\mathcal{H}}$ and negative examples:
>
> We thank the reviewer for raising this question. Please allow us to clarify some key points regarding this topic:
>
> 2.1. We fully agree that the characterization of $\bar{\mathcal{H}}$ is crucial in the analysis of PINN. In this paper, we examine several situations in this paper where the elements of $\bar{\mathcal{H}}$ can be completely characterized.
>
> These situations, which might be of your interest, include: 1. The Sobolev RKHS $\mathcal{H}=H^\beta$, whose closure $\bar{\mathcal{H}}$ coincides with $H^2$ if $\beta>2$; 2. The RKHS of the NTK on spheres or tori, whose closure $\bar{\mathcal{H}}$ is shown to be the Sobolev space $H^2$, as is shown in Theorem 5.9.
>
> 2.2. In addition to the example presented in Section 5.2, it is also possible to construct further negative examples. For your convenience, we provide a brief sketch of such a construction below.
>
> Consider the Poisson equation $\Delta u=f$ on the $d$-dimensional cube $[-\pi/2,\pi/2]^d$, where $f=\cos x_1+\dots+\cos x_d$. Its solution is $u^* =-f$. If the kernel function $k$ is set to be a translation-invariant kernel ($k(x+\pi m,y+\pi n)=k(x,y)$ for any $m,n\in Z^d$) such as the NTK on torus, then its eigenfunctions are Fourier basis functions, and we claim that the integral of the laplacian of any function in $\bar{\mathcal{H}}$ must be zero (the proof is similar with the discussions in section I), hence the solution $u^*$ does not lie in $\bar{\mathcal{H}}$.
>
> 3. Concerning the codiagonalization assumption:
>
> We would like to clarify that this assumption is not required in the main results of this paper, including Theorem 3.7, its application to network (Theorem 5.8) and the convergence of NTK (Theorem 5.4). In fact, it is used only in the construction of the examples in Section 5.
>
> 4. Concerning the capacity-source condition:
>
> We thank the reviewer for raising up this question. We would like to clarify several reasons why the capacity-source condition is not adopted in this paper:
>
> 4.1. The capacity condition states that the eigenvalues $\lambda_i$ of the kernel function satisfies polynomial decay rate $\lambda_i\asymp i^{-\beta}$ ($\beta>1$), while it is not the property that our analysis exploits. Instead, our discussions on PINN rely on the differentiability of the kernel function, hence we simply impose the $C^2$ assumption (Assumption 3.4 and 5.2) directly.
>
> 4.2. The source condition, which states that the target function lies in the $s$-interpolation space the RKHS of the kernel function, is used to evaluate the generalization ability of kernel methods in related literatures. For example, in previous works such as [4,5], the source condition helps determine the generalization error convergence rate of kernel ridge regression. In contrast, our current work focuses primarily on consistency rather than the convergence rate of the generalization error. Moreover, the source condition in the context of PINNs has not yet been well described. We plan to explore this topic further, and aim to derive convergence rates for the generalization error of PINNs in future research.
>
> We hope these explanations are helpful in addressing your concerns on this topic.
>
> [4] Fischer S, Steinwart I. Sobolev norm learning rates for regularized least-squares algorithms[J]. Journal of Machine Learning Research, 2020, 21(205): 1-38.
>
> [5] Zhang H, Li Y, Lu W, et al. On the optimality of misspecified kernel ridge regression[C]//International Conference on Machine Learning. PMLR, 2023: 41331-41353.
>
> 5. Concerning low convergence rate:
>
> We thank the reviewer for this insightful question.
>
> We have successfully constructed such an example where PINN is consistent while the convergence can be arbitrarily low. We will include it in the revision. For your convenience, we provide an outline of the construction below:
>
> Consider the Poisson equation (24) on the one-dimensional ring $S^1$. We take the non-homogeneous term $f$ to be $f(\theta)=\sum_{i=1}^\infty f_i \sin i\theta$. Let $k(\theta,\varphi)=a_0+\sum_{i=1}^\infty(a_i\cos i\theta\cos i\varphi+a_i\sin i\theta\sin i\varphi)$ be a Mercer kernel on $S^1$, and assume that $a_i\asymp i^{-\beta}$ (Note that the NTK on $S^1$ will be in this form). Let $\hat{u}_t$ be the kernel gradient flow estimator defined in (12). Then we claim that, for any $s>1/\beta$, there exists some $f_i$ such that $f_i/(b_i^{s/2} )$ is an $l^2$ and the convergence rate of $L(\hat{u}_T)$ is no greater than $n^{-\frac{s\beta-1}{2s\beta}}$ for any possibly selected stopping time $T=T(n)$ ($n$ is the sample size). Since $s$ can be arbitrarily close to $1/\beta$, then we conclude that the convergence rate $n^{-\frac{s\beta-1}{2s\beta}}$ can be arbitrarily low. A detailed proof will be provided in the revised paper.
>
> 6. Concerning other types of PDE problems:
>
> We thank the reviewer for this question and would like to provide you with some clarificatins on this topic:
>
> 6.1. We are confident that the main results--including Theorems 3.7, 5.4, and 5.8--extend to a broad class of linear differential equations. The key derivations and estimates in the proofs remain valid for the general linear setting with only minor, routine adjustments.
>
> 6.2. We will add discussions on what we benefit from the second-order elliptic equation into the paper. The reasons why we conduct analysis on this type of PDE include:
>
> First, the Green function method of this type of PDE helps us establish the norm control (Lemma 3.9), which provides an estimation for the mean square generalization error of PINN.
>
> Second, for Poisson equations on spheres or tori, PINN satisfies the convenient codiagonalization condition, which enables us to construct the examples, especially the worth-noting negative example in Section 5.2.
>
> Extending these results to other types of PDEs is both an interesting and important problem, and we plan to pursue this direction in future work.
>
> We hope these clarifications help address your concerns. Thank you once again for taking the time to read our response and for your thoughtful feedback.

---

> > ### Comment · Reviewer_yJxF · 2025-08-05
> >
> > Thanks for the detailed response, I would like to keep my positive score.

---

### Official Review · Reviewer_VWqv · 2025-07-05

**Clarity:** 2
**Significance:** 2
**Originality:** 2
**Rating:** 3
**Confidence:** 4

**Summary:**

This paper investigates the consistency of Physics-Informed Neural Networks (PINNs) applied to second-order elliptic partial differential equations (PDEs) with Dirichlet boundary conditions. The authors provide a rigorous theoretical framework built upon the Neural Tangent Kernel (NTK) theory to analyze the training dynamics and convergence of PINNs. The primary contributions of this paper include:

1. The derivation of necessary and sufficient conditions for the consistency of PINNs under the assumption that the PDE solution lies in a certain Hilbert space $\overline{H}$.

2. A non-asymptotic bound on the loss of PINN estimators under stronger assumptions (i.e., the solution lies in the RKHS of the NTK kernel).

3. The construction of a counter example where the PINN method fails to achieve consistency.


While the theoretical results are correct and address important aspects of PINNs, they are limited to the analysis of single-hidden-layer networks. Extending these results to deep neural networks (DNNs) is critical for practical relevance, as DNNs are the dominant architecture in real-world PINN applications.

**Questions:**

The analysis is for two-layer NNs, which are rarely used in practice. Please consider the analysis on deep neural networks.

**Ethical Concerns:**

["NO or VERY MINOR ethics concerns only"]

**Final Justification:**

I would like to keep the score.

**Limitations:**

The results rely on the assumption that the PDE solution lies in a specific Hilbert space $\overline{H}$ or RKHS, is it possible to extend the results in Sobolev spaces?

**Quality:**

3

**Strengths And Weaknesses:**

Strengths:

1. The paper provides a mathematically rigorous framework for analyzing the consistency of PINNs.

2. The use of NTK theory to connect the training dynamics of wide neural networks with kernel methods is a theoretical contribution.

3. The construction of a counter example for the potential failure modes of PINNs and emphasizes the importance of theoretical analysis before applying PINNs to real-world problems.


Weaknesses：
1. Limited Practical Relevance. The analysis is restricted to single-hidden-layer networks, which are rarely used in practice. Modern PINN applications predominantly rely on deep neural networks (DNNs), and the results presented here may not directly extend to such architectures.

2. Strong Assumptions. The results rely on the assumption that the PDE solution lies in a specific Hilbert space $\overline{H}$ or RKHS, which may not hold in many practical scenarios. These assumptions should be carefully validated or relaxed for broader applicability.

---

> ### Author Rebuttal · Authors · 2025-07-31
>
> We sincerely thank Reviewer VWqv for your feedback. We greatly appreciate the time and effort you have devoted to reviewing our paper. In response to your questions and concerns, we would like to offer the following clarifications and explanations:
>
> 1. Concerning multi-layer networks:
>
> We thank the reviewer for the valuable advice regarding the importance of analyzing multi-layer networks, and we fully share your concerns on this point. We would like to take this opportunity to offer some clarifications on this topic:
>
> The two-layer networks was adopted in this paper only for the purpose of brevity of computations. Actually, our results hold for multi-layer networks. We will add a proof for the multi-layer networks in the revision. For your convenience, we provide a sketch of the key steps below:
>
> (I) First, the two-layer or multi-layer structure of the network is required only in the proof of Theorem 5.4 in Section G, which establishes the convergence of NNK to NTK (The NNK and NTK are defined in Section 3.1). Our discussion of multi-layer networks will focus primarily on establishing the multi-layer counterparts of the results presented in Section G.
>
> (II) Section G begins by establishing the convergence of NNK to NTK at initial time. Following the approach taken for the two-layer case, we will first derive an explicit expression for NNK and NTK of multi-layer network that parallels the results in (162)–(163); This representation will then help us prove the desired uniform convergence at initialization.
>
> (III) The remainder of Section G turns to the convergence of NNK to NTK during training. Once again guided by the two-layer argument, we establish—via an analogue of Lemma G.11—an estimation for the evolution of the multi-layer network parameters, which in turn helps us establish uniform convergence of during training.
>
> Thank you for raising this point. We will incorporate the related discussions on multi-layer networks into the revision.
>
>
> 2. Concerning the RKHS assumption:
>
> We thank the reviewer for raising the concern regarding the assumption that the PDE solution lies in the RKHS or in $\bar{\mathcal{H}}$, which may appear strong and difficult to verify in general. We would like to take this opportunity to clarify this point and address any possible misunderstandings.
>
> 2.1 First, we would like to clarify that Theorem 3.12 is the only result in the paper that requires the PDE solution to lie in the RKHS. This result serves merely as a supplementary result. We include it in this paper because it arises as a natural and interesting corollary of the proof of the main theorem.
>
> To better reflect its auxiliary nature, we will relabel it as Corollary 3.12 in the revised version of the manuscript.
>
>
> 2.2. Second, the condition that the PDE solution lies in $\bar{\mathcal{H}}$ plays a fundamental and, in fact, indispensable role in our analysis of PINN.
>
> On the one hand, Theorem 3.7 indicates that if the true solution lies outside $\bar{\mathcal{H}}$, then the PINN estimator cannot converge to it. Consequently, deriving a consistency result relying on the assumption that the PDE solution lies in Sobolev spaces is equivalent to identifying conditions under which a given Sobolev function belong to $\bar{\mathcal{H}}$.
>
> On the other hand, we believe that the space $\bar{\mathcal{H}}$ could be quite large in practice. In particular, when $\mathcal H=H^\beta=W^{\beta,2}$ or $\mathcal{H}$ is the RKHS of the NTK on sphere or torus (cases that might be of your interest), we have shown that $\bar{\mathcal{H}}$ coincides with the Sobolev space $H^2=W^{2,2}$.
>
> We hope these clarifications help address your concerns. Thank you once again for taking the time to read our response and for your thoughtful feedback.

---

> > ### Comment · Reviewer_VWqv · 2025-08-08
> >
> > Thank you for your detailed responses. It is not obvious to extend the result to DNN, I would like to keep the score.

---

> > > ### Author Response · Authors · 2025-08-09
> > >
> > > Here is our outline of proof for DNNS:
> > >
> > > Our goal is to prove an analogue of Theorem 5.4 for multi-layer networks. Consider the network with $L$ layers in the following form:
> > > $$z^1 (x) = W^0 x + b^0 , \quad z^{l+1} = \frac{1}{\sqrt{m_{l} }} W^l \sigma(z^{l} (x)) + b^{l} ,\quad l=1,\dots,L,$$
> > > where the $l$-th layer of the network has width $m_l$. The network is initialized in the form of mirror initialization (we refer to page 6 of [3] for details) in order to ensure that $z^l (x) = 0$ at initial time $t=0$.
> > > Let $\theta_l = (W^1 ,b^1 , \dots, W^l ,b^l )$ be the group of the parameters of the first $l$ layers. The neural network kernel (NNK) of the $l$-th layer is defined to be $K_{\theta,ij}^l (x,y) = \nabla_{\theta_l} z_i^l(x) \cdot \nabla_{\theta_l}z_j^l (y)$. As an analogue of the equality (162) in the paper, NNK is computed in the following form:
> > > $$K_ {\theta,ij}^{l+1}(x,y)=1+\frac{1}{m_l}\sum_{i=1}^{m_l}\sigma(z_i^ {l}(x))\sigma(z_j^{l}(y))+\frac{1}{m_l} \sum_{i,j=1}^{m_L} W_{1i}^L W_{1j}^L \dot{\sigma}(z_i^{l}(x))\dot{\sigma}(z_j^{l}(y))\cdot K^{l}_{\theta,ij}(x,y).$$
> > >
> > > Note that the $(L+1)$-th layer is the output layer with width $m_{L+1}=1$, and we denote $K_ {\theta(t)} (x,y)=K^ {L+1}_ {\theta(t),11}(x,y)$ for simplicity. It is well-known ([2] for example) that at $t=0$, as $m_1,\dots,m_l$ go to $\infty$, $K_ {\theta (0),ij}^ l (x,y)$ converges to a time-independent kernel $\delta_{ij} K^ l (x,y)$ pointwisely, and $K^ l (x,y)$, called the NTK, is determined by
> > > $$K^ 1 (x,y)=R^ 1 (x,y)=1+\langle x,y\rangle,\quad R^ l (x,y)=1+\mathrm{E}_ {f\sim N(0,R^{l-1})} (\sigma(f(x))\sigma(f(y))),$$
> > > $$K^l(x,y)=R^l(x,y)+K^{l-1}(x,y)\mathrm{E}_{f\sim N(0,R^{l-1})}(\dot{\sigma}(f(x))\dot{\sigma}(f(y))).$$
> > > and $z^l(x)$ converges to a Gaussian process $N(0,R^l)$ with covariance function $R^l$.
> > >
> > > 1. Convergence at $t=0$:
> > >
> > > We claim the following lemma:
> > >
> > > Lemma 1. By setting $\sigma=\tanh$, For any multi-index $\alpha$, $\beta$ with $|\alpha|\leq 2$, $|\beta|\leq 2$, we have
> > > $$\nabla_x^\alpha\nabla_y^\beta K_{\theta,ij}^l(x,y)\to\delta_{ij}\nabla_x^\alpha\nabla_y^\beta K^l(x,y)$$
> > > uniformly in probability for $i,j=1,2,\dots,m_l$, as $m_0,\dots,m_{l-1}$ goes to $\infty$.
> > >
> > > Once Lemma 1 is proved, then we immediately obtain that $\mathcal{L}_ x K_ {\theta(0)}(x,y)$, $\mathcal{L}_ y K_ {\theta(0)}(x,y)$ and $\mathcal{L}_ x \mathcal{L}_ y K_ {\theta(0)}(x,y)$ converges uniformly to $\mathcal{L}_x K(x,y)$, $\mathcal{L}_y K(x,y)$ and $\mathcal{L}_x \mathcal{L}_y K(x,y)$, respectively, which is exactly what we need in proving Theorem 5.4 for DDNs.
> > >
> > > The proof of Lemma 1 is by induction on $l$. For $l=0$, it is trivial since $K_{\theta(0),ij}^1(x,y)=\delta_{ij}(1+\langle x,y\rangle)=\delta_{ij}K^1(x,y)$. Suppose that the Lemma holds for $1,\dots,l$. For $l+1$, recall the definition of $K_{\theta,ij}^{l+1}(x,y)$ above, and by induction we have
> > > $$K_{\theta(0),ij}^{l+1} (x,y)\to 1+\frac{\delta_{ij}}{m_l}\sum_{k=1}^{m_l}\sigma(G_k^l(x))\sigma(G_k^l(y))+\frac{1}{m_l}K^l(x,y)\sum_{k=1}^{m_l}W_{ik}^lW_{jk}^l\dot{\sigma}(G_k^l(x))\dot{\sigma}(G_k^l(y)),$$
> > > where $G_k^l\sim N(0,R^l)$ is a Gaussian process with covariance $R^l(x,y)$. Then, by weak law of large number, by taking $m_l\to\infty$, we have $K_{\theta(0),ij}^{l+1} (x,y)\to\delta_{ij}K^{l+1}(x,y)$ pointwisely.
> > >
> > > In order to prove the uniform convergence, we apply Lemma B.2 of [3], and what we still need to prove is that for any $\delta>0$, we have $\|\nabla_x^\alpha\nabla_y^\beta K_{\theta(0),ij}^{l+1} (x,y)\|_{C^0}\leq C$ for some $C>0$ independent of $m_0,\dots,m_l$ with probability at least $1-\delta$. This inequality is checked by tedious computations similar with page 9--11 of [4].

---

> ### Author Response · Authors · 2025-08-09
>
> Thank you for sharing your concerns on the discussions of DNNs.
>
> I. We agree that extending our results from two-layer networks to deep neural networks (DNNs) is indeed nontrivial.
>
> The structure of DNNs is considerably more complex, and we have found that some of our computations—particularly those involving network derivatives—cannot be directly extended from the two-layer setting to the multi-layer case.
>
> II. To address this challenge, we follow a sequence of related works such as [1,3,4] which have developed effective tools for analyzing the convergence properties of the derivatives and NTKs of DNNs. For example, Lemma B.2 of [4] provides a powerful framework to extend pointwise convergence result to uniform $C^k$ convergence result for stochastic processes, which is well-suited for analyzing the NTKs of PINNs.
>
> III. Following these prior works, we have successfully completed a proof for DNNs. We will add the detailed discussions into the revised version of paper. Thank you again for raising this problem.
>
> For your convenience, we are happy to share additional technical details of our proof. We provide you with an outline in the next comment.
>
> [1] Huang K, Wang Y, Tao M, et al. Why Do Deep Residual Networks Generalize Better than Deep Feedforward Networks?---A Neural Tangent Kernel Perspective[J]. Advances in neural information processing systems, 2020, 33: 2698-2709.
>
> [2] Jacot A, Gabriel F, Hongler C. Neural tangent kernel: Convergence and generalization in neural networks[J]. Advances in neural information processing systems, 2018, 31.
>
> [3] Li Y, Yu Z, Chen G, et al. Statistical optimality of deep wide neural networks[J]. arXiv preprint arXiv:2305.02657, 2023.
>
> [4] Gan W, Li Y, Lin Q, et al. Neural Tangent Kernel of Neural Networks with Loss Informed by Differential Operators[J]. arXiv preprint arXiv:2503.11029, 2025.

---

> ### Author Response · Authors · 2025-08-09
>
> 2. Convergence during training.
>
> In order to prove the uniform convergence of NNK to NTK during training, it suffices to prove that the network parameters evolves at a very low speed when the network has very large width. The following lemma is an analogue of Lemma G.11 in our paper:
>
> Lemma 2. Suppose that $m_l>m$, $l=1,\dots,L$. Then for $m$ sufficiently great, with probability $1-O(m^{-\frac{1}{2}})$ (the randomness arises from the initialization of network parameters), we have
> $$\sup_k\sup_{t\in[0,T]}|\theta_k(t)-\theta_k(0)|=O(\frac{m_1^2+\dots+m_L^2}{\sqrt{m_1\cdots m_L}}),$$
> where the first supreme is taken over all network parameters $\theta_k=W_{ij}^l$ or $b^l_i$, and the convention $O$ hides all the constants and the terms involving $\log m$.
>
> The proof of Lemma 2 is based on the evolution equations of parameters:
> $$\frac{d}{dt}\theta_k=-\frac{2}{N_u}\sum_{i=1}^{N_u}\frac{\partial z^{L+1}(X_i)}{\partial\theta_k}\cdot z^{L+1}(x_i)-\frac{2}{N_f}\sum_{j=1}^{N_f}\frac{\partial\mathcal{L} z^{L+1}(Y_j)}{\partial\theta_k}\cdot(\mathcal{L} z^{L+1}(Y_j)-f(Y_j)).$$
> We first note that for any $n=1,\dots,d$, by direct computations we have
>
> $$\frac{\partial}{\partial x_ n} z^ {L+1} (x)=\frac{1}{\sqrt{m_ 1 \cdots m_ L }}W^ L W^ {L-1} \cdots W^ 1 W^ 0_  n \cdot \dot{\sigma} (z^ L (x)) \dot{\sigma} (z^ {L-1} (x))\cdots \dot{\sigma} (z^ 1 (x)),$$
> where $W^0_{\cdot,n}$ is the $n$-th column of $W^0$. Then we have
> $$\frac{\partial}{\partial W^L_{1k}}\left(\frac{\partial}{\partial x_n} z^{L+1}(x)\right)=\frac{1}{\sqrt{m_1\cdots m_L}}W^ {L-1}_ {k,\cdot}\cdots W^1 W^0_{\cdot,n}\cdot\dot{\sigma}(z^L(x))\dot{\sigma}(z^{L-1}(x))\cdots\dot{\sigma}(z^1(x))$$
>
> Define $\|M\|=\sum_{i,j}|M_{ij}|$ for matrix $M=(M_{ij})$. Note that $\sigma=\tanh$ has bounded derivative $|\dot{\sigma}|\leq C$. Then we can prove that
> $$\left|\frac{\partial}{\partial W^L_{1k}}\left(\frac{\partial}{\partial x_n} z^{L+1}(x)\right)\right| \leq\frac{\|W^{L-1}\|\cdots\|W^0\|}{\sqrt{m_1\cdots m_L}}\leq\frac{C}{\sqrt{m_1\cdots m_L}}(1+\sum_{l,i,j}|W^l_{ij}|^\alpha)$$
> for some constant $C$ and $\alpha$ depending only on $L$. Likewise, we can also prove that
> $$\left|\frac{\partial}{\partial W^l_{ij}}\left(\frac{\partial}{\partial x_n} z^{L+1}(x)\right)\right|\leq\frac{C}{\sqrt{m_1\cdots m_L}}(1+\sum_{l,i,j}|W^l_{ij}|^\alpha),$$
> $$\left|\frac{\partial}{\partial W^l_{ij}}\left(\frac{\partial^2}{\partial x_n\partial x_m} z^{L+1}(x)\right)\right|\leq\frac{C}{\sqrt{m_1\cdots m_L}}(1+\sum_{l,i,j}|W^l_{ij}|^\alpha),$$
> for all $l,i,j$. Then we can use these inequalities to prove that
> $$\left|\frac{\partial}{\partial t}\sum_{l,i,j}\left|W^l_{ij}(t)-W^l_{ij}(0)\right|\right|\leq\frac{C(\log m)^\beta(m_L+m_Lm_ {L-1}+\cdots+m_1m_0+m_0d}{\sqrt{m_1\cdots m_L}}(1+(\sum_{l,i,j}|W^l_{ij}(t)-W^l_{ij}(0)|)^\alpha)$$
> with high probability $1-O(m^{-\frac{1}{2}})$ for some constants $C$, $\alpha$, $\beta$ depending only on $L$. Finally, using the Gronwall's inequality, we prove the Lemma for the parameters $W^l_{ij}$.The proof for $B^l_i$ is similar.
>
> Finally, since the NNK function $K_{\theta}(x,y)$ is explicitly decided by the parameters $W^ {l}_ {ij}$ and $B^l_i$, we can use Lemma 2 to prove that $K_ {\theta} (x,y)$ converges uniformly in $C^2$.

---

### Note · Authors · 2025-08-13

Dear reviewers and ACs,

Thank you very much for the time and effort you have devoted to reviewing our paper. We would like to offer some clarifications on the key points discussed.

1. Concerning multi-layer networks:

1.1. One of the major concerns that reviewers have raised is the lack of discussions on multi-layer networks.  At first, we adopted the two-layer setting only for brevity. We agree that extending our original proofs directly to multi-layer networks is not straightforward, mainly due to the nontrivial form of the PINN loss--especially the involvement of network derivatives.

1.2. Nevertheless, we have completed a proof for multi-layer PINNs by following a sequence of prior works. For your convenience, we have also provided an outline of our proof in our previous responses.

We will add these discussions on multi-layer networks into the revised version of paper.

2. Concerning comparison with previous works:

2.1. Our study builds upon a series of existing researches on PINNs and NTK theory, while the results we obtain are distinct from the existing literature.

Previous works have investigated a variety of related topics, including the convergence rates of empirical minimizers of PINN loss, the convergence of empirical PINN loss function during network training, and the relationship between NTK and the spectral bias of PINN.

We will add more previous works on these topics into the references. A table comparison will also be added in the paper for clarity.

2.2. To the best of our knowledge, our paper presents the first result that state the sufficient and necessary condition for the consistency of PINN. It is a very general result proved under very mild assumptions. This is where we hope that our results could make contributions to the existing literature.

3. Concerning other updates in the revision:

Besides the above two points, we plan to make several updates, including but not limited to: 1. We will add an example into the paper, where PINN is consistent but the convergence rate could be arbitrarily low. 2. We will provide more precise explanations for the term "consistency", and use it more carefully in this paper. 3. We will provide experiments under more complicated settings.

Once again, we sincerely thank the reviewers for their careful reading and insightful feedback, which we believe will significantly improve the quality and clarity of our work.

---

### Decision · Program_Chairs · 2025-09-17

**Decision:**

Accept (poster)

**Comment:**

**Summary**
This paper investigates the consistency of Physics-Informed Neural Networks (PINNs) when applied to solve second-order elliptic partial differential equations (PDEs) with Dirichlet boundary conditions. The core contribution is the establishment of a necessary and sufficient condition for the consistency of the PINN method, framed within the infinite-width Neural Tangent Kernel (NTK) theory. This condition states that a PINN is consistent if and only if the true PDE solution lies in the closure of the NTK's Reproducing Kernel Hilbert Space (RKHS) under the PINN metric. The paper further derives non-asymptotic loss bounds under stronger assumptions and constructs an insightful pathological counterexample that demonstrates the failure of PINNs when this condition is not met。

**Strengths of the Paper**
1. The paper addresses a foundational question regarding the reliability of PINNs by being the first to establish a necessary and sufficient condition for their consistency. This lays a solid theoretical groundwork for understanding when PINNs can be expected to work.
2. The study employs a rigorous mathematical framework, leveraging kernel gradient flow and NTK theory to analyze the training dynamics of PINNs.
3. The authors construct a concrete example where the PINN method is inconsistent, which clearly illustrates the practical implications of their theoretical findings and highlights a key failure mode of PINNs.

**Weaknesses of the Paper**
1. The paper's most significant initial weakness was its focus on shallow networks, which limits practical relevance. While the authors have provided a credible proof sketch for extending the results to DNNs, the full proof has not yet been peer-reviewed
2. The analysis relies on the infinite-width NTK regime and a continuous gradient flow, which are idealizations of practical finite-width networks trained with discrete optimizers. Furthermore, verifying the core assumption about the true solution's membership in the RKHS closure can be intractable in practice.
3. The experiments presented were noted to be on a small scale, and the authors have been encouraged to include more complex, multi-dimensional problems.

**Summary of the Review Process**
The paper was thoroughly evaluated by four reviewers, with final ratings consisting of one "Borderline reject" (3) , one "Borderline accept" (4) , and two "Accept" (5). All reviewers acknowledged the paper's theoretical rigor and novelty , particularly appreciating the fundamental nature of the necessary and sufficient condition and the insightful counterexample that clarifies the limitations of PINNs.

The primary points of concern revolved around the scope and applicability of the theoretical analysis. The most critical weakness, raised by Reviewer VWqv, was that the initial analysis was restricted to two-layer (single-hidden-layer) networks, limiting its practical relevance as deep networks are dominant in practice. Other concerns included the analysis being confined to a specific class of PDEs , the reliance on strong assumptions that may be difficult to verify (e.g., the solution lying in a specific Hilbert space) , and the idealization of using an infinite-width network in a continuous gradient flow setting.

The authors provided a very detailed and technical rebuttal. To address the critical issue of multi-layer networks, they stated that a proof for deep neural networks (DNNs) has been completed, and they provided a comprehensive proof sketch to support this claim. They also offered sound justifications for other points, such as the rationale for focusing on second-order elliptic PDEs and their plans to expand the experiments in the revised version.

The authors' rebuttal was successful in convincing the majority of the reviewers. Reviewers yJxF and nmHw maintained their high scores after the rebuttal, and Reviewer Aooh also expressed satisfaction. The sole exception was Reviewer VWqv, who, despite the detailed response, remained unconvinced about the extension to DNNs and maintained their "Borderline reject" score.


**Recommendation: Accept**
Despite some limitations, the theoretical contributions of this paper are significant and foundational. It provides deep insights into a core question in the important and growing field of PINNs. The authors addressed the reviewers' main concerns convincingly in their rebuttal, particularly the critical point of extending their theory to deep networks, for which they provided a detailed technical sketch. While one reviewer remains skeptical, the overall consensus is positive.